


**Do sun spots influence the onset of ENSO and PDO events in the Pacific Ocean?**
*Franklin Isaac Ormaza-González[1], and María Esther Espinoza-Celi[1]*
1)  ESPOL Polytechnic University, Escuela Superior Politécnica del Litoral, ESPOL, (Facultad de
5        Ingeniería Marítima, Ciencias Biológicas, Oceánicas y Recursos Naturales), Campus Gustavo
6        Galindo Km. 30.5 Vía Perimetral, P.O. Box 09-01-5863, Guayaquil, Ecuador

Corresponding author: formaza@espol.edu.ec.
The sea surface temperature (SST), anomalies (SST), ONI (Oceanographic El Niño Index) and MEI
(Multivariate ENSO Index) in regions El Niño 1+2 (80°W-90°W, 0°-10°S) and 3.4 (5°N-5°S, 170°W-
120°W) as well as the Pacific Decadal Oscillation (PDO) and Atlantic Multidecadal Oscillation (AMO)
indexes were correlated to sun spots number (SS) from cycles (SS#) 19 to 24 (1954-2017). Degree-six
polynomial regression functions represented each of the six cycles with an average  $r^2$>0.89
(p<0.001). The series of correlations at different lag times (0, 6, 12, 24, 36 and 48 months) gave a
response time: 12-36 months. In the 1954-2017 period, the whole series of SS cycles did not show a
strong correlation with the variables and SST Anomaly in the El Niño areas 1+2 and 3.4. The highest
correlations $r^2$ were up to: 0.043, 0.029, 0.040 and 0.021 for PDO, MEI, ONI and SST Anomaly (in 3.4)
respectively, suggesting that there is still a correlation with high confidence (p≤0.01). Analysing for
the period 1990-2016, the correlations improved up to 0.11, 0.12, and 0.17 for ONI, SST (in 3.4) and
PDO correspondingly. The SST correlations against individual SS cycles in regions 3.4 and 1+2  were
up to 0.219 (SS# 23) and <0.0675 (SS# 19) correspondingly. SST Anomaly, ONI and MEI correlated
with $r^2$ of 0.250, 0.3943 and 0.2510, one-to-one; the lag time was 24-48 months and linear curves
had positive slope. In general, in 1+2 there was found a more inconstant and lower correlation than
in 3.4 (where also MEI and ONI are measured). On the longer time scales, the PDO (alike AMO)
seemed to respond in 36-48 months to SS cycles showing a high degree of correlation coefficient $r^2$
of 0.625 (SS# 19) and 0.766 (SS# 24); whilst AMO index gave up to 0.490 (SS# 20) with similar lag
time. Cycles 19 and 24 showed a better correlation in general. During the ascending phases of each
cycle the SST in region 1+2 rendered correlation coefficient $r^2$ and p-value from 0.205 and 0.0008
(SS# 23) to 0.163 and ≤0.0044 (SS# 19). In the region 3.4, $r^2$ were from 0.870 (SS# 24), to  0.556 (SS#
23). In each SS cycle lag time was around 36 months; all of them occurred at the ascending phase,
except in cycles 20 and 24. SST anomaly registered $r^2$ from 0.662 to 0.254 in the ascending phase
with a response time 0-48 months and positive linear regression slope (except SS# 23). On the other
hand, the descending phase showed a predominantly lower $r^2$: < 0.14 (p<0.01). The region 3.4 had
better $r^2$ than in 1+2, from 0.897 (SS# 24) to 0.239 (SS# 21) respectively in the ascending phase
except cycles 20 and 24. The lag time was consistent at 36 months. The highest $r^2$ of 0.897 at the end
of the SS# 24 peak, coincided with one of the strongest El Niño (2014-2015) and the second highest
$r^2$ (SS# 22 ascending phase) with two consecutive strong El Niño 1991-1995. ONI and MEI also
showed strong correlation; the three highest $r^2$ matched dates of strong El Niño 1987-1989, 1955-
1957 and 1997-1998 in the ascending phase. During the descending phases, the correlation
coefficients were lower, and ranged from 0.6082 to 0.2938; but with a lag time 0-12 months, and
positive slope. The index MEI, as with ONI, registered $r^2$ lower during descending phases. The PDOs
were linearly correlated from 0.7677 to 0.2855  (12 to 24 months)  and 0.3522 during ascending and
descending phases respectively. On the other hand, $r^2$ for AMO was up to 0.700. The strength of the
linear correlations substantially increased when the ascending and descending phase of each cycle
was analysed. During the ascending phase there is a stronger correlation than in the descending
phase. These results would indicate that warm events tend to occur in the ascending phase or at the
top of the cycle and have a delay time of around 36 months, whilst cold events are associated to a



descending phase but with a quicker response time. The sun spot activity should be considered as a
factor that could condition and trigger low (PDO and AMO) and high (ONI-El Niño) frequency
oceanographic events in the Pacific and Atlantic Oceans. During 2019, the cycle 25 should start, then
according to this work probably the next El Niño event would be around 2020-2021 or later.

**Key words:** Sun spots cycles, SST, SSTA, ONI, MEI, PDO, AMO, El Niño, La Niña





## Introduction.

Essentially, the only external source of energy to Earth is the sun which constantly radiates a flux of energy at a rate of 1360 W m$^{-2}$ or 1.36 kJ m$^{-2}$ s$^{-1}$ (Monteith, 1972) or 1.92 ly day$^{-1}$ (Ormaza-González and Sanchez, 1983) to the upper external atmosphere of Earth; also called the solar constant. Recently, Kopp and Lean (2011) have reported that the most accurate accepted solar constant value of 1360.8 ± 0.5 W m$^{-2}$ is lower than the canonical digit of 1365.4 ± 1.3 W m$^{-2}$, which was established in 1990. Of this flux of energy, 75-50 % reaches Earth and sea surface (Ormaza-González and Sanchez, 1983; Lindsey, 2009) after it is reflected and/or absorbed by clouds, particles, gases, etc. (Horning et al., 2003). 90-93% of that surface reaching energy is accumulated by the oceans (Trenberth et al., 2014; Clutz, 2017).  The solar constant is affected by the variable sunspot number (SS, among other solar activity parameters) at around 0.1%, i.e. 1.361 W m$^{-2}$ or 1.365 W m$^{-2}$. The Hale cycle (around 11 years) is characterized by the increasing and then decreasing SS number (Hathaway, 2015). Froelich (2013) suggested that the solar constant can vary up to 4.0 W m$^{-2}$ in two sun SS cycles, i.e. 22-year cycle, and proposed a simple relationship between SS and solar constant (SC), by assuming a direct relationship between SC and SS

*SC= 1353.6 + 0.089 (SS)*                    (r$^2$ of 0.71, 95-99% confidence).

The oceans store heat, alternately releasing and absorbing such energy. This system is basically placed at the surface-subsurface of the oceans that interacts with the lower atmosphere. One extensive work of Zhou and Tung (2010) reported the impact of SC on global SST along 150 years, finding signals of cooling and warming SSTs at the valley and peak of the SS cycles; although Schlesinger and Ramankutty (1994) did not imply an external force such as the SS, they reported a global cycle of 65-70 years that is possibly affected by greenhouse anthropogenic gases, sulphate aerosols and/or El Niño events. There are well known processes that are roughly periodic with low (25-30 years) or high (3-5 years) frequency events such as the Pacific Decadal Oscillation (PDO, Mantua et al., 1997; Mantua and Hare, 2002; Zhang et al., 1997; Yim et al., 2013), Atlantic



Multidecadal Oscillation (AMO, Enfield et al., 2001; Condron et al., 2005;  Gray et al., 2010) and
Interdecadal Pacific Oscillation (IPO, Henley et al., 2015); and El Niño (Busalacchi et al., 1983,
see **COAPS Library's: http://www.coaps.fsu.edu/lib/biblio/coaps-a.html**) or La Niña (Yuan and
Yan, 2012), respectively. During El Niño events, the surface and subsurface lose energy to the
atmosphere and the opposite during La Niña (Trenberth et al. 2014, Fasullo and Nerem, 2016) with
annual and interannual periods, while the decadal processes may take 25-30 years. The Interdecadal
oscillations have a series of impacts; e.g., the PDO gives rise to teleconnections between the tropic
and midlatitudes (Yoon and Yeh, 2010), and affects: 1) the ocean heat content (Wang et al., 2017), 2)
the lower and higher levels of trophic chain and small pelagic fisheries like tuna and sardines
(Ormaza et al., 2016a, 2016b), 3) biogeochemical air-sea $CO_2$ fluxes (McKinley et al., 2006),  4) the
frequency of la Niña/El Niño (Newman et al. 2003), etc. The interactions between decadal
oscillations PDO/IPO and AMO may affect also the ocean heat content (Chen and Tung, 2014).  All
these low and high frequency oceanographic events have a direct impact on local, regional and
global climate patterns; there is some evidence that the driving source of energy is the sun (Grey et
al., 2010). Thus, Huo and Xiao (2016) have found positive strong correlation between El Niño 2015-
2016 and SS; afterwards, they also found a strong correlation between the latter and El Niño Modoki
index (Huo and Xiao, 2016). White et al., (1997) reported that heat anomalies produced by variable
solar irradiance are stored in the upper layer driving SST changes of 0.01-0.03 K and 0.02-0.05 K on
decadal and interdecadal periods respectively; also later Zong et al. (2014), in their review about the
impact of SS 11-year cycle and the multidecadal climate projections, have found global SST
variations of 0.08 ± 0.06 K and 0.14 ± 0.02 during the 11 and 22 years Hale Cycle, and that there is a
response lag of 1-2 years in relation to the SS (see also, Kristoufek, 2017). Liu et al. (2015) have
reported that apart from volcanic eruptions, effective solar radiation plays a role in the modulation
of decadal ENSO-like oscillation. More recently, Yamakawa et al. (2016) have reported that solar
activities in terms of SS numbers not only affect troposphere but also the sea surface.
Acknowledging that the SS is only a partial parameter to measure solar activity (Scafetta, 2014), this


work attempts to understand how the sunspots could affect low and high frequency oceanic events
such as the Pacific Interdecadal (PDO), the Atlantic multidecadal oscillation (AMO), Sea Surface
Temperatures, its anomalies and El Niño and La Niña events.

**Material and methods.**
Data for monthly **sun spot number** (SS) was taken from the Royal Observatory of Belgium, Brussels,
World Data Center SILSO (http://www.sidc.be/silso/datafiles). Data sources for other variables were
as follow:  El Niño regions areas 3.4 (5°North-5°S, 170-120°W) and 1+2 (0-10°S, 90°W-80°W):
• **Sea surface temperatures (SST) and SST Anomaly**: The Monthly Extended Reconstructed
Sea Surface Temperature Version 4 (ERSSTv4, 1981-2010 base period). The Optimum
Interpolation 1/4 Degree Daily Sea Surface Temperature (OISST.v2, 1981-2010 base period),
http://www.cpc.ncep.noaa.gov/data/indices/.
• **Oceanic Niño Index** (ONI: Huang et al., 2014): ERSST.v4  for El Niño/La Niña events since
1950 till December 2017:
http://www.cpc.ncep.noaa.gov/products/analysis_monitoring/ensostuff/ensoyears.shtml.
• **Multivariate ENSO index** (MEI: Wolter and Timlin, 2011):
https://www.esrl.noaa.gov/psd/enso/mei/table.html.
• **Pacific Decadal Oscillation** (PDO, based on Mantua Index): The PDO index is based on
NOAA's extended reconstruction of SSTs (ERSST Version 4). It is constructed by regressing
the ERSST Anomaly against the Mantua PDO index for their overlap period, to compute a
PDO regression map for the North Pacific ERSST Anomaly. The ERSST Anomaly are then
projected onto that map to compute the NCEI index. The PDO index closely follows the
Mantua PDO index at: https://www.ncdc.noaa.gov/teleconnections/pdo/ (Wolter and Timlin
1993, 1998 and 2011).



• **Atlantic Multidecadal Oscillation** index:

https://www.esrl.noaa.gov/psd/data/timeseries/AMO/.

All indexes have data from April 1954 to December 2017. The analysis was done using Excel
and/or R statistical tools. The correlation exercises were executed using SS solar cycles as
complete time series against SST Anomaly (in 3.4 and 1+2), ONI, MEI, AMO and PDO indexes.
Correlations with lags of 0 to 48 months were carried out. For the SS cycles 19-23 and their
impact on the above mentioned dependent variables, correlations were carried out for the
whole time series 1954-2017, for 1990-2016,  for each cycle and for their respective ascending
and descending phases. Spectral analysis and polynomial regression fitting curves were
determined to obtain the slope of the ascending phases; the slopes were correlated to the
oceanographic indexes.





**Results and discussion:**

The time series (1954 to 2016) of SS, PDO, AMO, ONI and MEI are shown in Fig. 1. PDO, AMO, ONI

and MEI start at time: 0, 12, 24 and 36 months (panels a, b, c and d respectively); whilst SS series

starts at t=0 in the four panels. It has been reported the lag time responses to SS cycles of some

indexes are around 12-36 months (e.g., Zhao et al., 2014). From 1954 to the present, there has

occurred the cycles 19 to 24; cycles have a period of around 11 years (Hathaway, 2015); Dicke (1978)

established 11.2 years. The highest peak is seen in cycle 19 with around 250 SS/month; then, the

next cycle goes down to <150, but the peak of cycle 21 jumps to around 200, to decrease steadily

from cycle 22 to 24 to just over 100 SS/month. Cycle 24 is one with the lowest contemporary peaks

of a SS cycle comparable only to cycles 12-15 (around 1880-1930), and the lowest in the last 200

years (Clette et al., 2014). The negative or cold PDO phases (1947-1976, 2000-june/2016) are within

SS cycles 19 -20 and 23-24, but cycles 21 and 22 are within the positive or warm phase of the PDO

(1977-1999). As PDO and AMO indexes are displaced from 0 to 36 months on the time scale, some

peaks and troughs can be seen; these are at ascending and descending parts of the SS cycles; thus,

during cycles 19-20 and 23-24 PDO indexes are basically negative, and on the contrary during 21-22;

the exception is around 1990, where there is a strong negative peak. However, AMO phases seem to

be in opposition to and overlapping the SS cycles; a cold phase of AMO is between 60s and 90s,

whilst the warm one is from the 90s to the present (McCarthy and Haigh, 2015).

The ONI and MEI curves, both indicators of ENSO events, behave similarly throughout the study

period (April 1954 – December 2017). However, MEI has the highest anomaly peaks (> 2) compared

to ONI, specifically in: 1995, 1980 and mid-2016. In general, ONI and MEI curves indicate the highest

positive anomalies between 1978 and 1995, a period that coincides with the warm and cold phases

of PDO and AMO respectively. The opposite occurs before and after this period due to the inversion

of phases. In addition, the highest peaks of both indexes only occur during the ascending and

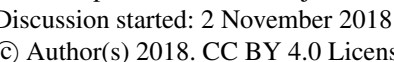

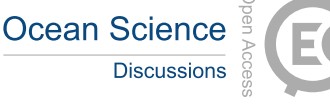

descending phases of the solar cycles; that is, they never coincide with the maximum period of
sunspots of the cycles. The two highest MEI peaks occur during the descending phase of solar cycle
21 and ascending phase of solar cycle 23. In the mid-2016 ( cycle 24) both indexes increased
reaching the third highest peak of this period during the descending phase of the solar cycle 24. On
the other hand, negative peaks of these indexes are noted to occur either in the high or low plateau
of the SS curves.

The number (N) of data in the analysis were: 765 (1965-2017); 312 (1990-2016); 108 (1990-1999);
(2000-2016). For individual cycles 19 to 24: 127, 141, 124, 117, 141 and 120 respectively. In the
same order for ascending-descending phases: 48-80, 50-92, 43-82, 33-85, 51-51 and 74-47.  The
degrees of freedom of residual were N-2. The degree of correlation in terms of Pearson coefficient is
referred as: High, moderate and low when its ±0.5 and ±1.0, ±0.3 to ±0.49 and ±0.29 respectively
(http://www.statisticssolutions.com/pearsons-correlation-coefficient/). All linear regression
residuals were autocorrelated using the Durbin-Watson (DW) test (Montgomery et al., 2001); for
1954-2017, 1990-2016, 1990-1999, 2000-2016, individual cycles, and ascending-descending phases.
The DW  tests for the long time series averaged 0.122, for individual cycles varied from 0.10 to 0.63
with an average of 0.18, and for the ascending and descending phases averaged 0.40 and varied
from 0.1 to 2.24.  SST Anomaly in region 1+2 has the lowest and PDO the highest.

**The whole series (1954-2017) correlations.** All variables (Table 1) were correlated in a linear and
polynomial (n= 2 to 6 order) basis using different lag times (0, 6, 12, 24 and 48 months) along the six
SS cycles. Polynomial correlation (not shown) as well as linear ones displayed poor correlation
coefficients; however, for the latter, the highest $r^2$ (p≤0.01) coefficients were found to occur at lag
time somewhere between 12 and 24 months, except for AMO (36-48 months). For SST and SST
Anomaly in 1+2 there was found no correlation. These results agree somewhat with Kristoufek



(2017), who suggested a surface thermal response of around 24-36 months. The highest correlation
$r^2$ values were up to: 0.043, 0.029, 0.040 and 0.021 for PDO, MEI, ONI and SST Anomaly (in 3.4)
respectively; suggests that there is still a correlation with high confidence (p-value ≤0.01), though of
small $r^2$. This fact could be explained as the sun activity (sun spots) in the long run balances
throughout the ups and downs of the cycles. This exercise would suggest there is not a good
correlation on these indexes in the Pacific and Atlantic at the studied time scale; nonetheless, on
longer time series, where SS are affected by other sun internal processes; e.g., the alleged Minimum
of Maunder (Eddy, 1976, Shindell et al., 2001, Ineson et al., 2015, Mörner, 2015, etc.) can impact on
a global and regional basis. Recently, Lockwood (2010 and 2013) has reported that a grand solar
minimum is coming as the behavior of the SS cycle 24 is developing; i.e., it is being observed that for
the last 9300 years, there has not been such solar activity decline as found in cycle 23 to 24. This fact
gives a likely occurrence of a solar minimum that may last through cycles 24, 25 and 26 (Hady, 2013).
Under these circumstances, it was decided to analyze correlations using individual cycles, from the
19 to 24.

**Period 1990-2016.** Further analysis was carried out using the period 1990 to 2016, that includes
cycles 22, 23 and 24. The time series was also split in 1990-1999 and 2000-2016, because during
1990-1999: a strong (1991-1992), moderate (1994-1995) and the strongest El Niño (1997-1998) of
the XX century occurred. On the other hand, in 2000-2016 (cold phase PDO) also a strong a La Niña
(2000-2002 and 2010-2012) and El Niño Modoki in 2015 (Huo and Xiao, 2016) were registered.  The
Figure 4 shows again a poor $r^2$: <0.011 (p>0.246), for the SST Anomaly in region 1+2 (blue bars),
although this region was gravely affected by the strong El Niño 1997-1998 which brought hundreds
of casualties and billions of US dollar losses to the Ecuadorian infrastructure (Glantz, 2001). The
linear correlation $r^2$ of SST in 3.4 (red bars) was around 0.1193  (p≤ 0.00001) in the whole period,
whilst a bit higher in the period 1990-1999: 0.1519 (p≤ 0.01). The ONI (green bars) was up to 0.1436


(p≤0.02) correlated to SS in the period 1990-2000 where high positive SST Anomaly were present for
almost 6 years. ONI correlated better than SST Anomaly in 3.4. The Pacific Decadal Oscillation (Fig.
4., grey bars) had an $r^2$ of 0.276 (p<0.0001), in the period 2000-2016 (PDO in a cold phase), with a
Pearson correlation of 0.523 that can be considered as high
(https://www.statisticssolutions.com/pearsons-correlation-coefficient/) . However, for the period
1990-1999 it was 0.239 and for the whole period was 0.402; i.e., a poor and fair degree of
correlation respectively.

**Individual Cycles.** Correlation analysis was split into SS cycles, from 19 to 24. The SS and SST $r^2$
correlation coefficient indicated poor correlation and confidence (p≥0.05) in region 1+2 in all cycles
(Fig. 2); most of the correlation $r^2$ were <0.050, except in cycle 19, where a $r^2$ of 0.0675 (p=0.0032);
in cycles 21 and 23 the highest $r^2$  was 0.046 (p=0.0173) and 0.048 (p=0.037) respectively. The lag
time varied between 6 to 36 months. In region 3.4 correlation were variable and rendering a  $r^2$ up to
0.219 (p≤0.0001)  and 0.213 (p≤0.0001) for cycles 23 and 19 respectively, with a lag time of 12-36
months with an exception in cycle 19 (Fig. 2), where the highest coefficient was after 6 months lag
time. Cycles 20 and 22 had $r^2$ of 0.105 (p≤0.0001) and 0.074 (p=0.003) in the same order.  The slopes
of the linear regression curves with the highest $r^2$ were positive in region 3.4, indicating a direct
correlation between SST and SS cycles. However, cycles 22 and 23 in the region 1+2 exhibited
inverse correlation (Fig. 2). Further polynomial correlation (n=2 to 6) analysis did not render any
substantially better $r^2$. In general, higher correlation was found in 3.4 than in 1+2.

**Anomalies SST.** SST Anomaly can change in terms of the reference used; there are 5 versions of ERRS
(Huang et al., 2017). These versions are nowadays used almost in every study related to El Niño;
currently is the version 5. In this work we used the ERRSv4 (Huang et al., 2014); Huang et al. (2017)
stated that there is not a noticeable difference between ERRSv4 and ERRSv5. The anomalies of SST in



3.4 and 1+2 were also correlated against every cycle; correlation $r^2$ was not found better than 0.396
(p≤0.0001) in both regions, with higher variability in 1+2 than 3.4 either in response time and
correlation coefficient (Fig. 3). In region 3.4, the highest correlations were 0.289 (p≤0.0001) and
0.270 (p≤0.0001) during cycles 19 and 23 respectively, with a lag time between 12 and 36 months,
and both during cold phases of PDO (1955-1978, and 2000-present). Surface winds plus other
oceanographic variables (e.g.; upwelling) could play an important role in this high variability; these
winds are not only generated in this area but farther away, even by the trade winds of the western
Atlantic (Ormaza-González and Cedeño, 2017). Also, ENSO processes in the western Pacific could add
variability in the SST Anomaly. The slopes of the linear correlation were basically positive for 3.4 and
negative for 1+2, similar as for SST correlations for cycles 19, 23 and 24 (cold PDO phase) for the
highest $r^2$. Again, the anomalies in 3.4 were better correlated than in 1+2 region.

**ONI.** The El Niño index (Fig. 5) displayed  $r^2$ values from around 0.053 (p=0.01, cycle 22) up to 0. 25
(p<< 0.0001, cycle 24), in a poor to fair correlation with a positive slope mainly in SS#: 19, 23 and 24.
During SS# 24,  ONI reached high values of 2.6C (Nov-Dec-Jan 2015/2016) and -1.7C (Oct-Nov-Dec
2010). The highest $r^2$ were again found again somewhere 24-48 months lag time. Cycle 21 did not
show any confident correlation with ONI; however, cycles 20, 22 and 24 had $r^2$ of 0.144 (p<<0.001),
0.131 (p<<0.0001) and 0.252 (p<<0.0001) respectively, with lag times of: 48, 12, and 24 months
respectively. Recently, Huo and Xiao (2016) found strong correlation between SS and El Niño Modoki
during 2015 (SS# 24). The variability of the $r^2$ could arise from: 1) SS number importantly varying
from one month to another, 2) regional meteorological conditions (particularly cloudiness), ocean
surface currents that exchanges heat of the region 3.4, Kelvin waves (Gill, 1982), the Southern
Oscillation Index (SOI: Southern Oscillation Index: http://www.cpc.ncep.noaa.gov/data/indices/soi),
etc., that in turn affect the SSTs; and, on top of that, 3) the way ONI is obtained; i.e., ONI has variable
reference period of 30 years; thus for 1950 to 1955 the reference period is 1936-1965; for 1956-



1960; 1941-1970; the ERRSv4 uses the period 1981-2010.  The reference period is moved every 5
years (Lindsey, 2013); the most recent ONIs (v4/v5) are supposed to have better and more
consistent data as equipment acquisitions improve in time.

**MEI.** This index, which is another index for El Niño events,  correlated at lower $r^2$; thus, the highest
value was 0.3943 (p<<0.00101, SS# 19), the next 0.3028 (p<<0.00001, SS# 24), 0.2421 (p<<0.00001,
SS# 23) and 0.1566 (p<<0.0001, SS# 20); in cycles 21 and 22 there was not found correlation better
than 0.1232 (p<<0.0001). The lag time ranges from 24-48 months, and linear regression curves were
with mainly positive slope.

**PDO.** This interdecadal index (Fig. 5) is linearly correlated to SS cycles somewhere between 36 and
48 months lag time, with highest $r^2$  of 0.391 (p<<0.00001) in cycle 19, and 0.586 (p<<0.00001) for
cycle 24), with strong correlation: 0.625 and 0.762 respectively. Both cycles are within the cold
phase PDO. The next highest $r^2$ with p-values <<0.0001 were 0.218, 0.1361, 0.218 and 0.260 for
cycles 20-23. In all cycles, the highest $r^2$ were positively correlated, except cycle 20. For some
reason, there appears a better fit with both PDO and ONI in cycles 19 and 24, which are within the
period of cold phase of PDO, even though these cycles have remarkable different shape and peaks
(Fig. 12); cycle 19 registered SS counts over 250 whilst cycle 24 just around 100; also, the plateau of
their peak was different: very sharp and extended respectively.

**AMO.** This index proved to render correlation coefficients $r^2$ up to 0.490 (p<<0.00001) and down to
0.162 (p=0.0004) in cycles 20 and 24 respectively) with a lag time of 48 months, which is the elapsed
time where the highest correlation was found in cycles 20, 23 and 24, whilst at cycle 19, 21 and 22
the elapsed time was 24-36 months. Gray et al. (2016) reported lag time response of mean-sea-level





pressure over the Atlantic to SS cycles of 36-48 months, in a longer time series study of 32 solar
cycles. The Fig. 6 shows the bar distribution of the $r^2$; it displays linear regression with  positive and
negative slopes for cycles 19, 23 and 24; and 20 to 22, respectively. This coincides with the phases of
the AMO, negative from around 1965 to 1998 (SS# 20-22), and positive; 1930-1965 (SS# 19) and
after 1998-present (SS# 23-24), http://appinsys.com/globalwarming/amo.htm . It is noteworthy to
say that the slopes of the PDO and AMO linear regression show to be negative/positive respectively
in cycle 21 and 22, but in concordance in cycles 19, 20, 23 and 24 whose periods correspond to the
cold phase PDO.
**Ascending and descending phases** of solar cycles. As the SS cycles, which last about 11.2 years,
impact on variables studied on a response time from 24 to 36 months, there was the need to study
their influence during the ascending and descending phases, which have roughly 5-6 years duration.
Polynomial regression analysis was performed to stablish a function that could best describe every
SS cycle.  Sixth-order polynomial curves (Fig. 12) were found to render a very strong correlation
coefficient averaging 0.89 (p≤0.001). The functions allowed to analyse the correlations in the
ascending and descending phases.

**SST in 1+2 and 3.4.** In region 1+2, the highest correlation coefficient $r^2$ and p-value were 0.205 and
0.0008 (SS# 23), 0.189 and ≤0.0036 (SS# 21), and 0.163 and ≤0.0044 (SS# 19). All linear regression
coefficients $r^2$ over 0.0847 (p<0.05 to =0.0008) occurred in the ascending phase of the SS cycles, but
lower ones occurred in the descending phase, with no definite lag time pattern from 0 to 36 months.
The slope (positive/negative) of the linear regression (Fig. 7) curves showed no pattern. These low
and variable $r^2$ reflect that region 1+2 is subjected to the conjunction of many diurnal and seasonal
oceanographic and meteorological variables; for example, during the first quarter of 2017 (cycle 24),
in 1+2 there was higher than usual SST because the southern trade winds in eastern Pacific
weakened and the North Atlantic ones strengthened; thus, these passed through the Panama





Isthmus, and blew warm (up to 30C) surface waters from Panama Bay down south to 1+2 provoking
a rapid and relatively short lived surface warming (Ormaza-González and Cedeño, 2017), while
region 3.4 was registering La Niña conditions. This cold event also strengthens the Cromwell
Undercurrent (Knauss, 1959) and Humboldt (Montecino and Lange, 2009) currents related to
upwelling processes in 1+2. During the ascending phases of the cycles, the correlation of SSTs was
higher than in the descending phase of cycles. All these facts would mask the SS signal in this area.

In the region 3.4, the maximum $r^2$ of SST in each SS cycle was found at lag time of 36 months; all of
them occurred at the ascending phase, except in cycles 20 and 24. The four highest $r^2$  were 0.870
(p=0.021, 24), 0.613 (<<0.0001, 22), 0.574 (p<<0.0001, 19), and 0.556 (p<<0.0001, 23); i.e., the
Pearson coefficients were: 0.9327, 0.7803, 0.7576 and 0.7456, respectively, which show strong
degree of linear correlation. Linear regression slopes were variable (Fig. 7), although there was a sort
of tendency in cycles 20, 21 and 22 (warm PDO) for negative slopes and for positive slopes for cycles
19, 23 and 24 (cold phase PDO). In 3.4 the SST response to SS was much clearer than 1+2, as in this
region (10N-10S and 120W-180W) there is not influence of coastal processes. The highest $r^2$ (0.870,
p=0.021; lag time 36 months) in the descending phase in cycle 24 coincided with the strongest El
Niño, and the second-highest $r^2$ (0.613, p<<0.00001) during ascending phase of cycle 22 with two
consecutive strong El Niño 1991-1995; the third $r^2$ (0.574, p<<0.00001) during cycle 19, with el Niño
1955-1957, and finally the fourth $r^2$ (0.556, p<<0.00001) with 1997-1998 warm event during cycle 23.
It seems that short time expressions of SS cycles, either on their initial ascending or descending
phases, trigger effects on the SSTs.

**SST Anomaly.** In region 1+2  (Fig. 8), the anomalies registered high $r^2$  (p<<0.0001) of 0.662 (SS# 22),
0.637 (SS# 19), 0.523 (SS# 21), 0.480 (SS# 23), 0.359 (SS# 24); and 0.254 (p=0.0002, SS# 20)
respectively, in the ascending phase of the SS cycles and with a positive linear regression slope



(except SS# 23). The response lag time was somewhere between 0 to 48 months. On the other hand,
the descending phase showed a predominantly lower $r^2$, less than 0.14 with lower significance ($p \leq$
0.02), with the exception in SS# 19 , 0.304 ($p<<0.0001$). The results indicate that during cold phase
PDOs, as the surface ocean waters in 1+2 are relatively colder, the correlations tend to be higher.

**SST Anomaly in 3.4.** There was a high and consistent $r^2$ (Fig. 8) that reached up to 0.897 ($p=0.014$;
SS# 24); 0.863 ($p<<0.0001$; SS# 22); 0.665 ($p<<0.0001$; SS# 19), 0.826 ($p<<0.0001$; SS# 23), then fell
to 0.211 ($p=0.008$; SS# 20); 0.239 ($p=0.0009$; SS# 21) respectively; all of them were in the ascending
phase except cycles 20 and 24. The lag time was consistent at 36 months. Linear regression slopes
were variable (Fig. 8); negative slopes in cycles 20-22 (warm phase PDO); and positive slopes in 19,
23 and 24 cycles (cold phase PDO). The highest $r^2$ of 0.897 in the initial moment of the descending
phase in 24 coincided with one of the strongest El Niño and the second $r^2$ (SS# 22 ascending phase)
with two consecutive strong El Niño 1991-1995. The third and fourth highest $r^2$ were during El Niño
1955-1957 and 1997-1998 warm event (SS# 23 ascending) respectively. The results are suggesting
that SS cycles are strongly correlated to SST Anomaly in both El Niño regions, but much more in 3.4.

**The ONI index.** This index as well as SST and its anomalies in 3.4, were equally strongly associated
with the ascending phase of the SS cycles (Fig. 9); with a lag time of 24-36 months and the highest
correlation $r^2$ per each cycle were in the ascending phase; the predominant linear regression curve
slopes were positive, except SS# 20. The highest $r^2$ ($p<<0.0001$) were: 0.817 (SS# 22), 0.693 (SS# 19),
0.637 (SS# 23), 0.3547 (SS# 24), 0.2876 (SS# 20); 0.1936 ($p=0.003$, SS# 21). The three highest $r^2$
match with the dates of strong El Niño 55-57, 87-89, and 97-98 (Fig. 9) with positive slope and
ascending phase. The ascending phase coincides with El Niño, or after 2-3 years the peak or valley of
the cycle. In the descending phase the $r^2$ ($p<<0.0001$) in cycles 24, 23, 22 and 20 with 0.366, 0.284,


0.255, and 0.242 respectively. All of them have a lag time 0-12 months and positive slope. Cycles 19
and 21 showed neither strong correlation (<0.1) nor confidence (p=0.2).

Warm events tend to occur in both ascending/descending phase after the peak/trough, and have a
delay time of 36 months, which is like what was found by Huo and Xiao (2016). The descending
phase of the cycles (Fig. 9), with a smaller slope than the ascending phase, produce quicker
responses (0-12 months) of the ocean surface SST and ONI that could trigger neutral or cold events
more cogently; most of la Niña events occur during the descending phase or approaching the cycle
minimum (Fig. 10). The weakest SS# 24 has had three La Niña: 2007-2009, 2010-2012, 2016-2017
(Fig, 12). A plausible reason is that during this cycle the number of sun spots (i.e., sun activity) is the
lowest in the last two centennials (Clette et al., 2014); therefore, less energy has hit the ocean
surface producing a cooling effect. Two important exceptions are La Niña 1988-1989 (22) and 2000-
2002 (cycle 23) that occurred in the ascending phase.

**The MEI index.** The Multivariate ENSO Index does not only consider the SST Anomaly but also sea-
level pressure (Allan and Ansell, 2006) and other variables. These variables include surface winds
(meridional and zonal), surface air temperature and total cloudiness fraction of the sky (Wolter and
Timlin, 1998). The MEI correlated at slightly lower levels with SS cycles with $r^2$: 0.784 (p ≤ 0.0001),
0.770 (p<<0.0001), 0.5972 (p ≤ 0.0001); 0.3396 (p ≤ 0.0001); 0.2368 (p=0.0003); and 0.222 (p=0.001)
for SS cycles 19, 22, 23, 24, 20, and 21, respectively.  All of them in the ascending phase of the cycles
with a lag time from 12 to 48 months (except cycles 23 and 24), with a positive linear regression
slope; except 22 and 20 where the $r^2$ was largest with zero lag. During the descending phase, like the
ONI, the $r^2$ were lower: 0.321 (p=0.0004, SS# 24); 0.3145 (p<<0.0001, SS# 19); 0.2234 (p<<0.0001,
SS# 22); 0.2088 (p<<0.0001 , SS# 20), and 0.1438 (p=0.0002 , SS# 23) with positive slope (except 20)
and lag time, predominantly 0-48 months; cycle 21 did not have a $r^2$ above 0.010 (p>0.02). For the





index MEI, as with ONI, the $r^2$ were much lower during descending phases. The descending phases
showed to be also associated with la Niña, thus resembling somehow what was found with ONI. The
lower correlations could be because MEI has more influencing factors than ONI, which could obscure
the signal from the sun irradiation.

**PDO.** The Pacific Decadal oscillation linear correlation with SS gave larger $r^2$ in most of cycles except
in cycle 20 (0.2589; $p \leq 0.0002$) and 21 (0.2855; $p=0.0002$); thus 0.7677 ($p \leq 10^{-12}$), 06577 ($p \leq 10^{-12}$),
0.6734 ($p \leq 10^{-7}$) and 0.5062 ($p \leq 10^{-7}$) for the ascending phase SS# 19 (Apr/54-Nov/58), SS# 24
(Jan/08-Feb/14), SS# 22 (Sep/86-Jan/89) and SS# 23 (May/90-Jun/00) respectively. All these
coefficients were obtained at a lag time of 12-48 months, except 22 and 23 (t=0). The slopes of the
linear regressions were mainly positive during cold phase PDOs (cycles 19, 23 and 24), except cycle
20 when a cold PDO was transitioning to a warm PDO (21 and 22).  The figure 10 shows that linear
correlations in cycles 19, 21, 23 and 24  showed positive  slopes. The two highest $r^2$ are at lag time of
12-36 months, for cycles 19 and 24, as have been reported in other works (e.g., Huo and Xiao, 2017).
During the descending phase, the correlation $r^2$ tended to be much lower, with the highest 0.3522
($p<<0.00001$) and 0.3452 ($p<<0.00001$) at cycles 19 and 20. Sun spot energy variations on long time
scale (van Loon et al. 2007), even at very weak changes, could produce decadal and millennial
timescale impacts on global circulation thermohaline that affect in turn heat distribution (Bond et al.
2001, Gray et al., 2013).

**AMO.** The Atlantic Multidecadal Oscillation index is in opposite phase to the PDO (Enfield et al.,
2001; Condron et al., 2005); i.e., warm: 1930-1964 and 2000-present (cold PDO), and cold: 1965-
1999 (warm PDO). The correlations were generally higher at the descending phase of the SS cycles
(Fig. 11), practically the opposite to those for SS vs PDO, ONI, MEI, SST Anomaly. However, the
highest $r^2$ occurred on ascending (A) and descending (D) phases of SS cycles, thus: 0.700 ($p<< 10^{-10}$),



0.558 (p$\ll 10^{-10}$), 0.468 (p$\ll 10^{-10}$), 0.434 (p=0.03), 0.411 (p$\ll$0.00001) and 0.191 (p=0.001) for
cycles 20A, 22D, 19D, 24D, 21A and 23A, one-to-one. These $r^2$ showed strong degree of correlation,
although lower comparing to PDO. The lag time was basically between 24-48 months.



**Discussion and Conclusions**

**Period 1954-2017.** The SS peaks of the studied cycles decreased smoothly (Fig. 10); the SS peak

counts were around: 225 (SS# 21), 210 (SS# 22), 180 (SS# 23), and 110 (SS# 24), whilst at the

minimum for the cycles, SS counts were around 20-25. Thus, it could be said that Earth is receiving

less solar energy over these almost 7 decades. The reduction of SS peaks has been associated with

the beginning of Maunder Minimum (Mörner, 2015). Ineson et al. (2014) are projecting lower peaks

for next SS cycle (SS# 25); in fact, presently the SS counts per month are as low as 1.6 (July 2018) and

average 8.5 (Jan-Aug 2018) (http://www.sws.bom.gov.au/Solar/1/6), with expected counts to

decrease to 5.3 in February 2019.

Monthly SS counts correlations with SST, SST Anomaly (both in 3.4 and 1+2), ONI, MEI, AMO and

PDO through the whole time series (1954-2017) were poor; these had a correlation $r^2$ averaging

0.020 and a negative linear regression slope. It was observed that on the long run, there is not

strong correlation between SS and PDO, MEI, ONI and SST Anomaly in 3.4, where the $r^2$ were: 0.043,

0.029, 0.040 and 0.021 respectively. In the case of region 1+2, the correlation was even poorer:

<0.005.

The series of correlations at different lag times (6, 12, 24, 36 and 48 months) gave a response time of

12-24 months for all indexes, except for AMO (48 months), which align to what was previously

reported by Kristoufek, (2017), and Huo and Xiao (2016); i.e., in general the highest correlation

coefficients were found at that lag time (Table 1).

Changes of the SS could bring climate impact; there is a lengthy discussion about the effect on

climate change due to SS cycles. Gil-Alana et al. (2014) have found no significant statistical relation



between sun spots and global temperature; however, van Loon et al. (2007) suggested that even
though SS cycles produce weak changes on Solar Irradiation (SI) of about 0.07% according to Gray et
al. (2010); still, these can produce decadal and millennial impact on global thermohaline circulation
(Bond et al. 2001, Gray et al., 2016), due chiefly to UV energy fluctuation (Ineson et al., 2014). The
changes in UV (<100 nm to 350 nm) and near infrared (>800 nm to >1000 nm) are larger than in the
visible radiation (>350 nm-800 nm) and could have an important impact on global climate (Ermolli et
al. 2013); therefore, it is reasonable to deem some impact on the studied oceanographic indexes.
Recent data (Solar Radiation and Climate Experiment Satellite) suggest that the variability of UV
radiation during the declining phase of cycle 23 was larger than previous estimates (Harder et al.,
2009 and Haigh et al., 2010). Despite these small SI variations between the peaks and valleys of the
SS cycles, as 1) the total SI integrates over all the wavelengths, and 2) considering the huge heat
capacity of the seawater; these fluctuations give a strong possibility of increasing heat even at this
low SI variation. Also, there has been found that UV radiation penetrates down to 75-100 m depth of
the water column (Smyth, 2011), even though its variation is around 8% of total SI during the highs-
lows of SS cycles.

**Individual SS cycles (19-24)**. The SSTs correlations against individual SS cycles in regions 3.4 and 1+2
(Fig. 2) analysis rendered some insights; thus, in cycles 19, 21 and 23 were found the higher
correlations but variable in the region 3.4, up to 0.219 (SS# 23). The lag response time was 12-36
months in all cycles except 19, in line with Kristoufek, (2017) and Huo and Xiao (2016) reports. In
region 1+2, linear correlation $r^2$ was <0.0675 (SS# 19), and inconstant between cycles. Up to sixth-
degree polynomial correlation of SS against SST, SST Anomaly and other indexes were attempted
with equally poor correlation.

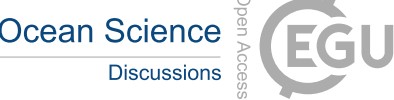

SS was correlated with anomalies SST (ERRSv4) in 3.4 and 1+2  (Fig. 3) in every cycle and did not have
a better $r^2$ than 0.396, for cycle 19 (in 1+2). Correlation factor $r^2$ showed  high variability as well the
response  time response (12-36 months). Correlation coefficient $r^2$ went up to 0.289 (in 3.4) and
0.396 (in 1+2) occurring in cycles 19 and 23 respectively, within the cold phase of PDO: 1955-1978
and 2000-present. In the period 1990-2016 in which occurred the two strongest El Niño (1997-1998
and 2015) and La Niña (2000-2002 and 2010-2012) in 1+2 region, still the SST Anomaly vs SS
correlation ($r^2$ of 0.127) was poorer than in 3.4 The slopes of the linear correlation were basically
positive for 3.4 and negative for 1+2. In general, in 1+2 there was found a more inconstant and lower
correlation than in 3.4. This is thought due to basically high seasonal and interannual variability of
oceanographic coastal conditions (Ormaza-González and Cedeño, 2017) that obscures the
correlation with SS. In 3.4, correlation is better, although it is still affected by fluctuation of regional
oceanographic and meteorological conditions expressed through some indexes; e.g., the Southern
Oscillation Index (Rasmussen and Carpenter, 1982; Barnston, 2015).

During the cycle 24, the ONI index was highly correlated to SS (Fig. 5), registering a $r^2$  up to 0.2510
p<<1.8E$^{-07}$) with a positive slope; which gives degree of correlation between high and moderate. It is
noteworthy to mention that in this cycle ONI reached 2.6C (Nov-Dec-Jan 2015/2016) and -1.7C (Oct-
Nov-Dec 2010). In cycles 20-21 the $r^2$ was low ($r^2$<0.04, p=0.02); however, from 22 to 24,  $r^2$
increased from 0.131 (p<<0.00006) to 0.251. The variability of the correlation can be ascribed also
to:
1) fluctuation of SS number from one month to another;
2) regional meteorological conditions (particularly cloudiness), ocean surface currents that transport
heat as Kelvin waves (Gill, 1982), SOI (http://www.cpc.ncep.noaa.gov/data/indices/soi), etc.  that in
turn affects the SSTs; and,



3) the way ONI is obtained: is the average of three successive months, but it has displacing reference
period of 30 years; thus, for the period 1950-1955, the reference period is 1936-1965; for 1956-
1960; 1941-1970 and so on. The EERSv4 uses the period 1981-2010, in which 3 very strong El Niño
and many others has occurred; thus, the reference moving period of 30 years  will slightly affect the
ONI index somehow (Lindsey, 2013).

The PDO is an index (Fig. 5) for interdecadal oscillation alike AMO; it seemed to respond in 36-48
months to SS cycle showing a high degree of correlation coefficient r of 0.625 (SS#19) and 0.766 (SS#
24); for cycles 20-23: 0.467-0.508, linearly correlated with fair-good degree and positive slope
(except in SS# 20). A better fit occurred in the cycles 19 and 24, which have the highest and lowest
SS peak of the six cycles analyzed (Fig. 10), both are in the cold phase of the PDOs. The AMO index
(Fig. 6) gave a variable correlation, from a weak  $r^2$ of 0.130 (p=0.00001) to strong 0.490
(p<<0.00001) with a response time of 48 months for cycles 23 and 20 respectively. Gray et al. (2016)
reported 36-48 months lag for mean-sea-level pressure in the North Atlantic in a study of 32 SS
cycles. The slopes of the PDO and AMO linear regression curves are negative and positive
respectively in cycle 21 and 22, but in concordance in 19, 20, 23 and 24 during cold phase PDO.
These two interdecadal oscillations proved to be correlated to SS; however, the PDO showed higher
correlation. Perhaps, since the North Pacific Ocean basin has a larger area than the North Atlantic;
there is higher heat storage capacity in the first.

The analysis through the ascending and descending phases of each cycle rendered clearer results.
The SSTs in 1+2 showed higher correlation with r up to 0.2052 in the ascending phases; in the
descending phase r was below 0.067 0.259; however, the response time and slope of the linear
correlation curves did not show a specific pattern. In region 3.4, there was a high degree of
correlation: 0.8699  (SS# 24), 0.6089 (SS# 22), 0.5736 (SS# 19) and0.5559 (SS# 23), at ascending





phase of the cycles (except SS# 20 and 24). The response time was 36 months. Slopes were negative
and positive during warm ( SS# 20-22) and cold (SS# 19, 23, 24) PDO phases respectively. The highest
$r^2$: 0. 870 in the descending phase in the cycle 24 coincided with the strongest El Niño (2015) and the
second highest (SS# 22) with two consecutive strong El Niño 1991-1995, the third with el Niño 1955-
1957, and finally the fourth during 1997-1998 warm event.  It seems that short time expressions of
SS cycles, either at the beginning of their ascending or descending phases, have a trigger effect on
the SSTs. This was observed through the polynomial regression curves (Fig. 12) that were found for
each SS cycle. The polynomial curves of order 6 were fitted with an average $r^2>0.89$ (p≤0.001).
However, a response time of 24-36 months seems to occur at the low or high plateau of the cycles
(Fig. 12), then the event is the strongest. Thus, El Niño1957-1958 (SS# 19), 1965-1966 (SS# 20), 1981-
1982 (SS# 21), 1987-1988 and 1991-1992 (SS# 22), 1997-1998 (SS# 23), 2015-2016 (SS# 24). On the
other hand, the cold events La Niña tends to occur after an El Niño at the middle of the ascending
phases (1988-1989, 1999-2001, 2010-2012) or when approaching the minimum of the cycles (1973-
1974, 1975-1975; 1995-1996, 1917-1918). The so called equatorial Pacific neutral conditions in 3.4
(see, https://iridl.ldeo.columbia.edu/maproom/ENSO/ENSO_Info.html), seems to span a longer
period after La Niña, vice versa after El Niño.

The ENSO indexes ONI and MEI also showed strong correlation associated to the ascending phase of
the SS cycles, with a lag time of 24-36 months. In four cycles, correlation coefficient r varied from
0.3913 (fair) to 0.9038 (strong) in the ascending phase with a positive slope of the linear regression
curve. The three highest r match dates of strong El Niño 87-89, 55-57 and 97-98 with positive slope
and ascending phase. During the descending phase, the correlation coefficients were lower and
ranged from 0.6082 to 0.2938, all of them with lag time 0-12 months and positive slope. In this
exercise, it was also found that warm events tend to occur in the ascending phase or at the top of
the cycle and have a delay time of 36 months, which was reported also by Huo and Xiao (2016),





whilst cold events are associated to a descending phase but with a quicker response time:  0-12
months, except La Niña 1988-1989 and 2010-2012.

Similarly, for MEI, $r^2$ ranged from : 0.784 (SS# 19) to 0.222 (SS# 21), in the ascending phase and lag
time 12-48 months (except SS# 23 and 24). During the descending phase, the correlation $r^2$
coefficients were lower and varied from 0.321 (SS# 24) to 0.143 (SS# 23) with positive slope and
quicker lag time 0-12 months. The index MEI resembles similar pattern as ONI; lower correlation
may arise as MEI take into consideration six variables that all together may mask the signal from sun
activity.

On a longer time scale, the interdecadal oscillation of the Pacific (PDO) and Atlantic (AMO) were
strongly correlated to SS. The PDO (Fig 10) was linearly correlated with  $r^2$ coefficient, ranging from
0.7677 to 0.2854  in the ascending phase with lag time 24-36 months, as reported by Huo and Xiao
(2016), with positive slopes in cold phase PDO (cycles 19, 23 and 24) and the contrary in warm PDO
(cycles 21 and 22); whilst cycle 20 was a sort of transitioning period between the PDO phases. The $r^2$
of AMO (Fig. 11) with individual SS cycles varied from 0.160 to 0.700. Similarly, the response time
was 24-48 months. These results correspond with van Loon et al. (2007), who stablished that even a
low change in the sun activity (SI) could produce decadal and millennial time scales affecting
thermohaline circulation (Bond et al. 2001, Gray et al., 2016), which in turn is reflected by PDO and
by the AMO index, which is somehow in opposite phase to PDO (Enfield et al., 2001; Condron et al.,

2005).

Recently, after the second quarter of 2018, many models and researchers are projecting El Niño to
occur sometimes in late northern hemisphere summer (see:
http://www.bom.gov.au/climate/enso/); but it did not occur. Then, the projections passed to the



beginning of autumn (http://www.cpc.ncep.noaa.gov/products/precip/CWlink/MJO/enso.shtml);
similarly, it did not happen. Now, on the third week of September, it is pronounced to occur in late
autumn, but with fewer models asserting the event. Most models seem to be failing to provide a
consistent projection. Probably, this is because, there has been two re-cooling processes in all El
Niño areas, that keep the ONI index within the realm of ENSO neutrality (-0.5C to 0.5C).  The PDO
index have been averaging -0.53, and it is in its cold phase
(https://www.ncdc.noaa.gov/teleconnections/pdo/). Sun spots have been at very low numbers, with
some weeks without any. During 2017 the average smoothed SS counts per month was 21.8, and for
2018 is 8.5, with just 1.6 in July (http://www.sws.bom.gov.au/Solar/1/6) with expected counts to
decrease to 5.3 in February 2019. Probably, there will not be an event El Niño during 2018 or at least
little chance according to present results.

**Data availability**. All data are publicly available on the Web (see Material and Methods).

**Author contributions.** Franklin Isaac Ormaza-González led and oversaw the whole project. He
conceptualized the hypothesis, researched the literature, designed the material and methods, and
wrote the paper in all its stages. María Esther Espinoza-Celi looked for and retrieved all data and
information, run statistical and spectral analysis, organized results, and designed graphs. She
designed the poster presentation.

**Acknowledgements.**  Authors are grateful to ESPOL authorities whose supported research allotting
time and financial resources to present paper in the "4TH INTERNATIONAL SYMPOSIUM: THE
CLIMATE CHANGE EFFECTS ON THE WORLD OCEANS" held in Washington DC, 4-8 June 2018, also



The National Chamber of Fisheries of Ecuador support is acknowledged. Work on the English by
Dafne Vera-Mosquera is indeed appreciated.



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

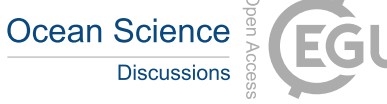




**Table 1.** Linear correlations $r^2$ and p-values between SS monthly counts and PDO, MEI, ONI, AMO, SST and SST Anomaly in 1+2, and 3.4. The time series April 1954 to December 2017. Values in yellow background mean negative slopes of the linear regression curves. Pink shading cells are p>0.05. Red: $r^2$ values below 0.001.

| Variable | | T+0 | (t+6) | (t+12) | (t+24) | (t+36) | (t+48) |
|---|---|---|---|---|---|---|---|
| PDO | $r^2$ | 0.00 | 0.02 | 0.04 | 0.04 | 0.03 | 0.02 |
| | p-value | 3.83E-01 | 1.75E-04 | 1.94E-07 | 1.26E-08 | 2.07E-06 | 3.11E-05 |
| MEI | $r^2$ | 0.01 | 0.02 | 0.03 | 0.02 | 0.01 | 0.00 |
| | p-value | 2.58E-02 | 9.68E-05 | 2.55E-06 | 1.63E-05 | 1.51E-02 | 9.30E-01 |
| ONI | $r^2$ | 0.01 | 0.03 | 0.04 | 0.03 | 0.01 | 0.00 |
| | p-value | 2.22E-03 | 4.93E-06 | 2.11E-07 | 1.91E-06 | 4.00E-03 | 4.13E-01 |
| AMO | $r^2$ | 8.66E-05 | 8.30E-04 | 9.66E-04 | 2.89E-03 | 1.39E-02 | 1.89E-02 |
| | p-value | 7.97E-01 | 4.28E-01 | 3.94E-01 | 1.44E-01 | 1.45E-03 | 2.21E-04 |
| SST 1+2 | $r^2$ | 2.54E-04 | 1.61E-03 | 3.08E-05 | 2.92E-06 | 3.04E-04 | 1.26E-03 |
| | p-value | 6.61E-01 | 2.71E-01 | 8.80E-01 | 9.63E-01 | 6.37E-01 | 3.44E-01 |
| SSTA 1+2 | $r^2$ | 0.00 | 0.00 | 0.00 | 0.00 | 0.00 | 0.00 |
| | p-value | 7.14E-01 | 1.67E-01 | 3.37E-01 | 2.57E-01 | 8.13E-01 | 3.73E-01 |
| SST 3.4 | $r^2$ | 0.00 | 0.01 | 0.01 | 0.01 | 0.00 | 0.00 |
| | p-value | 2.78E-01 | 5.93E-03 | 1.07E-03 | 1.32E-03 | 6.48E-02 | 9.10E-01 |
| SSTA 3.4 | $r^2$ | 0.00 | 0.01 | 0.02 | 0.02 | 0.01 | 0.00 |
| | p-value | 1.45E-01 | 2.25E-03 | 8.23E-05 | 1.23E-04 | 2.45E-02 | 9.79E-01 |



**Fig. 1.** Behaviour of monthly counts SS, ONI, MEI, PDO and AMO. The indexes start at t= 0, 12, 24 and 36 months (panels a, b, c and d respectively). The SS series starts at t=0 in the four panels. The vertical axe gives the values for the indexes and SS numbers (multiply by 100).

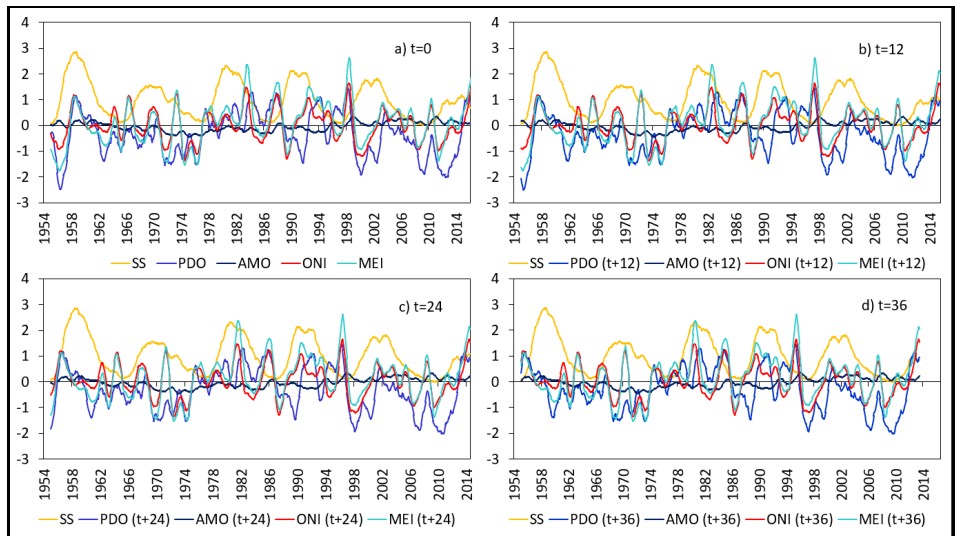






**Fig. 2.** Linear regression correlation coefficient r$^2$  (p<0.05) of SS monthly counts for cycles 19-24
against SST in regions El Niño 1+2 (blue) and 3.4 (red).

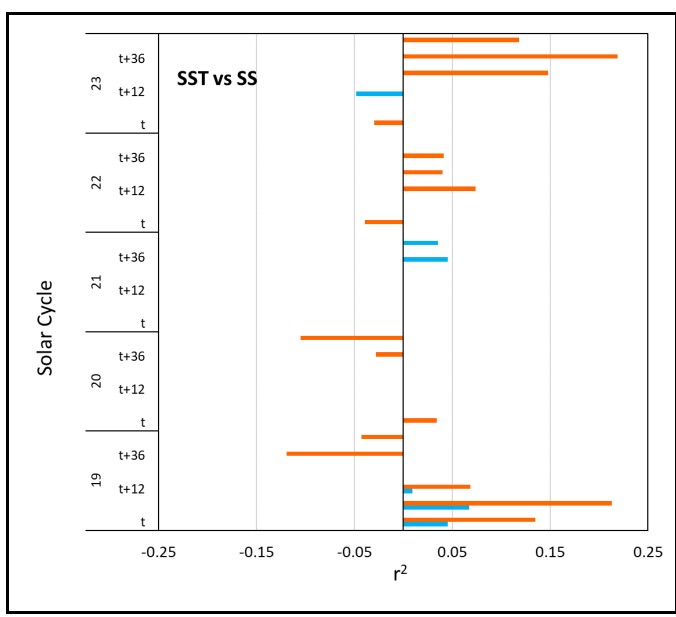




**Fig. 3.** Linear regression correlation coefficient r$^2$ (p<0.05) of SS monthly counts of cycles 19-24 against SST Anomaly in regions El Niño 1+2 (blue) and 3.4 (red).

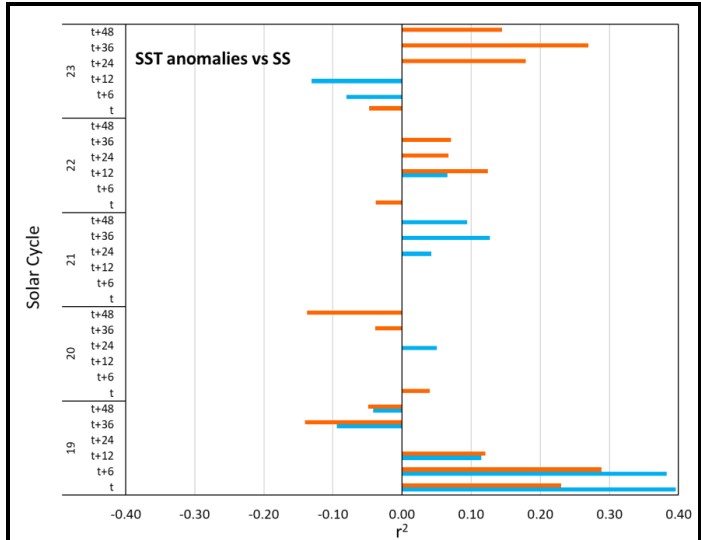

**Fig. 4.** Linear regression correlation coefficient r$^2$ (p<0.05) of SS monthly counts against SST Anomaly in regions El Niño 1+2 (blue) and 3.4 (red) and indexes PDO (grey) and ONI (green) through three time periods.

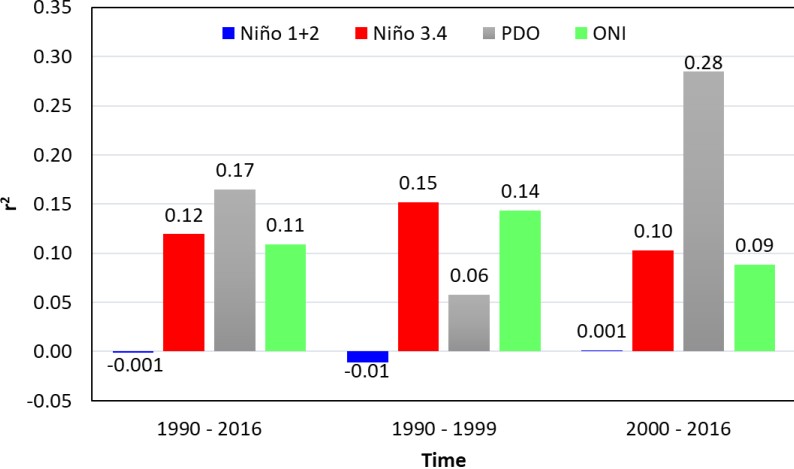





**Fig. 5.** Linear regression correlation coefficient $r^2$ (p<0.05) of SS monthly counts cycles 19-24 against
indexes: ONI (red) and PDO (blue). Negative slope (- $r^2$).

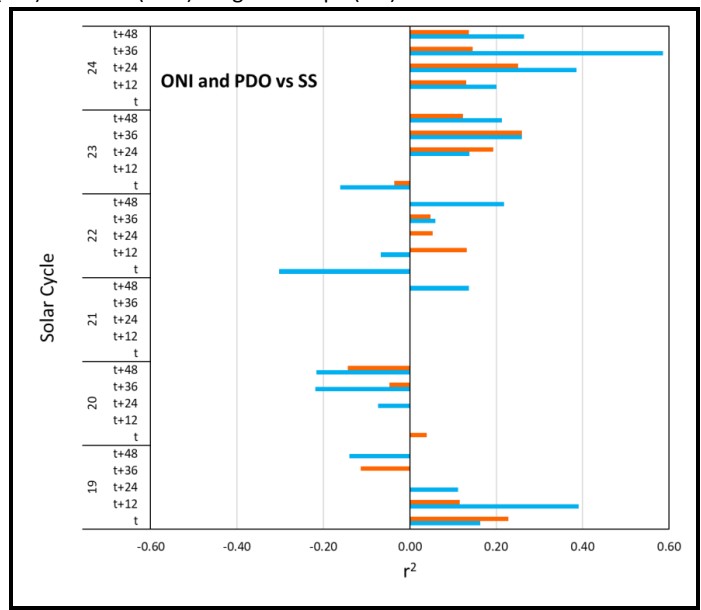

**Fig. 6.** Linear regression correlation coefficient $r^2$ (p<0.05) of SS monthly counts for cycles 19-24
against index AMO. Negative slope (- $r^2$).

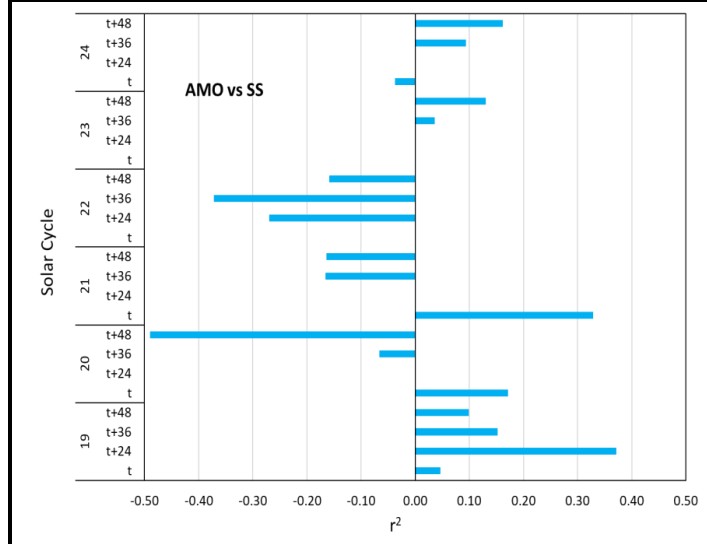




**Fig. 7.** Linear regression correlation coefficient $r^2$ (p<0.05) of SS monthly counts during the ascending
(blue) and descending (red) phases of SS for cycles 19-24 against SST in regions El Niño 1+2 (left) and
3.4 (right).

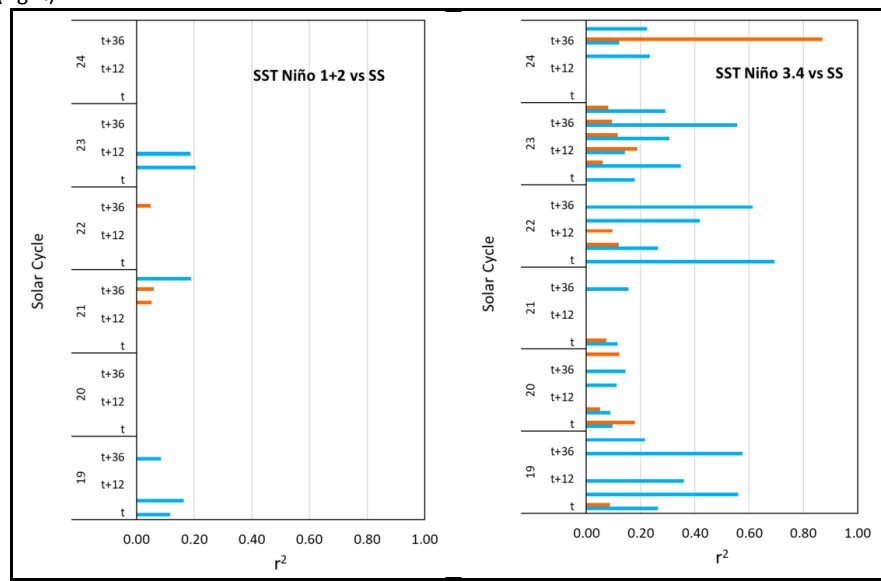

**Fig. 8.** Linear regression correlation coefficient $r^2$ (p<0.05) of SS monthly counts during the ascending
(blue) and declining (red) phases for SS cycles 19-24 against SST Anomaly in regions El Niño 1+2 (left)
and 3.4 (right).

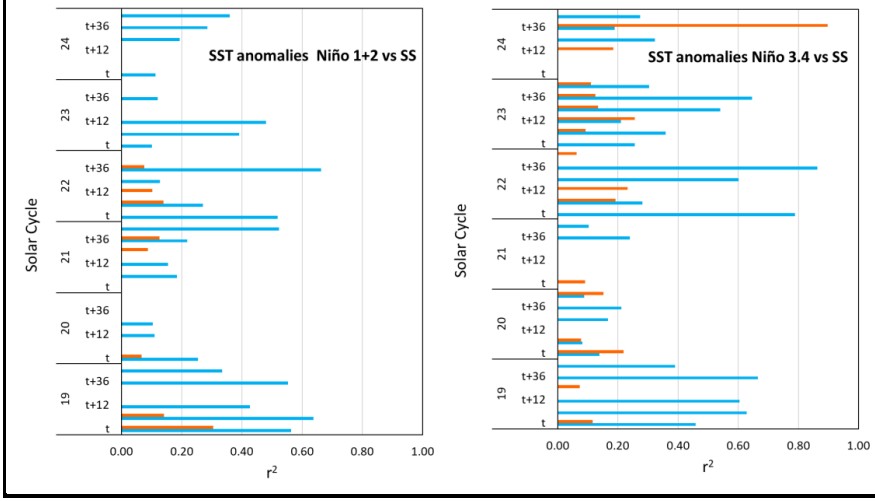






**Fig. 9.** Linear regression correlation coefficient $r^2$ of SS monthly counts during the ascending (blue)
and declining (red)  phases of SS cycles (19-24) against index ONI. Negative slope (- $r^2$).

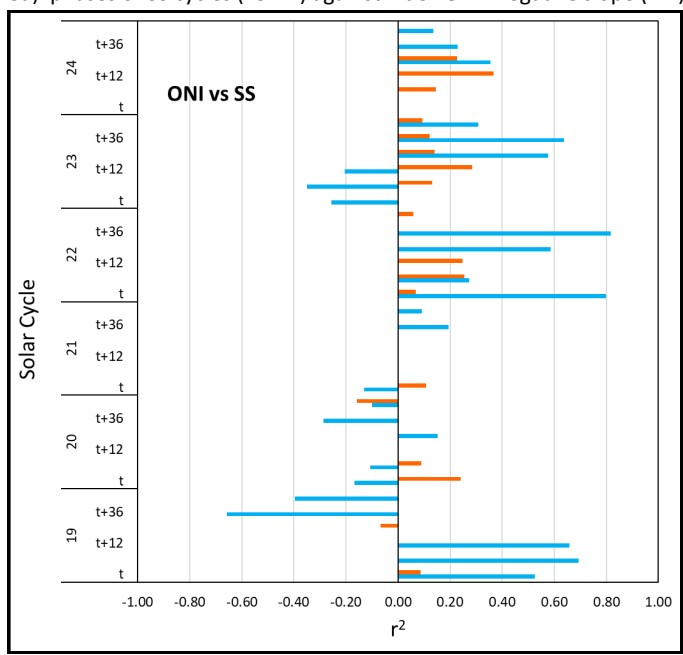



**Fig. 10.** Linear regression correlation coefficient $r^2$ (p<<0.05) of SS monthly counts during the
ascending (blue) and declining (red) phases of SS cycles (19-24) against index PDO. Negative slope (-
$r^2$).

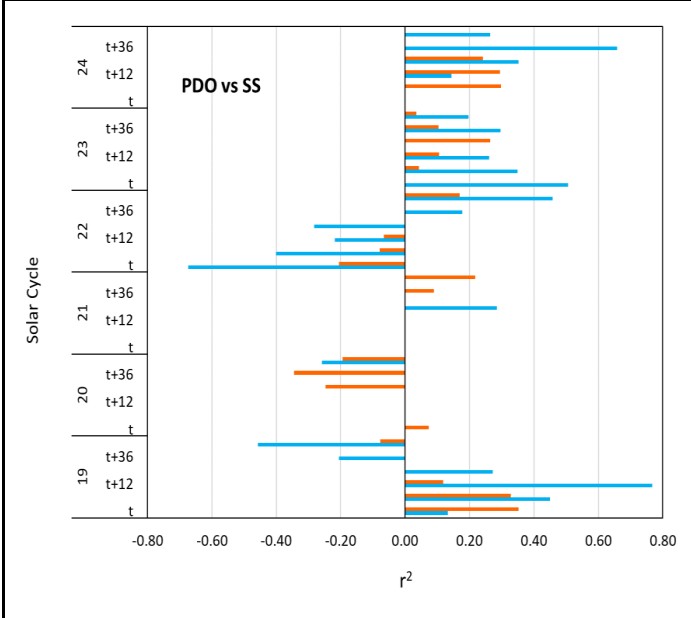


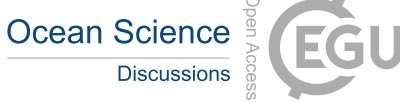



**Fig. 11.** Linear regression correlation coefficient r² of SS numbers during the ascending (blue) and
declining (red) phases of SS cycles (19-24) against index AMO. Negative slope (-r²)

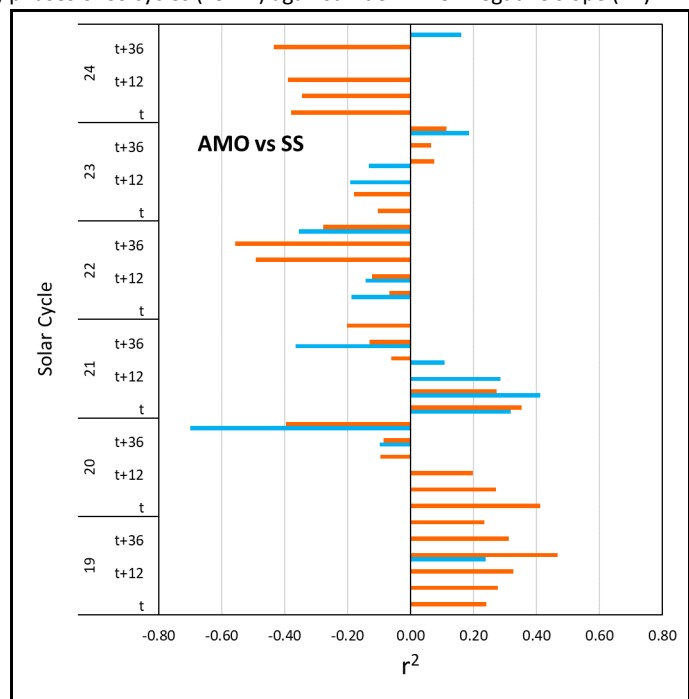



**Fig. 12.** Polynomial functions of 6 degrees (p<0.001), based on monthly SS counts. Red and blue lines
represent El Niño and La Niña event. The cycle number of the top of each panel.

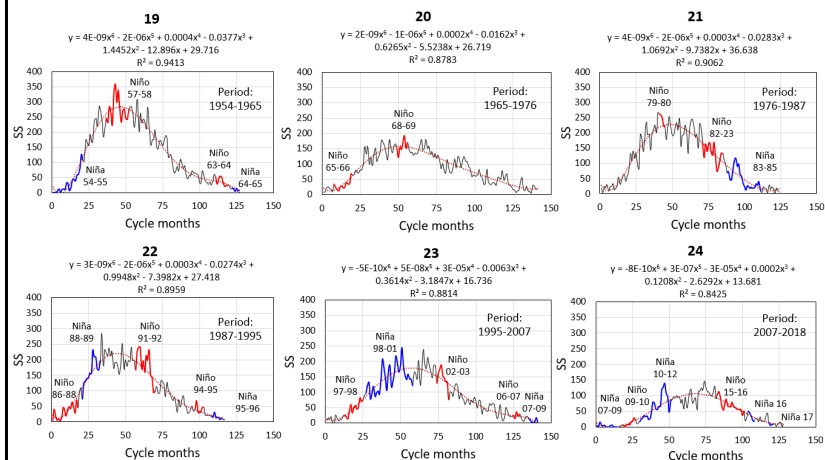
