# Peer review of "Do sun spots influence the onset of ENSO and PDO events in the Pacific Ocean?"

_Ocean Science, 2018_

## Referee Comment (RC1) · Anonymous Referee #1 · 17 Jan 2019

This paper is trying to answer the question of "Do sun spots influence the onset of ENSO and PDO events in the Pacific Ocean?". However, I could not find a robust answer based on their analysis and results. It is really hard for me to read through the whole paper and get the key points. I am trying to be more encouraging, however, I have to suggest to the editor to reject this submission due to reasons as follows:

1. No robust results. The full paper is telling correlation numbers from the beginning to the end, without any efforts to understand or explain the related physical processes and underlying mechanism. 2. Not valid method. For the solar impacts on climate, it's true that correlation has been used widely in previous studies. There is always a problem

while using correlation since a relative high correlation number might just be caused by occasional in phase of two variables. So, a long data record is necessary while considering the relationship between the sun spots numbers and other climate indices. This study did the correlations between the sun spots numbers with other variables for each individual cycle ( $\sim$ 11 years) and even for the ascending or descending phases (only 4-5 years), which is not meaningful to my understanding. During such a short period, a high correlation number just means that the sun spots number is in phase with other climate indices occasionally. 3. Overstated conclusion. There might be some connection between the sun spots number and other climate indices such as PDO and decadal ENSO. However, I do not believe we can do a prediction for El Niño or La Niña events only based on sun spots numbers as stated by the authors in the abstract and conclusion. 4. Not well organized and written. As I mentioned above, it is really hard for me to read through the paper and get the idea of what the authors want to improve our understanding to the scientific issue. For example, the whole abstract is just a lot of correlation numbers. It is really hard for the audience to follow and to figure out what do the numbers mean and what are the differences between them. In addition, I would also suggest the authors to ask help from a native speaker to improve their English written for the whole manuscript.

---

## Referee Comment (RC2) · Anonymous Referee #2 · 7 Mar 2019

In this paper the authors are exploring the relationship between sunspots and their impact on ENSO and PDO in the Pacific Ocean. It was difficult for me to find an answer to the title's question, "Do sun spots influence the onset of ENSO and PDO events in the Pacific Ocean?" It is difficult for me to tease out the main points of the paper because I find the structure unclear and the figures don't effectively summarize the main points either. Annotation of the figures could help address this. There was clearly a lot of work done but I do not feel like it is presented in a way that effectively supports their arguments and that correlations are insufficient for their claims, which remain a bit unclear. I recommend that this article be rejected as its structure is unclear and hard to read, main questions are not outlined clearly, and the presented statistics

don't support the conclusions being made.

The paper reads like a list of r^2 values and Pearson correlation coefficients and is lacking a coherent narrative. Not every correlation needs to be typed out, they can be in a table or figure and referred to. In these long lists it is hard to identify the most important ones. Their argument hinges on these statistics but I do not believe that they can sufficiently support their claim that ENSO and PDO are driven by sun spots. This is because correlation does not imply causation, slightly misleading from their title, and correlations can be artificially high due to a brief periods of in-phase activity. These caveats should be mentioned and can be remedied with longer time series. I do not think that the methods used can answer the titular question. There needs to be more interpretation and context with the r^2 values, rather than listing them. More error and uncertainty discussion would also improve the paper. Therefore, as is I do not think these statistics can be used as a predictor for ENSO.

This paper would require structural overhaul with clearly defined sections and goals. The figures should be annotated and streamlined to be more easily interpreted and more clearly support the main arguments.Additionally, the paper requires editing by a native English speaker, much of the science gets lost in the presentation.

---

## Author Comment (AC1) · 27 Mar 2019

**Franklin Isaac Ormaza-González and María Esther Espinoza-Celi**

formaza@espol.edu.ec

Dear reviewer, we are grateful for your time put on our manuscript and appreciate the comments made on it. We would like to make some notes with most respect and without intending any conflictive discussion. We think the paper contributes a better understanding of the ENSO (La Niña and El Niño) and also PDO and AMO longer scales fluctuations. We are sorry we have taken some time to reply you, but, as per your suggestion, we have revised the entire paper, we have reviewed and have re-rewriting many parts with the help of two well-known British scientists. We also were waiting for the second reviewer appraisal; we have now that. We will try to reply every

point you did, in the hope you re-consider your first revision.

1.- No robust results. The full paper is telling correlation numbers from the beginning to the end, without any efforts to understand or explain the related physical processes and underlying mechanism. We took as dependent variables, four indexes plus SST and anomaly SST from different oceans areas placed in the equatorial Pacific Ocean (from 170W to 82W) in which are regions areas 3.4 (5°North-5°S, 170-120°W) and 1+2 (0-10°S, 90°W-80°W). The first is the reference area for ONI and MEI indexes as well as SST and anomaly SST and it is an open area; whilst the second area is where much of what happens in 3.4 is reflected, but this one is close to coast. These four variables are inter-related and used to determine ENSO processes (El Niño, La Niña and neutral episodes), which are interannual, lasting 12-18 months. The AMO and PDO refer to SST behaviour of the North Pacific and Atlantic which are larger oceanic areas than 3.4 and 1+2. The relationship between and AMO and PDO is widely accepted (as shown in the paper) and ENSO is associated somehow to PDO; thus, during cold phase PDO, La Niña events are more frequent and intense than El Niño. When PDO is on warm phase the contrary. The most intense and damaging El Niño (fully developed) occurred between 1980-2000, which is a period of warm phase PDO. Two most intense and prolonged La Niña happened during colds phase PDO (1954-1979) and 2001-present. The independent variable was the monthly SS from 1954 to 2017 represent 6 cycles. The SS is an accepted way to estimate Sun activity and "The Sun's activity cycle governs the radiation" (Bhowmik and Nandy, 2018), which in turn affects the heat content of the ocean surface and therefore the indexes above mentioned; in the section Introduction we explain thoroughly. We add, Higginson et al. (2004, paragraph 39) said "Our analyses of recent SOI fluctuations, El Niño frequency and intensity, suggest a coupling between the âĹij11‐year solar luminosity cycle and the SOI. Specifically, if we filter the SOI for El Niño (shaded gray) and La Niña (solid black bars) excursions, the more gradual quasi‐cyclic trend of the SOI is inversely correlated with Sunspot Index (SSI) with approximately 24 months lag". See Fig 1. The SOI index is part of the ENSO, but we did not consider it, because it is highly volatile.

So, we have a clearly defined dependent and independent variables. We tried to evaluate the possible relationship taking the whole period of six cycles together, individual cycles and their ascending and descending phases, plus 1990-2017 and 2000-2017, per each variable and lag times of 6, 12, 24, 36 and 48 months Understandably there is an enormous amount of correlations, we did take the most important and perhaps risking being too reiterative and tedious we decided to report them for all variables and combinations, but our intention is to show how most of them are in concordance and all signalling how the dependent variables correlate with the independent one. Also, the intention was to show the reader them, so they can also judge and not only us. Nonetheless what we think, we have significantly reduced the number of correlations shown (please see new rewritten manuscript). Every effort was made to explain the poor and high correlations in terms of physical and oceanographic processes. It can be seen in lines: 245-254, 231-237, 260-263, 268-273, 277-285, 300-310, 318-326, 333-336, 342-347, 353-374, 384-388, 399-403, 410-413 and so on.

In the section Results and discussion every correlation is explained and understood. This section has been rewritten to make clearer what you mention. In the same form, the section conclusions have been modified accordingly. Thus, the results are robust (consistent correlations with p<0.01) and explained for the purpose, which basically "attempts to understand how the sunspots may affect low and high frequency oceanic events such as the Pacific Interdecadal Oscillation (PDO), the Atlantic multidecadal oscillation (AMO), anomalous sea surface temperatures, and El Niño and La Niña events".

2.- Not valid methods. For the solar impacts on climate, it's true that correlation has been used widely in previous studies. There is always a problem while using correlation since a relative high correlation number might just be caused by occasional in phase of two variables. So, a long data record is necessary while considering the relationship between the sun spots numbers and other climate indices. Again respectfully. If the method is not valid for this paper; why should it be OK for previous accepted and reported studies. Every scientific method has shortcoming and limits. Recently, a paper correlating SS and rain over Europe has been published, and is reporting correlations factors similar with lag times in the range reported by us: ".... Taking into account cause and effect, it is suspected that increases in Central European rainfall are actually triggered by the solar minimum some 3–4 years before the rainfall month, rather than the lagging solar maximum...." (see, Laurenz et al. 2019). Similarly, Higginson et al. (2004) reported SS association to SOI index; this index is part of ENSO, with similar length of data (see Fig. 1). Thus, these paper are talking about cause-effect. Correlation analysis can be accepted to find any association between an independent (SS) and dependent (SST, Anomaly SST, ONI, MEI, PDO, AMO) variables. All variables are inter-annual and decadal (not climatological). These correlations are logical as the dependent variables are affected by the sun radiation, which in turn can be estimated by sun spot activity (see formula 1, line 55). Simply, the physical process is energy from the most important source (sun) being transferred to sea surface water and loosing energy through other processes (evaporation, upwelling, friction, for example). In the present case we are not even attempting to say that dependent variables are only affected by SS. Not at all, but SS play its role, and this role sometimes is poor (low r2) and sometimes is important (high r2). These indexes are also affected by others oceanographic and even anthropogenic variables (see Laurentz et al., 2019). Consistent higher correlations were found in area 3.4 than 1+2; why? In 1+2 there are more intense dynamic processes affected by winds, interaction of different ocean water mass (north and south equatorial currents), transport of panama Bay heat content mediated by trade winds from the Atlantic, higher and variable cloudiness due to the geographical variability of the ITCZ (intertropical convergence zone), Humboldt and Cromwell upwelling, etc. whilst in 3.4 do not exist upwelling, cloudiness is less affected by land-atmospheric processes, not trade winds from the Atlantic. The outcome of the higher correlation values in 3.4 is logical.

2. ... This study did the correlations between the sun spots numbers with other variables for each individual cycle (âĹij11 years) and even for the ascending or descending

phases (only 4-5 years), which is not meaningful to my understanding. During such a short period, a high correlation number just means that the sun spots number is in phase with other climate indices occasionally The paper analysed the whole length of 6 cycles (1954-2017) of SS (monthly counts) and a period from 1990 to 2017. The correlation r2 of the 6 cycles showed consistency (even though at lower r2) with those analysed during shorter periods (1990-2017), individual cycles (around 130 months) and ascending/descending phases (around 5 years, 60-70 months). See lines 161-163, number of samples. El Niño event is aperiodic but tends to occur every 2-5 years (24-60 months) and it tends to last around 12-15 months. On the other hand, the lag time is on average period of 24-36 months. Thus, there would be at least one El Niño or La Niña event in every ascending or descending phase. At least 60 points (SS, index) were linearly correlated. On the other hand, it has been reported that a required size of the sample is 25 or more (David, 1938). In here, the freedom degrees are around 58 or more. When one dependent variable correlates well with its independent variable the r2 get closer to 1, that would imply is the only cause; but in this case the sun spots (solar activity) are not only cause of the index variation, and reasons are given for that. Respectfully, 6 ascending and 6 descending phases were analysed, and at all times the indexes were in "phase" at lower or higher value (r2) in a consistent manner. To us, to be in "phase" would mean the SS (sun energy) is the only force affecting the index directly and so the slope of the linear regression will be similar (or almost) in every ascending phase. It is hard to say the pair of variables were in "phase" when the slopes were different in every ascending/descending phase (we are going to add one or two graphs to the paper to show this). The p-values were always «0.01, so that means chances to be phase occasionally are very low. Finally, imagine, we have an area of ocean surface with the same initial SST, without any cloudiness, winds, currents, etc. being irradiated by a single slightly variable source of energy. Can we say the heat content would be directly influence by the source? The answer is yes. If the source have higher energy (ascending phase) the heat content will increase and vice versa. A change of 1.5 Wm-2 (1289 cal/hour) between peak

and valleys of the SS probably is important, although the IPPC (2001) has consider this value not significant in forcing climate. Climate is understood a time spam over 30 years (https://www.nasa.gov/mission_pages/noaa-n/climate/climate_weather.html).

3. Overstated conclusion. There might be some connection between the sun spots number and other climate indices such as PDO and decadal ENSO. However, I do not believe we can do a prediction for El Niño or La Niña events only based on sun spots numbers as stated by the authors in the abstract and conclusion. Re-reading, the way the abstract was written, you are quite right. It could lead the reader to arrive that conclusion is overstated. We are trying to put in the front line the fact that solar activity variation of energy in magnitude and quality (radiation spectre changes in the ascending/descending phase and cycles) affects the indexes. We do not pretend to forecast El Niño La Niña events just by using SS, not at all. However, it is heuristically reasonable to suggest the not occurrence of a full-fledged moderate-strong el Niño till 2020-2021. We said this when writing the paper by early 2018, and in fact by mid-2018 models were suggesting an El Niño in August-September, it did not happen, actually the region 3.4 and 1+2 cooled down by the end of 2018. The latest report from http://www.bom.gov.au/climate/enso/#tabs=Overview is saying "The El Niño–Southern Oscillation (ENSO) remains neutral. However, the Bureau's ENSO Outlook remains at El Niño WATCH, meaning there is approximately a 50% chance of El Niño developing during the southern hemisphere autumn or winter...". Nonetheless, NOAA view is that a weak El Niño conditions are present now and to expect to continue through the Northern Hemisphere spring 2019 (~55% chance), https://www.cpc.ncep.noaa.gov/products/analysis_monitoring/lanina/enso_evolution-status-fcsts-web.pdf. The fact is, El Niño is not fully-fledged; id est, the meteorological (SOI) and oceanographic are not connected. At this moment (march 24, 2019) the oceanographic and meteorological variables have not fully couple yet. During the 2108 and 2019 the SS have been very low in numbers (see text) and for many weeks they have disappeared from the sun, thus the sun energy is at the lowest radiation. Perhaps a fully fledged El Niño needs a "little" extra solar energy. The new ascending

phase would happen in 2019, having a lag of 24 months or so, 2020-2021 is not unreasonable period in which a fully-fledged El Niño could occur.

4. Not well organized and written.

We followed your suggestion, a well-known and prestigious Professor from UK helped with edition and review. Revised manuscript is being submitted. Finally, again we appreciate the time put to our paper. Your comments have made us to improve it.

References. Bhowmik and Nandy. 2018. Prediction of the strength and timing of sunspot cycle 25 reveal decadal-scale space environmental conditions. NATURE COMMUNICATIONS | (2018) 9:5209. https://doi.org/10.1038/s41467-018-07690-0 David, F.N. 1938. Tables of the ordinates and probability integral of the distribution of the correlation coefficient in small samples. Cambridge: Cambridge University Press Higginson, M. J., M. A. Altabet, L. Wincze, T. D. Herbert, and D. W. Murray (2004), A solar (irradiance) trigger formillennial-scale abrupt changes in the southwest monsoon?Paleoceanography,19, PA3015, doi:10.1029/2004PA001031 Intergovernmental Panel on Climate Change (IPCC) (2001), Third Assessment Report–Climate Change 2001, The Scientific Basis, Cambridge Univ. Press, New York. Laurenz, L., H.-J. Lüdecke, S. Lüning (2019): Influence of solar activity on European rainfall. J. Atmospheric and Solar-Terrestrial Physics, 185: 29-42, doi: 10.1016/j.jastp.2019.01.012

Fig. 1 from Higginson *et al.* (2004)

*Plot of long-term SOI over the past 50 years overlaid by SSI (total solar irradiance) best fit lagged by 24 months. Intervals when the SOI is significantly above the SSI curve are marked with solid black bars, and significant intervals when it drops below the SSI are indicated by cross-hatching. The largest negative excursions of the SOI from the SSI are shaded dark gray.*

[Figure]

**Fig. 1.** Fig. 1 from Higginson et al. (2004) Plot of long‐term SOI over the past 50 years overlaid by SSI (total solar irradiance) best fit lagged by 24 months. Intervals when the SOI is significantly above th

---

## Author Comment (AC2) · 27 Mar 2019

**Do sun spots influence the onset of ENSO and PDO events in the Pacific Ocean?**

*Franklin Isaac Ormaza-González[1], and María Esther Espinoza-Celi[1]*

1)  ESPOL Polytechnic University, Escuela Superior Politécnica del Litoral, ESPOL, (Facultad de
Ingeniería Marítima, Ciencias Biológicas, Oceánicas y Recursos Naturales), Campus Gustavo
Galindo Km. 30.5 Vía Perimetral, P.O. Box 09-01-5863, Guayaquil, Ecuador

Corresponding author: formaza@espol.edu.ec.

The sea surface temperature (SST), SST anomalies, ONI (Oceanographic El Niño Index) and MEI
(Multivariate ENSO Index) in regions El Niño 1+2 (80°W-90°W, 0°-10°S) and 3.4 (5°N-5°S, 170°W-
120°W) as well as the Pacific Decadal Oscillation (PDO) and Atlantic Multidecadal Oscillation (AMO)
indexes, were correlated to sun spots number (SS) from cycles (SS#) 19 to 24 (1954-2017).
Polynomial regression functions represented each of the six cycles with an average $r^2>0.89$
($p<0.001$). Series of correlations between SS and chosen indices at different lag times (0, 6, 12, 24,
36 and 48 months) gave a response time of between 12 and 36 months. Over the entire 1954-2017
period, the SS cycles did not show a strong correlation with the variables or SST Anomaly in the El
Niño areas 1+2 and 3.4. It seems that high and low SS balanced through the cycles. Improved
correlations were found for the shorter period 1990-2016. The SST correlations against individual SS
cycles in regions 3.4 and 1+2 were up to 0.219 (SS# 23) and <0.0675 (SS# 19) correspondingly. SST
Anomaly, ONI and MEI correlated with $r^2$ of 0.250, 0.3943 and 0.2510, one-to-one; the lag time was
24-48 months and linear curves had positive slope. In general, more inconsistent and lower
correlations were found in 1+2 than in 3.4. On longer time scale indexes, the PDO (as well as the
AMO) seemed to respond in 36-48 months to SS cycles ($r^2$ of 0.625, SS# 19) and 0.766, SS# 24);
whilst the AMO index gave a slightly lower correlation (0.490, SS# 20) with a similar lag time. Further
analysis of SS numbers and the oceanic indices above during the ascending and descending phases
of each cycle showed SST was best correlated with the ascending phase (up to $r^2$ of 0.870, with a lag
time between SS cycle and index of about 36 months) and this trend also applied to the SST
anomaly, although with slightly poorer correlations.  The highest $r^2$ values coincided with strong
ENSO events. The descending phase showed lower correlations between SS and ocean indices
including MEI and ONI. The PDO was linearly correlated to SS ($r^2$ 0.7677 to 0.2855 (12 to 24 months)
as was the AMO ($r^2$ up to 0.700) whilst during descending phases correlations were poorer. The SS
activity seemed to have a better correlation during the cold phase of PDO. These results show that
warm events tend to occur in the ascending phase or at the top of the cycle, and have a delay time
of around 36 months, whilst cold events are associated with descending phases but with a shorter
lag time. The correlation analysis given here indicates sun spot activity should be considered as a
factor that could condition and trigger low (PDO and AMO) and high (ONI-El Niño) frequency
oceanographic events in the Pacific and Atlantic Oceans.

**Key words:** Sun spots cycles, SST, SSTA, ONI, MEI, PDO, AMO, El Niño, La Niña

**Introduction.**

Essentially, the only external source of energy to Earth is the sun, which constantly radiates a flux of energy to the upper atmosphere (the solar constant) of 1360 W $m^{-2}$ or 1.36 kJ $m^{-2}$ $s^{-1}$ (Monteith,

1972) or 1.92 ly $day^{-1}$ (Ormaza-González and Sanchez, 1983). Recently, Kopp and Lean (2011) have reported that the most accurate accepted solar constant value is 1360.8 ± 0.5 W $m^{-2}$.  Of this flux of energy, 75-50 % reaches the Earth's surface (Ormaza-González and Sanchez, 1983; Lindsey, 2009)

and the remainder is reflected and/or absorbed by clouds, particles, gases, etc. (Horning et al.,

2003). About 90-93% of that energy reaching the surface is accumulated in the oceans (Trenberth et al., 2014; Clutz, 2017).  The solar constant is affected by variations in sun spot (SS) number (Bhowmik and Nandy, 2018) and other solar activity parameters by around 0.1%, i.e. on the order of 1.361 W

$m^{-2}$. The Hale cycle (around 11 years) is characterized by increasing and then decreasing SS numbers (Hathaway, 2015). Froelich (2013) suggested that the solar constant can vary by up to 4.0 W $m^{-2}$ over two SS cycles, i.e. a 22-year cycle, and proposed a simple relationship between SS and the solar constant (SC), by assuming a direct relationship between the two

*SC= 1353.6 + 0.089 (SS)*                   ($r^2$ of 0.71, 95-99% confidence).

The surface-subsurface layers of the ocean that interact with the lower atmosphere alternately release and absorb heat energy. The work of Zhou and Tung (2010) reported the impact of the SC on global SST over 150 years, finding signals of cooling and warming SSTs at the valley and peak of the

SS cycles.  Schlesinger and Ramankutty (1994) reported a global cycle of 65-70 years that is possibly affected by greenhouse anthropogenic gases, sulphate aerosols and/or El Niño events, but they did not imply an external force such as the SS. There are well known oceanic events that are roughly periodic with low (25-30 years) or high (3-5 years) frequencies.  These include the Pacific Decadal

Oscillation (PDO, Mantua et al., 1997; Mantua and Hare, 2002; Zhang et al., 1997; Yim et al., 2013),

Atlantic Multidecadal Oscillation (AMO, Enfield et al., 2001; Condron et al., 2005;  Gray et al., 2010)

and Interdecadal Pacific Oscillation (IPO, Henley et al., 2015), as well as El Niño (Busalacchi et al.,

1983, see **COAPS Library's: http://www.coaps.fsu.edu/lib/biblio/coaps-a.html**)  or La Niña (Yuan and Yan, 2012). During El Niño events, the surface and subsurface lose energy to the atmosphere and the opposite occurs during La Niña (Trenberth et al. 2014, Fasullo and Nerem, 2016); these events have a periodicity of 2-7 years while the decadal processes may take 25-30 years. The

Interdecadal oscillations have a series of impacts; e.g., the PDO gives rise to teleconnections between the tropic and mid-latitudes (Yoon and Yeh, 2010), and the effects include: 1) the ocean heat content (Wang et al., 2017), 2) the lower and higher levels of the trophic chain and small pelagic fisheries including tuna and sardines (Ormaza et al., 2016a, 2016b), 3) biogeochemical air-sea

$CO_2$ fluxes (McKinley et al., 2006), 4) the frequency of la Niña/El Niño (Newman et al. 2003). The interactions between decadal oscillations PDO/IPO and AMO may affect also ocean heat content (Chen and Tung, 2014).  All these low and high frequency oceanographic events have a direct impact on local, regional and global climate patterns, and there is growing evidence that the driving source of energy is the sun (Grey et al., 2010). Thus, Huo and Xiao (2016) have found a positive strong correlation between El Niño 2015-2016 and SS, as well as SS and the El Niño Modoki index. White et al., (1997) reported that heat anomalies produced by variable solar irradiance are stored in the upper ocean layer, driving SST changes of 0.01-0.03 K and 0.02-0.05 K on decadal and interdecadal periods respectively. Zong et al. (2014) in their review of the impact of the 11-year SS cycle and multidecadal climate projections, have found global SST variations of 0.08 ± 0.06 K and 0.14 ± 0.02

during the 11 and 22 year Hale Cycle, combined with a response lag of 1-2 years in relation to the SS

(see also, Kristoufek, 2017). Liu et al. (2015) have reported that effective solar radiation plays a role in the modulation of decadal ENSO (El Niño and the Southern Oscillation) oscillation. More recently,

Yamakawa et al. (2016) have reported that solar activities in terms of SS numbers not only affect the troposphere but also the sea surface, even though SS abundance is only a partial measure of solar activity (Scafetta, 2014).   The work reported here investigates how sun spots may affect low and high frequency oceanic events such as the Pacific Interdecadal Oscillation (PDO), the Atlantic multidecadal oscillation (AMO), anomalous sea surface temperatures, and El Niño and La Niña events.

**Material and methods.**

Data for monthly sun spot number (SS) were taken from the Royal Observatory of Belgium, Brussels,

World Data Center SILSO (http://www.sidc.be/silso/datafiles). Data sources for other variables were as follow:  El Niño regions areas 3.4 (5°North-5°S, 170-120°W) and 1+2 (0-10°S, 90°W-80°W):

- 101 • **Sea surface temperatures (SST) and SST Anomaly**: The Monthly Extended Reconstructed
- 102   Sea Surface Temperature Version 4 (ERSSTv4, 1981-2010 base period). The Optimum
- 103   Interpolation 1/4 Degree Daily Sea Surface Temperature (OISST.v2, 1981-2010 base period),
- 104   http://www.cpc.ncep.noaa.gov/data/indices/.
- 105 • **Oceanic Niño Index** (ONI: Huang et al., 2014): ERSST.v4  for El Niño/La Niña events since
- 106   1950 till December 2017:
- 107   http://www.cpc.ncep.noaa.gov/products/analysis_monitoring/ensostuff/ensoyears.shtml.
- 108 • **Multivariate ENSO index** (MEI: Wolter and Timlin, 2011):
- 109   https://www.esrl.noaa.gov/psd/enso/mei/table.html.
- 110 • **Pacific Decadal Oscillation** (PDO, based on Mantua Index): The PDO index is based on
- 111   NOAA's extended reconstruction of SSTs (ERSST Version 4). It is constructed by regressing
- 112   the ERSST Anomaly against the Mantua PDO index for their overlap period, to compute a
- 113   PDO regression map for the North Pacific ERSST Anomaly. The ERSST Anomaly are then
- 114   projected onto that map to compute the NCEI index. The PDO index closely follows the
- 115   Mantua PDO index at: https://www.ncdc.noaa.gov/teleconnections/pdo/ (Wolter and Timlin
- 116   1993, 1998 and 2011).
- 117 • **Atlantic Multidecadal Oscillation** index:
- 118   https://www.esrl.noaa.gov/psd/data/timeseries/AMO/.

All indexes have data from April 1954 to December 2017. The analysis was done using Excel and/or R statistical tools. The correlation exercises were executed using SS solar cycles as complete time series against SST Anomaly (in El Niño regions 3.4 and 1+2), ONI, MEI, AMO and

PDO indexes. Correlations with lags of 0 to 48 months were carried out. For the SS cycles 19-23

and their impact on the above mentioned dependent variables, correlations were carried out for the whole time series (1954-2017), and for 1990-2016, for each cycle and for their respective ascending and descending phases. Spectral analysis and polynomial regression fitting curves were determined to obtain the slope of the ascending phases; the slopes were correlated to the oceanographic indexes.

**Results and discussion:**

**Results and discussion:**

The time series (1954 to 2016) of SS, PDO, AMO, ONI and MEI are shown in Fig. 1. The PDO, AMO,

ONI and MEI cycles have been offset by 0, 12, 24 and 36 months (panels a, b, c and d respectively), whilst the SS series starts at t=0 in the four panels. It has been reported that the lag times for responses of some indexes to SS cycles (SS#) are around 12-36 months (e.g., Zhao et al., 2014). From

1954 to the present time each cycle 19 to 24 has occurred with a period of around 11 years (Hathaway, 2015), which is slightly less than the 11.2 years reported by Dicker (1978). The highest SS

activity is seen in cycle 19 with around 250 SS/month, followed by <150, and at cycle 21 around 200, before decreasing steadily over cycles 22 to 24 to just over 100 SS/month. Cycle 24 is the lowest contemporary value of SS activity that is comparable only to cycles 12-15 (around 1880-1930) and is the lowest in the last 200 years (Clette et al., 2014). The negative or cold PDO phases (1947-1976,

2000-June/2016) are within SS cycles 19-20 and 23-24, whilst cycles 21 and 22 are within the positive or warm phase of the PDO (1977-1999). As the PDO and AMO indexes are displaced from 0

to 36 months on the time scale (Fig. 1), some peaks and troughs relative to SS activity can be seen.

These are at ascending and descending parts of the SS cycles, e.g. during cycles 19-20 and 23-24 PDO

indexes are basically negative, whilst during 21-22 they are more positive; an exception is around

1990, where there is a strong negative peak. However, AMO phases seem to be in opposition to and overlapping the SS cycles; a cold phase of AMO was between the 1960s and 1990s, whilst the warm phase is from the 1990s to the present (McCarthy and Haigh, 2015).

The ONI and MEI curves, both indicators of ENSO events, behave similarly throughout the study period (April 1954 – December 2017). However, MEI has the highest anomaly peaks (> 2) when compared to ONI. In general, ONI and MEI curves indicate the highest positive anomalies between

1978 and 1995, a period that coincides with the warm and cold phases of PDO and AMO respectively (see Maleski and Martinez, 2018). The opposite trend occurs before and after this period due to the inversion of phases. In addition, the highest peaks of both indexes only occur during the ascending and descending phases of the solar cycles; that is, they never coincide directly with the maximum period of sunspots in the cycles, except in 1959. The two highest MEI peaks occur during the descending phase of solar cycle 21 and ascending phase of solar cycle 23. In mid-2016  both indexes increased reaching the third highest peak of this period during the descending phase of

SS#24. Negative peaks of these indexes occurred either in the high or low plateaus of the SS curves.

The number (N) of data in the analysis were: 765 (1965-2017); 312 (1990-2016); 108 (1990-1999);

192 (2000-2016). For individual cycles 19 to 24: 127, 141, 124, 117, 141 and 120 respectively. In the same order for ascending-descending phases: 48-80, 50-92, 43-82, 33-85, 51-51 and 74-47.  The degrees of freedom of residuals were N-2. The degree of correlation in terms of Pearson coefficient is referred to as: high, moderate and low when the coefficient is between ±0.5 and ±1.0, ±0.3 to

±0.49 and less than ±0.29 respectively (http://www.statisticssolutions.com/pearsons-correlation- coefficient/). All linear regression residuals were auto correlated using the Durbin-Watson (DW) test (Montgomery et al., 2001) for 1954-2017, 1990-2016, 1990-1999, 2000-2016, individual cycles, and ascending-descending phases. The DW test results for the long time series averaged 0.122, for individual cycles it varied from 0.10 to 0.63 with an average of 0.18, and for the ascending and descending phases it averaged 0.40 and varied from 0.1 to 2.24.  The SST Anomaly in region ENSO

1+2 has the lowest and PDO the highest.

**Whole series (1954-2017) correlations.** All variables (Table 1) were correlated on a linear and polynomial (n= 2 to 6th order) basis using different lag times (0, 6, 12, 24 and 48 months) over the six SS cycles. Polynomial correlation (not shown) as well as linear ones displayed poor correlation coefficients, with the highest linear $r^2$ (p≤0.01) coefficients occurring at lag times between 12 and 24

months, and 36-48 months for the AMO. For SST and SST Anomaly in ENSO areas 1+2 there was no correlation. These results are like those of Kristoufek (2017), who suggested a surface thermal response of around 24-36 months. The highest correlation $r^2$ values with SS were up to: 0.043, 0.029,

0.040 and 0.021 for PDO, MEI, ONI and SST Anomaly (in 3.4) respectively, indicating there is a correlation with high confidence (p-value ≤0.01), though small $r^2$. This fact could reflect sun activity (sun spots) in the long term being balanced by the ups and downs of the cycles. This correlation exercise suggests there is not a good correlation of these indexes in the Pacific and Atlantic over the studied time scale.  Nonetheless, on longer time scales, where SS cycles are affected by other sun internal processes, e.g., the hypothesized Minimum of Maunder (Eddy, 1976, Shindell et al., 2001,

Ineson et al., 2015, Mörner, 2015, etc.), there can be an impact on a global and regional basis.

Recently, Lockwood (2010 and 2013) has reported that a grand solar minimum is coming as the SS

cycle 24 is developing. There has not been a solar activity decline such as that found in SS# 23 to 24

over the last 9300 years, and such a solar minimum may last through cycles 24, 25 and 26 (Hady,

2013). Under these circumstances where anomalous conditions appear to be developing, it was decided to analyze correlations using individual cycles in the range 19 to 24.

**Period 1990-2016.** Further analysis was carried out for the period 1990 to 2016, that includes cycles

22, 23 and 24. The time series was also split into 1990-1999 and 2000-2016, because during 1990-

1999 a strong (1991-1992), moderate (1994-1995) and the strongest El Niño (1997-1998) of the twentieth century occurred. On the other hand, in 2000-2016 (cold phase PDO) there were strong La

Niña events (2000-2002 and 2010-2012) and an El Niño Modoki event in 2015 (Huo and Xiao, 2016).

Figure 2 shows again a poor correlation (<0.011, p>0.246), for the SST Anomaly in region 1+2 (blue bars), although this region was gravely affected by the strong El Niño in 1997-1998 which brought hundreds of casualties and losses of billions of US dollars to the Ecuadorian infrastructure (Glantz,

2001). The linear correlation $r^2$ of SST in 3.4 (red bars) was around 0.1193 (p≤ 0.00001) over the whole period, whilst it was somewhat higher in the period 1990-1999 (0.1519, p≤ 0.01). The ONI

(green bars) correlation coefficient was up to 0.1436 (p≤0.02) when compared to SS in the period

1990-2000, where high positive SST anomalies were present for almost 6 years, and the ONI

correlated better than SST anomaly with SS in 3.4. The Pacific Decadal Oscillation (Fig. 2., grey bars)

had an $r^2$ of 0.276 (p<0.0001), in the period 2000-2016 (PDO in a cold phase), with a Pearson correlation of 0.523 that can be considered as high (https://www.statisticssolutions.com/pearsons- correlation-coefficient/). However, for the period 1990-1999 it was 0.239 and for the whole period was 0.402; i.e. a poor and fair degree of correlation respectively.

**Individual Cycles.** Correlation analysis was split into SS cycles from 19 to 24. The SS and SST $r^2$

coefficient indicated a poor correlation and confidence level (p≥0.05) in region 1+2 in all cycles (Fig.

3); most of the correlation $r^2$ values were <0.050, except in cycle 19, ($r^2$ of 0.0675, p=0.0032); in cycles 21 and 23 the highest $r^2$ was 0.046 (p=0.0173) and 0.048 (p=0.037) respectively. In general, the lag time varied between 6 to 36 months in region 3.4, whose correlations $r^2$ were up to 0.219

(p≤0.0001) and 0.213 (p≤0.0001) for cycles 23 and 19 respectively with a lag time of 12-36 months.

Cycles 20 and 22 had $r^2$ of 0.105 (p≤0.0001) and 0.074 (p=0.003) respectively. The slopes of the linear regression curves with the highest $r^2$ were positive in region 3.4, indicating a direct correlation between SST and SS cycles. However, cycles 22 and 23 in the region 1+2 exhibited inverse correlation (Fig. 3). Further polynomial correlation (n=2 to 6) analysis did not provide a better $r^2$.

Over all, higher correlations were found in El Niño regions 3.4 than in 1+2.

**Anomalies in SST.** The magnitude of the SST Anomaly can change depending on the reference used; there are 5 versions of ERRS (Huang et al., 2017). Currently version 5 tends to be used in El Niño studies. Here we used the ERRSv4 (Huang et al., 2014); Huang et al. (2017) stated that there is not a noticeable difference between ERRSv4 and ERRSv5. The anomalies of SST in 3.4 and 1+2 were also correlated against every SS cycle; correlation $r^2$ values were not better than 0.396 (p≤0.0001) in both regions, with higher variability in 1+2 than 3.4 both in response time and correlation coefficient (Fig.

4). In region 3.4, the highest correlations were 0.289 (p≤0.0001) and 0.270 (p≤0.0001) during cycles

19 and 23 respectively, with a lag time of between 12 and 36 months, with both occurring during cold phases of PDO (1955-1978, and 2000-present). Surface winds plus other oceanographic variables (e.g. upwelling) could play an important role in this high variability. Winds are not only generated in the local area but farther away, including the trade winds of the western Atlantic (Ormaza-González and Cedeño, 2017). Also, ENSO processes in the western Pacific could add variability in the SST Anomaly. The slopes of the linear correlation were basically positive for 3.4 and negative for 1+2, and for SST correlations for cycles 19, 23 and 24 (cold PDO phase) had the highest

$r^2$. Again, the anomalies in 3.4 were better correlated than in region 1+2.

**ONI.** The El Niño index (Fig. 5) displayed $r^2$ values when correlated with SS activity from around

0.053 (p=0.01, SS# 22) up to 0. 25 (p<< 0.0001, SS#24); there were poor to fair correlations with a positive slope in SS cycles 19, 23 and 24. During SS# 24, ONI reached extreme values of 2.6C (Nov-

Dec-Jan 2015/2016) and -1.7C (Oct-Nov-Dec 2010). The highest $r^2$ were again found with a 24-48

months lag time. Cycle 21 did not show any significant correlation with ONI; however, cycles 20, 22

and 24 had $r^2$ values of 0.144 (p<<0.001), 0.131 (p<<0.0001) and 0.252 (p<<0.0001) and lag times of

48, 12, and 24 months respectively. Recently, Huo and Xiao (2016) found strong correlation between

SS and El Niño Modoki during 2015 (SS#24). The variability in correlations  could arise from: 1) SS

numbers showing large variations from one month to another, 2) regional meteorological conditions (particularly cloudiness), ocean surface currents that exchanges heat in region 3.4, 3) Kelvin waves (Gill, 1982), 4) the Southern Oscillation Index (SOI: Southern Oscillation Index:

http://www.cpc.ncep.noaa.gov/data/indices/soi).   All these may affect the SSTs and together with the way ONI is obtained, as the ONI has a variable reference period of 30 years; thus for 1950 to

1955 the reference period is 1936-1965; for 1956-1960; 1941-1970.  The ERRSv4 uses the period

1981-2010. The reference period is changed every 5 years (Lindsey, 2013); the most recent ONIs (v4/v5) are supposed to use better and more consistent information as data acquisition improves.

**MEI.** This additional index for El Niño events had a lower correlation ($r^2$) with SS; thus, the highest value was 0.3943 (p<<0.00101, SS# 19), the next 0.3028 (p<<0.00001, SS#24), 0.2421 (p<<0.00001,

SS# 23) and 0.1566 (p<<0.0001, SS# 20); in cycles 21 and 22 no correlation better than 0.1232

(p<<0.0001) was found. The lag time for sun spot activity (with the highest correlations) ranges from

24-48 months; and linear regression curves were with mainly positive slopes. The lower correlation found could be explained as this index comprehend six variables, and some of these could not be directly or are weakly related to SS; like zonal and meridional components of surface wind, surface air temperature, cloudiness (Wolter and Timlin, 1993).

**PDO.** This interdecadal index (Fig. 5) is linearly correlated to SS cycles with a lag time between 36

and 48 months, with the highest $r^2$ of 0.391 (p<<0.00001) in cycle 19, and 0.586 (p<<0.00001) for cycle 24. Both cycles are within the cold phase of the PDO. The next highest $r^2$ with p-values

<<0.0001 were 0.218, 0.1361, 0.218 and 0.260 for cycles 20-23. In all cycles, the highest $r^2$ were directly correlated, except cycle 20. For some reason, there appears a better fit with both PDO and

ONI in cycles 19 and 24, which are within the cold phase of the PDO, even though these cycles have remarkably different shapes and peaks (Fig. 12). Cycle 19 registered SS counts of over 250 whilst cycle 24 was just around 100; also, the peaks were different being respectively very sharp and extended. The direct relationship between PDO and ONI has been reported extensively (e.g.

Ormaza-Gonzalez et al., 2016 a, Jia and Ge, 2017).

**AMO.** This index gave correlation coefficients ($r^2$) with SS numbers of up to 0.490 (p<<0.00001) and down to 0.162 (p=0.0004) in cycles 20 and 24 respectively, when a lag time of 48 months was used.

With cycles 19, 21 and 22 the best fit elapsed time was 24-36 months. Gray et al. (2016) reported lag time responses of mean-sea-level pressure over the Atlantic to SS cycles of 36-48 months over a longer time series study of 32 solar cycles. Figure 6 shows the bar distribution of the $r^2$; it displays linear regression with positive and negative slopes for cycles 19, 23 and 24; and 20 to 22, respectively. This coincides with the phases of the AMO, negative from around 1965 to 1998 (SS

cycles 20-22), and positive; 1930-1965 (SS cycle 19) and after 1998 (SS cycles 23-24), http://appinsys.com/globalwarming/amo.htm . It is noteworthy that the slopes of the PDO and AMO

linear regressions are negative and positive respectively for cycles 21 and 22, but in concordance in cycles 19, 20, 23 and 24 (cold phase PDO).

**Ascending and descending phases** of solar cycles. As the SS cycles are best related to variables studied on a response time from 24 to 36 months, there was the need to study their influence during the ascending and descending phases, which have roughly 5-6 years duration. Polynomial regression analysis was performed to establish a function that could best describe every SS cycle.  Sixth-order polynomial curves (Fig. 12) were found to render a very strong correlation coefficient averaging 0.89

(p≤0.001). These functions allowed the analysis of correlations in the ascending and descending phases.

**SST in 1+2 and 3.4.** In region 1+2, the highest correlation coefficient $r^2$ and p-value were, respectively, 0.205 and 0.0008 (SS# 23), 0.189 and ≤0.0036 (SS# 21), and 0.163 and ≤0.0044 (SS# 19).

All linear regression coefficients $r^2$ over 0.0847 (p<0.05 to =0.0008) occurred in the ascending phase of the SS cycles, whilst those in the descending phase were lower, with no definite lag time pattern from 0 to 36 months. The slope (positive/negative) of the linear regression (Fig. 7) curves showed no pattern. These low and variable $r^2$ values reflect region 1+2 being subjected to the combined impact of many diurnal and seasonal oceanographic and meteorological variables.  For example, during the first quarter of 2017 (cycle 24), in 1+2 there was a higher than usual SST because the southern trade winds in the eastern Pacific weakened whilst those in the North Atlantic strengthened.  These winds passed through the Panama Isthmus and blew warm (up to 30C) surface waters from the Panama

Bay southwards towards area 1+2, thus provoking a rapid and relatively short lived surface warming event (Ormaza-González and Cedeño, 2017), whilst region 3.4 was registering La Niña conditions.

This cold event also strengthens the Cromwell Undercurrent (Knauss, 1959) and Humboldt (Montecino and Lange, 2009) currents that impact upwelling processes in 1+2. During the ascending phases of the cycles, the correlation of SSTs was higher than in the descending phase of cycles. All these factors would mask the SS signal in this area.

In the region 3.4, the maximum $r^2$ of SST in each SS cycle was found at a lag time of 36 months with all of them occurring at the ascending phase, except in cycles 20 and 24. The four highest $r^2$ values were 0.870 (p=0.021, SS# 24), 0.613 (<<0.0001, SS# 22), 0.574 (p<<0.0001, SS# 19), and 0.556

(p<<0.0001, SS# 23) with Pearson coefficients of 0.9327, 0.7803, 0.7576 and 0.7456, respectively, thus showing a strong degree of linear correlation. Linear regression slopes were variable (Fig. 7), although there was a tendency in cycles 20, 21 and 22 (warm PDO) for negative slopes and for positive slopes for cycles 19, 23 and 24 (cold phase PDO). In area 3.4 the SST response to SS was much clearer than for 1+2, as in this region (10N-10S and 120W-180W) there is no influence of coastal processes. The highest $r^2$ (0.870, p=0.021; lag time 36 months) in the descending phase in cycle 24 coincided with the strongest El Niño, and the second-highest $r^2$ (0.613, p<<0.00001) during ascending phase of cycle 22 with two consecutive strong El Niño 1991-1995; the third $r^2$ (0.574, p<<0.00001) during cycle 19, with el Niño 1955-1957, and finally the fourth $r^2$ (0.556, p<<0.00001)

with the 1997-1998 warm event during cycle 23. It seems that over the relatively short time scales of

SS cycles, either on their initial ascending or subsequent descending phases, impacts on the SSTs can be triggered.

**SST Anomaly.** In region 1+2 (Fig. 8), the anomalies registered high $r^2$ (p<<0.0001) of 0.662 (SS# 22),

0.637 (SS# 19), 0.523 (SS# 21), 0.480 (SS# 23), 0.359 (SS# 24); and 0.254 (p=0.0002, SS# 20)

respectively, in the ascending phase of the SS cycles and with a positive linear regression slope (except SS# 23). The response lag time was somewhere between 0 and 48 months. On the other hand, the descending phase showed a predominantly lower $r^2$, less than 0.14 with lower significance (p ≤ 0.02), with the exception in SS# 19, 0.304 (p<<0.0001). The results suggest that during cold phase PDOs when Northern Pacific basin surface ocean waters are relatively colder, the correlations in this area tend to be higher, as the increasing sun radiation augments the heat content (SST) of the ocean surface.

In region 3.4, there was a high and consistent $r^2$ (Fig. 8) that reached up to 0.897 (p=0.014; SS# 24);

0.863 (p<<0.0001; SS# 22); 0.665 (p<<0.0001; SS# 19), 0.826 (p<<0.0001; SS# 23), then fell to 0.211

(p=0.008; SS# 20); 0.239 (p=0.0009; SS# 21) respectively; all of them were in the ascending phase except cycles 20 and 24. The lag time was consistent at 36 months. Linear regression slopes were variable (Fig. 8) with negative slopes in cycles 20-22 (warm phase PDO), and positive slopes in 19, 23

and 24 cycles (cold phase PDO). The highest $r^2$ of 0.897 at the start of the descending phase in 24

coincided with one of the strongest El Niño and the second $r^2$ (SS# 22 ascending phase) with two consecutive strong El Niño 1991-1995. The third and fourth highest $r^2$ were during El Niño 1955-1957

and 1997-1998 warm event (SS# 23 ascending) respectively. The results suggest that SS cycles are strongly correlated to SST Anomalies in both El Niño regions, with the strongest relationship in 3.4.

**The ONI index.** This index as well as SST and its anomalies in 3.4, were equally strongly associated with the ascending phase of the SS cycles (Fig. 9), with lag times of 24-36 months.  The highest correlation $r^2$ for each cycle were in the ascending phase, the predominant linear regression slopes were positive, except for SS# 20. The highest $r^2$ ($p \ll 0.0001$) were: 0.817 (SS# 22), 0.693 (SS# 19),

0.637 (SS# 23), 0.3547 (SS# 24), 0.2876 (SS# 20); 0.1936 ($p=0.003$, SS# 21). The three highest $r^2$

match the dates of full-fledged strong El Niño 1955-1957, 1987-1989, and 1997-1998 (Fig. 9) with positive slopes on the ascending phase. In the descending phase the $r^2$ ($p \ll 0.0001$) in cycles 24, 23,

22 and 20 were 0.366, 0.284, 0.255, and 0.242 respectively. All had a lag time 0-12 months and positive slopes. Cycles 19 and 21 showed neither strong correlation (<0.1) or confidence values ($p=0.2$). The ONI correlations are in accordance to what found with SST anomalies in 3.4.

Warm events tend to occur in both ascending/descending phases after the peak/trough, and have a delay time of 36 months, which is similar to findings of Huo and Xiao (2016). The delay time, is probably due to the slow accumulation of solar heat over time in surface oceanic waters. The descending phase of the cycles (Fig. 9), with a smaller slope than the ascending phase, produces a quicker response (0-12 months) to the ocean surface SST and ONI that could trigger neutral or cold events more cogently.  Most of the la Niña events occur during the descending phase or approaching the cycle minimum (Fig. 10), when the solar irradiance (SI) decreases slightly as does the number of sunspots. The weakest sunspot cycle (SS# 24) has had three La Niña events: 2007-2009, 2010-2012,

2016-2017 (Fig, 12). A plausible reason is that during this cycle the number of sun spots (i.e. sun activity) is the lowest in the last two centennials (Clette et al., 2014); therefore, less energy has hit the ocean surface allowing a cooling effect. Two important exceptions are La Niña 1988-1989 (22)

and 2000-2002 (cycle 23) that occurred in the ascending phase.

**The MEI index.** The Multivariate ENSO Index does not only consider the SST Anomaly but also sea- level pressure (Allan and Ansell, 2006) and other variables. These variables include surface winds (meridional and zonal), surface air temperature and cloudiness (Wolter and Timlin, 1998). The MEI

correlated at slightly lower levels with SS cycles with $r^2$: 0.784 (p ≤ 0.0001), 0.770 (p<<0.0001),

0.5972 (p ≤ 0.0001); 0.3396 (p ≤ 0.0001); 0.2368 (p=0.0003); and 0.222 (p=0.001) for SS cycles 19, 22,

23, 24, 20, and 21, respectively.  All of them were in the ascending phase of the cycles with lag times from 12 to 48 months (except cycles 23 and 24), with a positive linear regression slope. Exceptions were 22 and 20 where the $r^2$ was largest with a zero lag time. During the descending phase, as with the ONI, the $r^2$ were lower: 0.321 (p=0.0004, SS# 24); 0.3145 (p<<0.0001, SS# 19); 0.2234

(p<<0.0001, SS# 22); 0.2088 (p<<0.0001, SS# 20), and 0.1438 (p=0.0002, SS# 23) with positive slopes (except SS# 20), and lag times predominantly in the 0-48 month range; cycle 21 did not have a $r^2$

above 0.010 (p>0.02). For the MEI index, as with ONI, the $r^2$ were much lower during descending phases, when there is less sun radiation energy (see formula 1), thus La Niña events could be expected as it actually has occurred in the six cycles. The lower correlations could be because the

MEI uses five variables more than the ONI, and these could thus help obscure the signal from the sun's irradiation.

**PDO.** The Pacific Decadal oscillation gave positive linear correlations and slopes with SS in most cycles except cycles 20 and 21. Correlation coefficients of 0.7677 (p ≤ $10^{-12}$), 06577 (p ≤ $10^{-12}$), 0.6734

(p ≤ $10^{-7}$) and 0.5062 (p ≤ $10^{-7}$) for the ascending phase SS# 19 (Apr/54-Nov/58), SS# 24 (Jan/08-

Feb/14), SS# 22 (Sep/86-Jan/89) and SS# 23 (May/90-Jun/00) respectively were found. All these coefficients were obtained at a lag time of 12-48 months, except 22 and 23 (t=0). The slopes of the linear regressions were mainly positive during cold phase PDOs (cycles 19, 23 and 24), except cycle

20 when a cold PDO was transitioning to a warm PDO (cycles 21 and 22).  Figure 10 shows that linear correlations in cycles 19, 21, 23 and 24 showed positive slopes. The two highest $r^2$ values are at a lag time of 12-36 months, for cycles 19 and 24, as has been reported (e.g., Huo and Xiao, 2017). During the descending phase, the correlation $r^2$ tended to be much lower, with the highest 0.3522

(p<<0.00001) and 0.3452 (p<<0.00001) at cycles 19 and 20. Sun spot energy variations on long time scales (van Loon et al. 2007), even with very weak changes, could produce decadal and millennial timescale impacts on global thermohaline circulation that in turn affect heat distribution (Bond et al.

2001, Gray et al., 2013).

**AMO.** The correlations were generally higher at the descending phase of the SS cycles (Fig. 11), which is opposite to those for SS vs PDO, ONI, MEI, and the SST Anomaly. However, the highest $r^2$

occurred on ascending (A) and descending (D) phases of SS cycles, thus: 0.700 ($p \ll 10^{-10}$), 0.558 ($p \ll$

$10^{-10}$), 0.468 ($p \ll 10^{-10}$), 0.434 ($p=0.03$), 0.411 ($p \ll 0.00001$) and 0.191 ($p=0.001$) for cycles 20A, 22D,

19D, 24D, 21A and 23A, respectively. These high $r^2$ values show a strong degree of correlation, although lower than PDO correlations. The lag time was between 24-48 months. The results found could be explained as The Atlantic Multidecadal Oscillation index has the opposite phase to the PDO

(Enfield et al., 2001; Condron et al., 2005); i.e. warm in periods 1930-1964 and 2000-present (cold

PDO), and cold in 1965-1999 (warm PDO).

**Conclusions**

**Period 1954-2017.** Over this period sun spot numbers have decreased from between 225 (SS# 21)

and 110 (SS# 24) at cycle maxima, to minima SS counts of around 20-25. Thus, the Earth is receiving decreasing solar energy over this almost 7-decade period. The reduction of SS peaks has been associated with the beginning of the Maunder Minimum (Mörner, 2015). Ineson et al. (2014) are projecting lower peaks for the next SS cycle (SS# 25) and presently SS counts per month are as low as

1.6 (July 2018) and with an average of 8.5 (Jan-Aug 2018); counts are expected to decrease to 5.3 in

February 2019, actually it decreased to 0.8 (http://www.sws.bom.gov.au/Solar/1/6)

Monthly SS count correlations with SST, SST Anomaly (both 3.4 and 1+2), ONI, MEI, AMO and PDO

through the whole time series (1954-2017) were poor; these had a correlation $r^2$ values averaging

0.020 and a negative linear regression slope. Thus, in the long term there are no strong correlations between SS and PDO, MEI, ONI and SST Anomaly in 3.4 (correlation coefficients were between 0.043

and 0.021). In the case of region 1+2, the correlation was even poorer: <0.005.

The series of correlations at different lag times (6, 12, 24, 36 and 48 months) gave a response time (i.e. the lag time with highest correlation coefficients; Table 1) of 12-24 months for all indexes, except for AMO (48 months), which align to what was previously reported by Kristoufek, (2017), and

Huo and Xiao (2016);

Changes of the SS cycle could have climate impacts. Gil-Alana et al. (2014) have found no significant statistical relation between sun spots and global temperature; however, van Loon et al. (2007)

suggested that even though SS cycles produce weak changes on Solar Irradiation (SI) of about 0.07%

according to Gray et al. (2010), these can still produce decadal and millennial impacts on global thermohaline circulation (Bond et al. 2001, Gray et al., 2016), due chiefly to UV energy fluctuation (Ineson et al., 2014). The changes in UV (<100 nm to 350 nm) and near infrared (>800 nm to >1000

nm) are larger than in the visible range (>350 nm-800 nm) and could have an important impact on global climate (Ermelo et al. 2013).  It is therefore reasonable to expect some impact on the studied oceanographic indexes.  Recent data (Solar Radiation and Climate Experiment Satellite) suggest that the variability of UV radiation during the declining phase of cycle 23 was larger than previous estimates (Harder et al., 2009 and Haigh et al., 2010).  The SI variations are strongly correlated to SS, and even though these are relatively small (Hansen et al., 2013), they may impact surface ocean heat content because: 1) the total SI integrates over all the wavelengths, and 2) the heat capacity of the seawater is huge.  Also, UV radiation penetrates down to 75-100 m depth in the water column (Smyth, 2011), thus adding to the heat content of the deeper layers.

**Individual SS cycles (19-24)**. The SST shows some correlations against individual SS cycles in regions

3.4 and 1+2. In the first (Fig. 3), these were up to 0.219 (SS# 23) whilst the lag time was 12-36

months in all cycles (except SS# 19), in line with reports from Kristoufek (2017) and Huo and Xiao (2016). On the other hand, in region 1+2, the linear correlation $r^2$ was low at <0.0675 (SS# 19), and highly inconstant between cycles. The SS and SST anomaly correlations in 3.4 and 1+2 (Fig. 4), showed important variability with highest values of 0.289 (in 3.4) and 0.396 (in 1+2) during cycles 19

and 24 respectively, with both having the same response time range. These values are within the cold phase of the PDO, suggesting that during this phase the signal from SS is clearer. In the period

1997-2016 the two strongest El Niño (1997-1998 and 2015) and La Niña (2000-2002 and 2010-2012), events occurred, and they were most evident in the 1+2 region. The slopes of the linear correlation were positive for 3.4 and negative for 1+2, and in general it was found that correlations were more inconstant and lower in 1+2 than in 3.4.  These results do suggest that Sun spot activity can influence the SST and SST anomaly behavior, but the relatively weak signals may well reflect high seasonal and interannual variability in coastal oceanographic conditions (Ormaza-González and Cedeño, 2017)

that obscures the correlation with SS. In zone 3.4, correlations are better, although the influence of regional oceanographic and meteorological conditions will still be there as expressed through, e.g., the Southern Oscillation Index (Rasmussen and Carpenter, 1982; Barnston, 2015). During cycle 24, the ONI index was highly correlated to SS ($r^2$ up to 0.2510) with a positive slope (Fig. 5).  The ONI

index reached 2.6C (Nov-Dec-Jan 2015/2016) and -1.7C (Oct-Nov-Dec 2010). In cycles 20-21 the $r^2$

was low ($r^2$<0.04); however, from 22 to 24, $r^2$ increased from 0.131 (p<<0.00006) to 0.251; thus, confirming the SS activity has a better correlation during the cold phase of PDO.

The PDO aligned best with SS cycles at lag times of 36-48 months during SS#19 (0.625) and SS#24

(0.766) when the PDO was in a cold phase, whilst lower correlations (0.467-0.508 for cycles 20-23)

were found when it is in a warm phase (1979-2000) or in between them. The North Atlantic index counterpart, the AMO index, gave a variable correlation $r^2$ ranged from 0.130 to 0.490 with a response time of 48 months for cycles 23 and 20 respectively. Gray et al. (2016) reported 36-48

months lag for mean-sea-level pressure in the North Atlantic in a study of 32 SS cycles. The slopes of the PDO and AMO linear regression curves are negative and positive respectively in cycle 21 and 22, but in concordance in 19, 20, 23 and 24 during the cold phase of the PDO. These two interdecadal oscillations proved to be correlated to SS, with the PDO having a higher correlation. Presumably, as the North Pacific Ocean basin has a larger area than the North Atlantic the correlations and lag times may reflect the higher heat storage capacity in the North Pacific.

**Ascending and descending phases**. The SSTs in 1+2 showed higher correlations with SS in the ascending phases, relative to the descending phase ($r^2$ of up to 0.205 and below 0.067 respectively).

In region 3.4, there were again high degrees of correlation of SS and SST ($r^2$ between 0.87 and 0.56), during the ascending phase of the cycles, with a response time of 36 months. The highest $r^2$ of 0.870

in the descending phase in cycle 24 coincided with the strongest El Niño (2015) and the second highest (SS# 22) with two consecutive strong El Niño events in 1991-1995; the third and fourth highest corresponded with el Niño and warm events respectively in 1955-1957 and 1997-1998.  It seems that short time expressions of SS cycles, either at the beginning of their ascending or descending phases, have a trigger effect on the SSTs. This was observed through the polynomial regression curves (Fig. 12) that were found for each SS cycle, as the SI (equation 1) increases and decreases the amount of heating of surface waters follows suit. The polynomial curves (6$^{th}$ order)

were fitted with an average r$^2$>0.89. However, a response time of 24-36 months seems to occur at the low or high plateau of the cycles (Fig. 12). Thus, the warm events El Niño of: 1957-1958 (SS# 19),

1965-1966 (SS# 20), 1981-1982 (SS# 21), 1987-1988 and 1991-1992 (SS# 22), 1997-1998 (SS# 23),

2015-2016 (SS# 24). On the other hand, the cold events of La Niña tend to occur after an El Niño at the middle of the ascending phases (1988-1989, 1999-2001, 2010-2012) or when approaching the minimum of the cycles (1973-1974, 1975-1975; 1995-1996, 2017-2018). The so called equatorial

Pacific neutral conditions in 3.4 (see, https://iridl.ldeo.columbia.edu/maproom/ENSO/ENSO_Info.html), seems to span a longer period after La Niña, and vice versa after El Niño.

The ENSO indexes ONI and MEI also showed strong correlations to the ascending phase of the SS

cycles, with a lag time of 24-36 months. In this analysis, it was also found that warm events tend to occur in the ascending phase (SI increases) or at the top of the cycle and have a delay time of 36

months (as also reported by Huo and Xiao, 2016), whilst cold events are mostly associated with a descending phase (SI decrease) but with a quicker response time of 0-12 months. During the descending phase, the correlation coefficients were lower and variable with positive slopes and shorter lag times of 0-12 months. The MEI index has a similar pattern to the ONI, but with lower correlations that may arise as the MEI takes into consideration six variables that in combination may mask the signal from sun activity.

The PDO (Fig 10) was linearly correlated to SS ($r^2$ ranging from 0.285 to 0.768) in the ascending phase with lag times of 24-36 months (Huo and Xiao, 2016), whilst correlations with AMO (Fig. 11)

varied with $r^2$ between 0.160 to 0.700. Similarly, the response time was 24-48 months. These results correspond with those of van Loon et al. (2007), who established that even a small change (1.5 W m$^-$

$^2$) in sun activity (SI) could produce decadal and millennial time scales influences on thermohaline circulation (Bond et al. 2001, Gray et al., 2016); nonetheless the Intergovernmental Panel on Climate

Change: IPPC (2001) considers this fact too small to drive climate changes. These influences of this change can be reflected by PDO and AMO indexes, which are found to be in the opposite phase to

PDO (Enfield et al., 2001; Condron et al., 2005). The ascending and descending phases of SS could re- inforce and weaken these indexes.

Recent predictions for an El Niño event in the late northern hemisphere summer of 2018 (see:

http://www.bom.gov.au/climate/enso/) did not occur; then, projections were pushed back to the beginning of autumn (http://www.cpc.ncep.noaa.gov/products/precip/CWlink/MJO/enso.shtml)

and most recently late autumn and winter. Thus, most current models are failing to provide a consistent projection. This is likely related to two re-cooling events in all El Niño areas, that have kept the ONI index within the realm of ENSO neutrality (-0.5C to 0.5C). Also, the PDO indexes have been negative, averaging -0.53, and it is now in its cold phase (https://www.ncdc.noaa.gov/teleconnections/pdo/). During 2017 the average smoothed SS counts per month was 21.8, and for 2018 it was 8.5 with many weeks without any SS.  In July the average SS

count was just 1.6  (http://www.sws.bom.gov.au/Solar/1/6) with counts expected to decrease to 5.3

in February 2019 (actually they were just 0.8). Therefore, the input of solar heat has been at its lowest values since the 1950s, and its trigger effect on ENSO system is not enough for a full-fledge El

Niño event. At the time of writing this paper, the expected and modelled 2018 full-fledged El Niño

2018 did not occur, it did not happen, actually the region 3.4 and 1+2 cooled down by the end of

2018. During March 2019, the latest report from http://www.bom.gov.au/climate/enso/#tabs=Overview is saying "The El Niño–Southern Oscillation (ENSO) remains neutral. However, the Bureau's *ENSO Outlook* remains at El Niño WATCH, meaning there is approximately a 50% chance of El Niño developing during the southern hemisphere autumn or winter…". Nonetheless, NOAA view is that a weak El Niño conditions are present now and to expect to continue through the Northern Hemisphere spring 2019 (~55% chance), https://www.cpc.ncep.noaa.gov/products/analysis_monitoring/lanina/enso_evolution-status-fcsts- web.pdf. The fact is, El Niño is not fully-fledged; *id est,* the meteorological (SOI) and oceanographic are not connected. At this moment (march 24, 2019) the oceanographic and meteorological variables have not fully couple yet.

The ENSO modellers should take into account in some way the presence of SS or any variable that could measure the variability of SI as an input to the models, specially the determinists ones.

[revised manuscript text omitted]

26. Hansen J, Kharecha P, Sato M, Masson-Delmotte V, Ackerman F, et al. (2013) Assessing

''Dangerous Climate Change'': Required Reduction of Carbon Emissions to Protect Young

People, Future Generations and Nature. PLoS ONE 8(12): e81648.

doi:10.1371/journal.pone.0081648

27. Henley, B.J., Gergis, J., Karoly, D.J., Power, S., Kennedy, J., and Folland, C.K.: A Tripole Index for the Interdecadal Pacific Oscillation, Clim. Dynam., 45, 3077, doi:10.1007/s00382-015-

2525-1, 2015.

28. Horning, N., Russell, C., and Goetz, S.: Energy from the Sun to Earth's Surface. In Chapter 2:

Earth's Radiation Balance and the Global Greenhouse, https://people.ucsc.edu/~mdmccar/migrated/ocea80b/public/lectures/lect_notes_1/03_En ergy_Balance_MDM_11F.pdf (last access: 25 April 2018), 2003.

29. Huang, B., Banzon, V.F., Freeman, E., Lawrimore, J., Liu, W., Peterson, T.C., Smith, T.M.,

Thorne, P. W., Woodruff, S. D., and Zhang, H. M.: Extended Reconstructed Sea Surface

Temperature version 4 (ERSST.v4): Part I. Upgrades and intercomparisons, J. Climate, 28,

911–930, doi:10.1175/JCLI-D-14-00006, 2014.

30. Huang, B., Thorne, P.W., Banzon, V.F., Boyer, T., Chepurin, G., Lawrimore, J.H., Menne, M.J.,

 Smith, T. M, Vose, R. S., and Zhang, H. M.: Extended Reconstructed Sea Surface

 Temperature, Version 5 (ERSSTv5): Upgrades, Validations, and Intercomparisons, J.

 Climate, 30, 8179–8205, doi:10.1175/JCLI-D-16-0836.1, 2017.

31. Huo, W.J., and Xiao, Z.N.: The impact of solar activity on the 2015/16 El Niño event,

 Atmospheric and Oceanic Science Letters, doi: 10.1080/16742834.2016.1231567, 2016.

32. Ineson, S., Maycock, A. C., Gray, L. J., Scaife, A. A., Dunstone, N. J., Harder, J. W., Knight, J. R.,

 Lockwood, M., Manners, J. C., and Wood, R. A.: Regional climate impacts of a possible future

 grand solar minimum, Nat. Commun., 6, 7535, doi:10.1038/ncomms8535, 2015.

33. Intergovernmental Panel on Climate Change (IPCC) (2001), Third Assessment Report–Climate

 Change 2001, The Scientific Basis, Cambridge Univ. Press, New York.

34. Jia, X. and Ge, J. (2017), Modulation of the PDO to the relationship between moderate ENSO

 events and the winter climate over North America. Int. J. Climatol, 37: 4275-4287.

 doi:10.1002/joc.5083

35. Knauss, J. A.: Measurements of the Cromwell current, Deep-Sea Res., 6, 265-286,

 doi.org/10.1016/0146-6313(59)90086-3, 1959.

36. Kopp, G., and Lean, J. L.: A new, lower value of total solar irradiance: Evidence and climate

 significance, J. Geophys. Res. Lett., 38L01706, doi:10.1029/2010GL045777, 2011.

37. Kristoufek, L.: Has global warming modified the relationship between sun spot numbers and

 global temperatures? Physica A., 468, 351-358, doi: 10.1016/j.physa.2016.10.089, 2017.

38. Labitzke, K., Austin, J., Butchart, N., Knight J., Takahashi, M., Nakamoto, M., Nagashima, T.,

 Dorothy, J., and Williams, V.: The global signal of the 11-year solar cycle in the stratosphere:

 Observations and model results, J. Atmos. Terr. Phys., 64, 203-210, doi:10.1016/S1364-

 6826(01)00084-0, 2002.

39. Lindsey, R.: Climate and Earth's Energy Budget. In NASA Earth Observatory, https://earthobservatory.nasa.gov/Features/EnergyBalance/ (last access: 25 April 2018),

2009.

40. Lindsey, R.: In Watching for El Niño and La Niña, NOAA Adapts to Global Warming, https://www.climate.gov/news-features/understanding-climate/watching-el-ni%C3%B1o- and-la-ni%C3%B1a-noaa-adapts-global-warming (last access: 25 April 2018), 2013.

41. Liu, F., Chai, J., Huang, G., Liu, J., and Chen, Z.: Modulation of decadal ENSO-like variation by effective solar radiation, Dynam. Atmos. Oceans, 72, 52-61, ISSN: 0377-0265, 2015.

42. Lockwood, M.: Solar change and climate: an update in the light of the current exceptional solar minimum, P. Roy. Soc. A-Math Phy., 466, 303–329, 2010.

43. Lockwood, M.: Reconstruction and prediction of variations in the open solar magnetic flux and interplanetary conditions, Living Rev. Sol. Phys., 10, 4, 2013.

44. Maleski, J. J. and Martinez, C. J. (2018), Coupled impacts of ENSO AMO and PDO on temperature and precipitation in the Alabama–Coosa–Tallapoosa and Apalachicola–

Chattahoochee–Flint river basins. Int. J. Climatol, 38: e717-e728. doi:10.1002/joc.5401

[revised manuscript text omitted]

---

## Author Comment (AC3) · 2 Apr 2019

**Franklin Isaac Ormaza-González and María Esther Espinoza-Celi**

formaza@espol.edu.ec

Dear reviewer, we much appreciate your time put on our manuscript and consider invaluable your comments made on it. We would like to make some notes to them with most respect and without intending any conflictive discussion.

Similar comments were done from the AR1. We have responded her/him and we deem much of the reply to AR1 will satisfy yours (hope so). Please see the attached document or you could please go directly to the reply to AC1. In any case, we are trying to respond to your comments.

[Figure]

We are sorry we have taken some time to reply you, but, as per your suggestion, we have revised the entire paper, we have reviewed and have rewritten many parts with the help of two well-known British scientists.

We will try to reply every point you mentioned, in the hope you re-consider your first revision.

1) You. In this paper the authors are exploring the relationship between sunspots and their impact on ENSO and PDO in the Pacific Ocean. It was difficult for me to find an answer to the title's question, "Do sun spots influence the onset of ENSO and PDO events in the Pacific Ocean?" It is difficult for me to tease out the main points of the paper because I find the structure unclear and the figures don't effectively summarize the main points either. Annotation of the figures could help address this.

We. We think the paper contributes to a better understanding of the ENSO (La Niña and El Niño) interannual events and also PDO and AMO decadal scales fluctuations. There have been many studies on how the SS could affect ENSO processes (please see introduction section), rain in Europe (Laurenz et al., 2019), SOI (Higginson et al., 2014), etc. Here we attempt to use as many variables as we can in different oceanic areas: Equatorial Central Pacific (area 3.4), Eastern equatorial Pacific (area 1+2), Northern hemisphere Pacific and Atlantic basins using 6 indexes. So far, forecasting models do not take into account the influence of sun spots number or any parameter that measures the solar energy heating the surface area. Perhaps, this is one of the reasons why sometimes they fail to predict El Niño (area 3.4) and its impact (in area 1+2 for example). Nonetheless what said, we do accept the paper has a weak and perhaps confusing writing, which has been amended by the edition and revision of two scientists from UK. All sections have been worked, specially abstract and results/discussion and conclusions. The manuscript is now stronger. Figures have been reduced to 10 and we are working on improving their captions.

2) You. There was clearly a lot of work done but I do not feel like it is presented in a
way that effectively supports their arguments and that correlations are insufficient for their claims, which remain a bit unclear. I recommend that this article be rejected as its structure is unclear and hard to read, main questions are not outlined clearly, and the presented statistics don't support the conclusions being made.

Thank you very much for recognizing the amount of work done, yes we run over hundreds of tests that included spectral analysis, non-linear regression analysis and multiple regression analysis was tried. It was a painstaking work, though very perfectible one. We did believe to show how SS correlate to these indexes and how they are consistent, our objective was that, it did not attempt to model SS and ENSO and PDO. We tried to find if there is any influence of SS on the ENSO and PDO indexes

Correlation analysis can be accepted to find any association between an independent (SS) and dependent (SST, Anomaly SST, ONI, MEI, PDO, AMO) variables. All variables are inter-annual and decadal (not climatological). These correlations are logical as the dependent variables are affected by the sun radiation, which in turn can be estimated by sun spot activity (see formula 1, line 55). Simply, the physical process is energy from the most important source (sun) being transferred to sea surface water and loosing energy through other processes (evaporation, upwelling, friction, for example). Fig. 1 shows the variation of ONI in terms of SS ascending phases of the six cycles (left panel) with a lag time of 24 months and per each cycle (right panel). The linear regression curves at SS close to zero shows that the ONI is somewhere between 0C to -2C, whilst in the range 50-200 sun spots the ONI is predominantly positive. Overall all ascendant phases together give an r2 (p<0.01) of 0.11, in some cycles (22 and 23) the r2 was 0.6. We deem this a clear evidence on how SS (read solar radiation) affect the studied indexes.   ——————— Fig. 1. Linear regression curves for all ascendants phases of cycles 19-24 (left panel) and per each cycle (right panel). Note. This figure is going to be part of the revised manuscript. ——————

In the present case we are not even attempting to say that dependent variables are only affected by SS. Not at all, but SS play its role, and this role sometimes is poor (low

r2) and sometimes is important (high r2). These indexes are also affected by other oceanographic and even anthropogenic variables (see Laurenz et al., 2019). Consistent higher correlations were found in area 3.4 compared to region 1+2; why? In 1+2 there are more intense dynamic processes affected by winds, interaction of different ocean water mass (north and south equatorial currents), transport of panama Bay heat content mediated by trade winds from the Atlantic, higher and variable cloudiness due to the geographical variability of the ITCZ (Intertropical Convergence Zone), Humboldt and Cromwell upwelling, etc. whilst in 3.4 it does not exist upwelling, cloudiness is less affected by land-atmospheric processes, there are not trade winds from the Atlantic, not upwelling in Panama Bay, etc. not Humboldt neither Cromwell, although variable SOI occurs. The outcome of the higher correlation values in 3.4 is logical as it was for other indexes though variable correlation. Note. The three last paragraph were taken from Reply to AC1.

3) You. The paper reads like a list of rËĘ2 values and Pearson correlation coefficients and is lacking a coherent narrative. Not every correlation needs to be typed out, they can be in a table or figure and referred to. In these long lists it is hard to identify the most important ones.

We mainly used r2, in a very few occasions used Pearson coefficients, perhaps we lack of good written narrative, but we think now it has been improved (see revised manuscript). Of many dozens r2 we tried to get the most relevant, even though we are aware we have written too many risking being tedious and repetitive, but we want the readers have chance to see much of the correlation found so they can judge. By choosing the best one, you could perhaps eventually become biased. Perhaps it was our naivety to have written a lot of r2 and their respective p-value. Respectfully, we do think report r2 and its respective p-value have to be done to show how strong (not casual) is the correlation.

4) You. Their argument hinges on these statistics but I do not believe that they can sufficiently support their claim that ENSO and PDO are driven by sun spots. This is

because correlation does not imply causation, slightly misleading from their title, and correlations can be artificially high due to a brief periods of in-phase activity. These caveats should be mentioned and can be remedied with longer time series.

Please see answer to comment 2, and also reply to AR1. Again respectfully, our titular question we think is answered without any misleading, and correlations cannot be considered artificial, it that was the case the slopes of linear regression should have been the same, are not. For these reason we decide to report most of slopes, r2 and p-value to demonstrate consistency through time, space and between indexes. Many other researchers have done so (e.g. Laurenz et al. 2019, please references).

We know, high correlation does not necessarily mean causation. That depends on the variables. If you for example correlates % of people that have access to internet in developing countries through the last 25-30 years (https://cacm.acm.org/magazines/2018/7/229046-bringing-the-internet-to-the-developing-world/fulltext), you will have a good r2, but you cannot say % of access is due to time. But if we measure photosynthesis rate (see e.g., http://biol14042013.blogspot.com/2013/02/factors-limiting-rate-of-photosynthesis.html) in a water body and irradiance (e.g. 600 nm wave length), probably there will be a good correlation (fig 2) of any regression analysis done. Then you and we could talk about causation, light is the independent variable and photosynthesis rate de dependent one. Laurenz et al. (2019) in their paper they talk that SS triggers rain events….   ———— Fig, 2. Rate of photosynthesis in terms of irradiance. Figure taken from http://biol14042013.blogspot.com/2013/02/factors-limiting-rate-of-photosynthesis.html. ————-

5) You. I do not think that the methods used can answer the titular question. There needs to be more interpretation and context with the rËĘ2 values, rather than listing them. More error and uncertainty discussion would also improve the paper.

We. The title opens the question about SS possible influence on ENSO and PDO

and the way to answer was legitimate use of linear regression analysis which allows to determine correlation and its p-value, thus respectfully we deem we answer the question. Yes, sure, there is room to improve interpretation, explanation and context. Please see lines (version attached to AR1): Every effort was made to explain the poor and high correlations in terms of physical and oceanographic processes. It can be seen in lines: 245-254, 231-237, 260-263, 268-273, 277-285, 300-310, 318-326, 333-336, 342-347, 353-374, 384-388, 399-403, 410-413 and so on

6) You. Therefore, as is I do not think these statistics can be used as a predictor for ENSO. This paper would require structural overhaul with clearly defined sections and goals.

We. Again respectfully. If the method is not valid for this paper; why should it be OK for previous accepted and reported studies. Every scientific method has shortcoming and limits. Recently, a paper correlating SS and rain over Europe has been published, and is reporting correlations factors similar with lag times in the range reported by us: ". . .. Taking into account cause and effect, it is suspected that increases in Central European rainfall are actually triggered by the solar minimum some 3–4 years before the rainfall month, rather than the lagging solar maximum. . ..." (see, Laurenz et al. 2019). Similarly, Higginson et al. (2004) reported SS association to SOI index; this index is part of ENSO, with similar length of data (see Fig. 1). We took as dependent variables, four indexes plus SST and anomaly SST from different oceans areas placed in the equatorial Pacific Ocean (from 170W to 82W) in which are regions areas 3.4 (5°North-5°S, 170-120°W) and 1+2 (0-10°S, 90°W-80°W). The first is the reference area for ONI and MEI indexes as well as SST and anomaly SST and it is an open area; whilst the second area is where much of what happens in 3.4 is reflected, but this one is close to coast. These four variables are inter-related and used to determine ENSO processes (El Niño, La Niña and neutral episodes), which are interannual, lasting 12-18 months. The AMO and PDO refer to SST behaviour of the North Pacific and Atlantic which are larger oceanic areas than 3.4 and 1+2. The relationship between and AMO and PDO

is widely accepted (as shown in the paper) and ENSO is associated somehow to PDO; thus, during cold phase PDO, La Niña events are more frequent and intense than El Niño. When PDO is on warm phase the contrary. The most intense and damaging El Niño (fully developed) occurred between 1980-2000, which is a period of warm phase PDO. Two most intense and prolonged La Niña happened during colds phase PDO (1954-1979) and 2001-present. The independent variable was the monthly SS from 1954 to 2017 represent 6 cycles. The SS is an accepted way to estimate Sun activity and "The Sun's activity cycle governs the radiation" (Bhowmik and Nandy, 2018), which in turn affects the heat content of the ocean surface and therefore the indexes above mentioned; in the section Introduction we explain thoroughly. We add, Higginson et al. (2004, paragraph 39) said "Our analyses of recent SOI fluctuations, El Niño frequency and intensity, suggest a coupling between the âĹij11‐year solar luminosity cycle and the SOI. Specifically, if we filter the SOI for El Niño (shaded gray) and La Niña (solid black bars) excursions, the more gradual quasi‐cyclic trend of the SOI is inversely correlated with Sunspot Index (SSI) with approximately 24 months lag". The SOI index is part of the ENSO, but we did not consider it, because it is highly volatile. Thus, Higginson et al. (2004), Laurenz et al., (2019) and others in the reference section are talking about cause-effect.

7) You. The figures should be annotated and streamlined to be more easily interpreted and more clearly support the main arguments. Additionally, the paper requires editing by a native English speaker, much of the science gets lost in the presentation. You are right dear reviewer. It has been done. Two figures have been compiled together and we have reduced from 12 to 10.

Finally, we are indeed grateful again for you review, your time to analyze and write the review, we know how busy you should be. Your comments have made us to improve our paper. Can we acknowledge your contribution, in the respective section? Again, we have been very respectful, if any expression caused an inconvenience, we apologize indeed.

Sincerely,

Franklin I. Ormaza-González and María Esther Espinoza-Celi,

  References. Bhowmik and Nandy. 2018. Prediction of the strength and timing of sunspot cycle 25 reveal decadal-scale space environmental conditions. NATURE COMMUNICATIONS | (2018) 9:5209. https://doi.org/10.1038/s41467-018-07690-0 Higginson, M. J., M. A. Altabet, L. Wincze, T. D. Herbert, and D. W. Murray (2004), A solar (irradiance) trigger formillennial-scale abrupt changes in the southwest monsoon?Paleoceanography,19, PA3015, doi:10.1029/2004PA001031 Laurenz, L., H.-J. Lüdecke, S. Lüning (2019): Influence of solar activity on European rainfall. J. Atmospheric and Solar-Terrestrial Physics, 185: 29-42, doi: 10.1016/j.jastp.2019.01.012

Please also note the supplement to this comment:
https://www.ocean-sci-discuss.net/os-2018-125/os-2018-125-AC3-supplement.pdf

––––––––––––––––––––

Interactive comment on "Do sun spots influence the onset of ENSO and PDO events in the Pacific
Ocean?" by Franklin Isaac Ormaza-González and María Esther Espinoza-Celi
Reply to Anonymous Referee #2
April 2, 2019

**Fig. 1.** Linear regression curves for all ascendants phases of cycles 19-24 (left panel) and per each
cycle (right panel).

[Figure]

Note. This figure is going to be part of the revised manuscript.

**Fig. 1.**
http://biol14042013.blogspot.com/2013/02/factors-limiting-rate-of-photosynthesis.html.

[Figure]

[Figure]

**Fig. 2.**

**Supplement:**

**Do sun spots influence the onset of ENSO and PDO events in the Pacific Ocean?**

*Franklin Isaac Ormaza-González[1], and María Esther Espinoza-Celi[1]*

1)  ESPOL Polytechnic University, Escuela Superior Politécnica del Litoral, ESPOL, (Facultad de
Ingeniería Marítima, Ciencias Biológicas, Oceánicas y Recursos Naturales), Campus Gustavo
Galindo Km. 30.5 Vía Perimetral, P.O. Box 09-01-5863, Guayaquil, Ecuador

Corresponding author: formaza@espol.edu.ec.

The sea surface temperature (SST), SST anomalies, ONI (Oceanographic El Niño Index) and MEI
(Multivariate ENSO Index) in regions El Niño 1+2 (80°W-90°W, 0°-10°S) and 3.4 (5°N-5°S, 170°W-
120°W) as well as the Pacific Decadal Oscillation (PDO) and Atlantic Multidecadal Oscillation (AMO)
indexes, were correlated to sun spots number (SS) from cycles (SS#) 19 to 24 (1954-2017).
Polynomial regression functions represented each of the six cycles with an average $r^2 > 0.89$
(p<0.001). Series of correlations between SS and chosen indices at different lag times (0, 6, 12, 24,
36 and 48 months) gave a response time of between 12 and 36 months. Over the entire 1954-2017
period, the SS cycles did not show a strong correlation with the variables or SST Anomaly in the El
Niño areas 1+2 and 3.4. It seems that high and low SS balanced through the cycles. Improved
correlations were found for the shorter period 1990-2016. The SST correlations against individual SS
cycles in regions 3.4 and 1+2 were up to 0.219 (SS#23) and <0.0675 (SS#19) correspondingly. SST
Anomaly, ONI and MEI correlated with $r^2$ of 0.250, 0.3943 and 0.2510, one-to-one; the lag time was
24-48 months and linear curves had positive slope. In general, more inconsistent and lower
correlations were found in 1+2 than in 3.4. On longer time scale indexes, the PDO (as well as the
AMO) seemed to respond in 36-48 months to SS cycles ($r^2$ of 0.625, SS#19) and 0.766, SS#24); whilst
the AMO index gave a slightly lower correlation (0.490, SS# 20) with a similar lag time. Further
analysis of SS numbers and the oceanic indices above during the ascending and descending phases
of each cycle showed SST was best correlated with the ascending phase (up to $r^2$ of 0.870, with a lag
time between SS cycle and index of about 36 months) and this trend also applied to the SST
anomaly, although with slightly poorer correlations.  The highest $r^2$ values coincided with strong
ENSO events. The descending phase showed lower correlations between SS and ocean indices
including MEI and ONI. The PDO was linearly correlated to SS ($r^2$ 0.7677 to 0.2855 (12 to 24 months)
as was the AMO ($r^2$ up to 0.700) whilst during descending phases correlations were poorer. The SS
activity seemed to have a better correlation during the cold phase of PDO. AMO and PDO proved to
be correlated to SS, with the PDO having a higher correlation. Presumably, as the North Pacific
Ocean basin has a larger area than the North Atlantic. The ascending and descending phases of SS
could re-inforce or weaken these indexes. The regression linear curves for ascending and descending
phases, shows high tendency that ONI, MEI and PDO being negative when SS number are in the
range 0-50, whilst the highest positive values are somewhere between 100-200 sun spots per
month. These results show that warm events tend to occur in the ascending phase, or at the top of
the cycle (as solar radiation increases), and have a delay time of around 24-36 months, whilst cold
events are associated with descending phases but with a shorter lag time, 12 months. The
correlation analysis given here indicates sun spot activity should be considered as a factor that could
condition and trigger low (PDO and AMO) and high (ONI-El Niño) frequency oceanographic events in
the Pacific and Atlantic Oceans.

**Key words:** Sun spots cycles, SST, SSTA, ONI, MEI, PDO, AMO, El Niño, La Niña

**Introduction.**

Essentially, the only external source of energy to Earth is the sun, which constantly radiates a flux of energy to the upper atmosphere (the solar constant) of 1360 W m$^{-2}$ or 1.36 kJ m$^{-2}$ s$^{-1}$ (Monteith,

1972) or 1.92 ly day$^{-1}$ (Ormaza-González and Sanchez, 1983). Recently, Kopp and Lean (2011) have reported that the most accurate accepted solar constant value is 1360.8 ± 0.5 W m$^{-2}$. Of this flux of energy, 75-50 % reaches the Earth's surface (Ormaza-González and Sanchez, 1983; Lindsey, 2009)

and the remainder is reflected and/or absorbed by clouds, particles, gases, etc. (Horning et al.,

2003). About 90-93% of that energy reaching the surface is accumulated in the oceans (Trenberth et al., 2014; Clutz, 2017). The solar constant is affected by variations in sun spot (SS) number (Bhowmik and Nandy, 2018) and other solar activity parameters by around 0.1%, i.e. on the order of 1.361 W

m$^{-2}$. The Hale cycle (around 11 years) is characterized by increasing and then decreasing SS numbers (Hathaway, 2015). Froelich (2013) suggested that the solar constant can vary by up to 4.0 W m$^{-2}$ over two SS cycles, i.e. a 22-year cycle, and proposed a simple relationship between SS and the solar constant (SI), by assuming a direct relationship between the two

*SI= 1353.6 + 0.089 (SS)*                    ($r^2$ of 0.71, 95-99% confidence).

The surface-subsurface layers of the ocean that interact with the lower atmosphere alternately release and absorb heat energy. The work of Zhou and Tung (2010) reported the impact of the SI on global SST over 150 years, finding signals of cooling and warming SSTs at the valley and peak of the

SS cycles. Schlesinger and Ramankutty (1994) reported a global cycle of 65-70 years that is possibly affected by greenhouse anthropogenic gases, sulphate aerosols and/or El Niño events, but they did not imply an external force such as the SS. There are well known oceanic events that are roughly periodic with low (25-30 years) or high (3-5 years) frequencies. These include the Pacific Decadal

Oscillation (PDO, Mantua et al., 1997; Mantua and Hare, 2002; Zhang et al., 1997; Yim et al., 2013),

Atlantic Multidecadal Oscillation (AMO, Enfield et al., 2001; Condron et al., 2005; Gray et al., 2010)

and Interdecadal Pacific Oscillation (IPO, Henley et al., 2015), as well as El Niño (Busalacchi et al.,

1983, see **COAPS Library's: http://www.coaps.fsu.edu/lib/biblio/coaps-a.html**)  or La Niña (Yuan and Yan, 2012). During El Niño events, the surface and subsurface lose energy to the atmosphere and the opposite occurs during La Niña (Trenberth et al. 2014, Fasullo and Nerem, 2016); these events have a periodicity of 2-7 years while the decadal processes may take 25-30 years. The

Interdecadal oscillations have a series of impacts; e.g., the PDO gives rise to teleconnections between the tropic and mid-latitudes (Yoon and Yeh, 2010), and the effects include: 1) the ocean heat content (Wang et al., 2017), 2) the lower and higher levels of the trophic chain and small pelagic fisheries including tuna and sardines (Ormaza et al., 2016a, 2016b), 3) biogeochemical air-sea

$CO_2$ fluxes (McKinley et al., 2006), 4) the frequency of la Niña/El Niño (Newman et al. 2003). The interactions between decadal oscillations PDO/IPO and AMO may affect also ocean heat content (Chen and Tung, 2014).  All these low and high frequency oceanographic events have a direct impact on local, regional and global climate patterns, and there is growing evidence that the driving source of energy is the sun (Grey et al., 2010). Thus, Huo and Xiao (2016) have found a positive strong correlation between El Niño 2015-2016 and SS, as well as SS and the El Niño Modoki index. White et al., (1997) reported that heat anomalies produced by variable solar irradiance are stored in the upper ocean layer, driving SST changes of 0.01-0.03 K and 0.02-0.05 K on decadal and interdecadal periods respectively. Zong et al. (2014) in their review of the impact of the 11-year SS cycle and multidecadal climate projections, have found global SST variations of 0.08 ± 0.06 K and 0.14 ± 0.02

during the 11 and 22 year Hale Cycle, combined with a response lag of 1-2 years in relation to the SS

(see also, Kristoufek, 2017). Liu et al. (2015) have reported that effective solar radiation plays a role in the modulation of decadal ENSO (El Niño and the Southern Oscillation) oscillation. More recently,

Yamakawa et al. (2016) have reported that solar activities in terms of SS numbers not only affect the troposphere but also the sea surface, even though SS abundance is only a partial measure of solar activity (Scafetta, 2014).   The work reported here investigates how sun spots may affect low and high frequency oceanic events such as the Pacific Interdecadal Oscillation (PDO), the Atlantic multidecadal oscillation (AMO), anomalous sea surface temperatures, and El Niño and La Niña events.

**Material and methods.**

Data for monthly sun spot number (SS) were taken from the Royal Observatory of Belgium, Brussels,

World Data Center SILSO (http://www.sidc.be/silso/datafiles). Data sources for other variables were as follow:  El Niño regions areas 3.4 (5°North-5°S, 170-120°W) and 1+2 (0-10°S, 90°W-80°W):

- **Sea surface temperatures (SST) and SST Anomaly**: The Monthly Extended Reconstructed
      Sea Surface Temperature Version 4 (ERSSTv4, 1981-2010 base period). The Optimum
      Interpolation 1/4 Degree Daily Sea Surface Temperature (OISST.v2, 1981-2010 base period),
      http://www.cpc.ncep.noaa.gov/data/indices/.

- **Oceanic Niño Index** (ONI: Huang et al., 2014): ERSST.v4  for El Niño/La Niña events since
      1950 till December 2017:
      http://www.cpc.ncep.noaa.gov/products/analysis_monitoring/ensostuff/ensoyears.shtml.

- **Multivariate ENSO index** (MEI: Wolter and Timlin, 2011):
      https://www.esrl.noaa.gov/psd/enso/mei/table.html.

- **Pacific Decadal Oscillation** (PDO, based on Mantua Index): The PDO index is based on
      NOAA's extended reconstruction of SSTs (ERSST Version 4). It is constructed by regressing
      the ERSST Anomaly against the Mantua PDO index for their overlap period, to compute a
      PDO regression map for the North Pacific ERSST Anomaly. The ERSST Anomaly are then
      projected onto that map to compute the NCEI index. The PDO index closely follows the
      Mantua PDO index at: https://www.ncdc.noaa.gov/teleconnections/pdo/ (Wolter and Timlin
      1993, 1998 and 2011).

- **Atlantic Multidecadal Oscillation** index:
      https://www.esrl.noaa.gov/psd/data/timeseries/AMO/.

All indexes have data from April 1954 to December 2017. The analysis was done using Excel and/or R statistical tools. The correlation exercises were executed using SS solar cycles as complete time series against SST Anomaly (in El Niño regions 3.4 and 1+2), ONI, MEI, AMO and

PDO indexes. Correlations with lags of 0 to 48 months were carried out. For the SS cycles 19-23

and their impact on the above mentioned dependent variables, correlations were carried out for the whole time series (1954-2017), and for 1990-2016, for each cycle and for their respective ascending and descending phases. Spectral analysis and polynomial regression fitting curves were determined to obtain the slope of the ascending phases; the slopes were correlated to the oceanographic indexes.

**Results and discussion:**

The time series (1954 to 2016) of SS, PDO, AMO, ONI and MEI are shown in Fig. 1. The PDO, AMO, ONI and MEI cycles have been offset by 0, 12, 24 and 36 months (panels a, b, c and d respectively), whilst the SS series starts at t=0 in the four panels. It has been reported that the lag times for responses of some indexes to SS cycles (SS#) are around 12-36 months (e.g., Zhao et al., 2014). From 1954 to the present time each cycle 19 to 24 has occurred with a period of around 11 years (Hathaway, 2015), which is slightly less than the 11.2 years reported by Dicker (1978). The highest SS activity is seen in cycle 19 with around 250 SS/month, followed by <150, and at cycle 21 around 200, before decreasing steadily over cycles 22 to 24 to just over 100 SS/month. Cycle 24 is the lowest contemporary value of SS activity that is comparable only to cycles 12-15 (around 1880-1930) and is the lowest in the last 200 years (Clette et al., 2014). The negative or cold PDO phases (1947-1976, 2000-June/2016) are within SS cycles 19-20 and 23-24, whilst cycles 21 and 22 are within the positive or warm phase of the PDO (1977-1999). As the PDO and AMO indexes are displaced from 0 to 36 months on the time scale (Fig. 1), some peaks and troughs relative to SS activity can be seen. These are at ascending and descending parts of the SS cycles, e.g. during cycles 19-20 and 23-24 PDO indexes are basically negative, whilst during 21-22 they are more positive; an exception is around 1990, where there is a strong negative peak. However, AMO phases seem to be in opposition to and overlapping the SS cycles; a cold phase of AMO was between the 1960s and 1990s, whilst the warm phase is from the 1990s to the present (McCarthy and Haigh, 2015).

The ONI and MEI curves, both indicators of ENSO events, behave similarly throughout the study period (April 1954 – December 2017). However, MEI has the highest anomaly peaks (> 2) when compared to ONI. In general, ONI and MEI curves indicate the highest positive anomalies between 1978 and 1995, a period that coincides with the warm and cold phases of PDO and AMO respectively (see Maleski and Martinez, 2018). The opposite trend occurs before and after this period due to the inversion of phases. In addition, the highest peaks of both indexes only occur during the ascending and descending phases of the solar cycles; that is, they never coincide directly with the maximum period of sunspots in the cycles, except in 1959. The two highest MEI peaks occur during the descending phase of solar cycle 21 and ascending phase of solar cycle 23. In mid-2016  both indexes increased reaching the third highest peak of this period during the descending phase of

SS#24. Negative peaks of these indexes occurred either in the high or low plateaus of the SS curves.

The number (N) of data in the analysis were: 765 (1965-2017); 312 (1990-2016); 108 (1990-1999);

192 (2000-2016). For individual cycles 19 to 24: 127, 141, 124, 117, 141 and 120 respectively. In the same order for ascending-descending phases: 48-80, 50-92, 43-82, 33-85, 51-51 and 74-47.  The degrees of freedom of residuals were N-2. The degree of correlation in terms of Pearson coefficient is referred to as: high, moderate and low when the coefficient is between ±0.5 and ±1.0, ±0.3 to

±0.49 and less than ±0.29 respectively (http://www.statisticssolutions.com/pearsons-correlation- coefficient/). All linear regression residuals were auto correlated using the Durbin-Watson (DW) test (Montgomery et al., 2001) for 1954-2017, 1990-2016, 1990-1999, 2000-2016, individual cycles, and ascending-descending phases. The DW test results for the long time series averaged 0.122, for individual cycles it varied from 0.10 to 0.63 with an average of 0.18, and for the ascending and descending phases it averaged 0.40 and varied from 0.1 to 2.24.  The SST Anomaly in region ENSO

1+2 has the lowest and PDO the highest. Every given caveat associated with correlations was taking into consideration.

**Whole series (1954-2017) correlations.** All variables (Table 1) were correlated on a linear and polynomial (n= 2 to 6th order) basis using different lag times (0, 6, 12, 24 and 48 months) over the six SS cycles. Polynomial correlation (not shown) as well as linear ones displayed poor correlation coefficients, with the highest linear $r^2$ (p≤0.01) coefficients occurring at lag times between 12 and 24

months, and 36-48 months for the AMO. For SST and SST Anomaly in ENSO areas 1+2 there was no correlation. These results are like those of Kristoufek (2017), who suggested a surface thermal response of around 24-36 months. The highest correlation $r^2$ values with SS were up to: 0.043, 0.029,

0.040 and 0.021 for PDO, MEI, ONI and SST Anomaly (in 3.4) respectively, indicating there is a correlation with high confidence (p-value ≤0.01), though small $r^2$. This fact could reflect sun activity (sun spots) in the long term being balanced by the ups and downs of the cycles. This exercise suggests there is not a high correlation of these indexes in the Pacific and Atlantic over the studied time scale.  Nonetheless, on longer time scales, where SS cycles are affected by other sun internal processes, e.g., the hypothesized Minimum of Maunder (Eddy, 1976, Shindell et al., 2001, Ineson et al., 2015, Mörner, 2015, etc.), there can be an impact on a global and regional basis. Recently,

Lockwood (2010 and 2013) has reported that a grand solar minimum is coming as the SS cycle 24 is developing. There has not been a solar activity decline such as that found in SS# 23 to 24 over the last 9300 years, and such a solar minimum may last through cycles 24, 25 and 26 (Hady, 2013).

Under these circumstances where anomalous conditions appear to be developing, it was decided to analyze correlations using individual cycles in the range 19 to 24.

**Period 1990-2016.** Further analysis was carried out for the period 1990 to 2016, that includes cycles

22, 23 and 24. The time series was also split into 1990-1999 and 2000-2016, because during 1990-

1999 a strong (1991-1992), moderate (1994-1995) and the strongest El Niño (1997-1998) of the twentieth century occurred. On the other hand, in 2000-2016 (cold phase PDO) there were strong La

Niña events (2000-2002 and 2010-2012) and an El Niño Modoki event in 2015 (Huo and Xiao, 2016).

Figure 2 shows again a poor correlation (<0.011, p>0.246), for the SST Anomaly in region 1+2 (blue bars), although this region was gravely affected by the strong El Niño in 1997-1998 which brought hundreds of casualties and losses of billions of US dollars to the Ecuadorian infrastructure (Glantz,

2001). The linear correlation $r^2$ of SST in 3.4 (red bars) was around 0.1193 (p≤ 0.00001) over the whole period, whilst it was somewhat higher in the period 1990-1999 (0.1519, p≤ 0.01). The ONI

(green bars) correlation coefficient was up to 0.1436 (p≤0.02) when compared to SS in the period

1990-2000, where high positive SST anomalies were present for almost 6 years, and the ONI

correlated better than SST anomaly with SS in 3.4. The Pacific Decadal Oscillation (Fig. 2., grey bars)

had an $r^2$ of 0.276 (p<0.0001), in the period 2000-2016 (PDO in a cold phase), with a Pearson correlation of 0.523 that can be considered as high (https://www.statisticssolutions.com/pearsons- correlation-coefficient/). However, for the period 1990-1999 it was 0.239 and for the whole period was 0.402; i.e. a poor and fair degree of correlation respectively.

**Individual Cycles.** Correlation analysis was split into SS cycles from 19 to 24. The SS and SST $r^2$

coefficients indicated a poor correlation and confidence level (p≥0.05) in region 1+2 in all cycles (Fig.

3); most of the correlation $r^2$ values were <0.050, except in cycle 19, ($r^2$ of 0.0675, p=0.0032); in cycles 21 and 23 the highest $r^2$ was 0.046 (p=0.0173) and 0.048 (p=0.037) respectively. In general, the lag time varied between 6 to 36 months in region 3.4, whose correlations $r^2$ were up to 0.219

(p≤0.0001) and 0.213 (p≤0.0001) for cycles 23 and 19 respectively with a lag time of 12-36 months.

Cycles 20 and 22 had $r^2$ of 0.105 (p≤0.0001) and 0.074 (p=0.003) respectively. The slopes of the linear regression curves with the highest $r^2$ were positive in region 3.4, indicating a direct correlation between SST and SS cycles. However, cycles 22 and 23 in the region 1+2 exhibited inverse correlation (Fig. 3). Further polynomial correlation (n=2 to 6) analysis did not provide a better $r^2$.

Over all, higher correlations were found in El Niño regions 3.4 than in 1+2.

**Anomalies in SST.** The magnitude of the SST Anomaly can change depending on the reference used; there are 5 versions of ERRS (Huang et al., 2017). Currently version 5 tends to be used in El Niño studies. Here we used the ERRSv4 (Huang et al., 2014); Huang et al. (2017) stated that there is not a noticeable difference between ERRSv4 and ERRSv5. The anomalies of SST in 3.4 and 1+2 were also correlated against every SS cycle; correlation $r^2$ values were not better than 0.396 (p≤0.0001) in both regions, with higher variability in 1+2 than 3.4 both in response time and correlation coefficient (Fig.

4). In region 3.4, the highest correlations were 0.289 (p≤0.0001) and 0.270 (p≤0.0001) during cycles

19 and 23 respectively, with a lag time of between 12 and 36 months, with both occurring during cold phases of PDO (1955-1978, and 2000-present). Surface winds plus other oceanographic variables (e.g. upwelling) could play an important role in this high variability. Winds are not only generated in the local area but farther away, including the trade winds of the western Atlantic (Ormaza-González and Cedeño, 2017). Also, ENSO processes in the western Pacific could add variability in the SST Anomaly. The slopes of the linear correlation were basically positive for 3.4 and negative for 1+2, and for SST correlations for cycles 19, 23 and 24 (cold PDO phase) had the highest

$r^2$. Again, the anomalies in 3.4 were better correlated than in region 1+2.

**ONI.** The El Niño index (Fig. 5) displayed $r^2$ values when correlated with SS activity from around

0.053 (p=0.01, SS#22) up to 0. 25 (p<< 0.0001, SS#24); there were poor to fair correlations with a positive slope in SS cycles 19, 23 and 24. During SS#24, ONI reached extreme values of 2.6C (Nov-

Dec-Jan 2015/2016) and -1.7C (Oct-Nov-Dec 2010). The highest $r^2$ were again found with a 24-48

months lag time. Cycle 21 did not show any significant correlation with ONI; however, cycles 20, 22

and 24 had $r^2$ values of 0.144 (p<<0.001), 0.131 (p<<0.0001) and 0.252 (p<<0.0001) and lag times of

48, 12, and 24 months respectively. Recently, Huo and Xiao (2016) found strong correlation between

SS and El Niño Modoki during 2015 (SS#24). The variability in correlations  could arise from: 1) SS

numbers showing large variations from one month to another, 2) regional meteorological conditions (particularly cloudiness), ocean surface currents that exchanges heat in region 3.4, 3) Kelvin waves (Gill, 1982), 4) the Southern Oscillation Index (SOI: Southern Oscillation Index:

http://www.cpc.ncep.noaa.gov/data/indices/soi).   All these may affect the SSTs and together with the way ONI is obtained, as the ONI has a variable reference period of 30 years; thus for 1950 to

1955 the reference period is 1936-1965; for 1956-1960; 1941-1970.  The ERRSv4 uses the period

1981-2010.  The reference period is changed every 5 years (Lindsey, 2013); the most recent ONIs (v4/v5) are supposed to use better and more consistent information as data acquisition improves.

**MEI.**  This additional index for El Niño events had a lower correlation ($r^2$) with SS; thus, the highest value was 0.3943 (p<<0.00101, SS#19), the next 0.3028 (p<<0.00001, SS#24), 0.2421 (p<<0.00001,

SS#23) and 0.1566 (p<<0.0001, SS#20); in cycles 21 and 22 no correlation better than 0.1232

(p<<0.0001) was found. The lag time for sun spot activity (with the highest correlations) ranges from

24-48 months; and linear regression curves were with mainly positive slopes. The lower correlation found could be explained as this index comprehend six variables, and some of these could not be directly or are weakly related to SS; like zonal and meridional components of surface wind, surface air temperature, cloudiness (Wolter and Timlin, 1993).

**PDO.** This interdecadal index (Fig. 5) is linearly correlated to SS cycles with a lag time between 36

and 48 months, with the highest $r^2$ of 0.391 (p<<0.00001) in cycle 19, and 0.586 (p<<0.00001) for cycle 24. Both cycles are within the cold phase of the PDO. The next highest $r^2$ with p-values

<<0.0001 were 0.218, 0.1361, 0.218 and 0.260 for cycles 20-23. In all cycles, the highest $r^2$ were directly correlated, except cycle 20. For some reason, there appears a better fit with both PDO and

ONI in cycles 19 and 24, which are within the cold phase of the PDO, even though these cycles have remarkably different shapes and peaks (Fig. 10).  Cycle 19 registered SS counts of over 250 whilst cycle 24 was just around 100; also, the peaks were different being respectively very sharp and extended. The direct relationship between PDO and ONI has been reported extensively (e.g.

Ormaza-Gonzalez et al., 2016a, Jia and Ge, 2017).

**AMO.** This index gave correlation coefficients ($r^2$) with SS numbers of up to 0.490 (p<<0.00001) and down to 0.162 (p=0.0004) in cycles 20 and 24 respectively, when a lag time of 48 months was used.

With cycles 19, 21 and 22 the best fit elapsed time was 24-36 months (Fig. 5). Gray et al. (2016)

reported lag time responses of mean-sea-level pressure over the Atlantic to SS cycles of 36-48

months over a longer time series study of 32 solar cycles. The linear regression analysis rendered positive and negative slopes for cycles 19, 23 and 24; and 20 to 22, respectively. This coincides with the phases of the AMO, negative from around 1965 to 1998 (SS cycles 20-22), and positive; 1930-

1965 (SS cycle 19) and after 1998 (SS cycles 23-24), http://appinsys.com/globalwarming/amo.htm . It is noteworthy that the slopes of the PDO and AMO linear regressions are negative and positive respectively for cycles 21 and 22, but in concordance in cycles 19, 20, 23 and 24 (cold phase PDO).

**Ascending and descending phases** of solar cycles. As the SS cycles are best related to variables studied on a response time from 24 to 36 months, there was the need to study their influence during the ascending and descending phases, which have roughly 5-6 years duration. Polynomial regression analysis was performed to establish a function that could best describe every SS cycle.  Sixth-order polynomial curves (Fig. 10) were found to render a very strong correlation coefficient averaging 0.89

(p≤0.001). These functions allowed the analysis of correlations in the ascending and descending phases.

**SST in 1+2 and 3.4.** In region 1+2, the highest correlation coefficient $r^2$ were: 0.205 (SS#23), 0.189

(SS#21), and 0.163 (SS#19). All linear regression coefficients $r^2$ over 0.0847 (p<0.05 to =0.0008)

occurred in the ascending phase of the SS cycles, whilst those in the descending phase were lower, with no definite lag time pattern from 0 to 36 months. The slope (positive/negative) of the linear regression (Fig. 6) curves showed no pattern. These low and variable $r^2$ values reflect region 1+2

being subjected to the combined impact of many diurnal and seasonal oceanographic and meteorological variables. For example, during the first quarter of 2017 (cycle 24), in 1+2 there was a higher than usual SST because the southern trade winds in the eastern Pacific weakened whilst those in the North Atlantic strengthened.  These winds passed through the Panama Isthmus and blew warm (up to 30C) surface waters from the Panama Bay southwards towards area 1+2, thus provoking a rapid and relatively short lived surface warming event (Ormaza-González and Cedeño,

2017), whilst region 3.4 was registering La Niña conditions. Even though, this cold event also strengthens the Cromwell Undercurrent (Knauss, 1959) and Humboldt (Montecino and Lange, 2009)

currents that impact upwelling processes in 1+2, the surface warm event was not supressed;

Ramírez and Briones (2017) called this event El Niño Costero. All these factors would mask the SS

signal in area 1+2. Nonetheless, it could be observed that during the ascending phases of the cycles, the correlation of SSTs was higher than in the descending phase of cycles

In the region 3.4, the maximum $r^2$ of SST in each SS cycle was found at a lag time of 36 months with all of them occurring at the ascending phase, except in cycles 20 and 24. The four highest $r^2$ values were 0.870 (p=0.021, SS#24), 0.613 (<<0.0001, SS#22), 0.574 (p<<0.0001, SS#19), and 0.556

(p<<0.0001, SS#23) with Pearson coefficients of 0.9327, 0.7803, 0.7576 and 0.7456, respectively, thus showing a strong degree of linear correlation. Linear regression slopes were variable (Fig. 6), although there was a tendency in cycles 20, 21 and 22 (warm PDO) for negative slopes and for positive slopes for cycles 19, 23 and 24 (cold phase PDO). In area 3.4 the SST response to SS was much clearer than for 1+2, as in this region (10N-10S and 120W-180W) there is no influence of coastal processes. The highest $r^2$ (0.870, p=0.021; lag time 36 months) in the descending phase in cycle 24 coincided with the strongest El Niño, and the second-highest $r^2$ (0.613, p<<0.00001) during ascending phase of cycle 22 with two consecutive strong El Niño 1991-1995; the third $r^2$ (0.574, p<<0.00001) during cycle 19, with el Niño 1955-1957, and finally the fourth $r^2$ (0.556, p<<0.00001)

with the 1997-1998 warm event during cycle 23. It seems that over the relatively short time scales of

SS cycles, either on their initial ascending or subsequent descending phases, impacts on the SSTs can be triggered.

**SST Anomaly.** In region 1+2 (Fig. 7), the anomalies registered high $r^2$ (p<<0.0001) of 0.662 (SS#22),

0.637 (SS#19), 0.523 (SS#21), 0.480 (SS#23), 0.359 (SS#24); and 0.254 (p=0.0002, SS#20)

respectively, in the ascending phase of the SS cycles and with a positive linear regression slope (except SS#23). The response lag time was somewhere between 0 and 48 months. On the other hand, the descending phase showed a predominantly lower $r^2$, less than 0.14 with lower significance (p ≤ 0.02), with the exception in SS# 19, 0.304 (p<<0.0001). The results suggest that during cold phase PDOs when Northern Pacific basin surface ocean waters are relatively colder, the correlations in this area tend to be higher, as the increasing sun radiation augments the heat content (i.e., SST) of the ocean surface.

In region 3.4, there was a high and consistent $r^2$ (Fig. 7) that reached up to 0.897 (p=0.014; SS#24);

0.863 (p<<0.0001; SS#22); 0.665 (p<<0.0001; SS#19), 0.826 (p<<0.0001; SS#23), then fell to 0.211

(p=0.008; SS# 20); 0.239 (p=0.0009; SS# 21) respectively; all of them were in the ascending phase except cycles 20 and 24. The lag time was consistent at 36 months. Linear regression slopes were variable (Fig. 8) with negative slopes in cycles 20-22 (warm phase PDO), and positive slopes in 19, 23

and 24 cycles (cold phase PDO). The highest $r^2$ of 0.897 at the start of the descending phase in 24

coincided with one of the strongest El Niño and the second $r^2$ (SS#22 ascending phase) with two consecutive strong El Niño 1991-1995. The third and fourth highest $r^2$ were during El Niño 1955-1957

and 1997-1998 warm event (SS#23 ascending) respectively. The results suggest that SS cycles are strongly correlated to SST Anomalies in both El Niño regions, with the strongest relationship in 3.4.

**The ONI index.** This index as well as SST and its anomalies in 3.4, were equally strongly associated with the ascending phase of the SS cycles (Fig. 8), with lag times of 24-36 months.  The highest correlation $r^2$ for each cycle were in the ascending phase, the predominant linear regression slopes were positive, except for SS#20. The highest $r^2$ (p<<0.0001) were: 0.817 (SS#22), 0.693 (SS#19),

0.637 (SS#23), 0.3547 (SS#24), 0.2876 (SS#20); 0.1936 (p=0.003, SS#21). The three highest $r^2$ match the dates of full-fledged strong El Niño 1955-1957, 1987-1989, and 1997-1998 (Fig. 9) with positive slopes on the ascending phase. In the descending phase the $r^2$ (p<<0.0001) in cycles 24, 23, 22 and

20 were 0.366, 0.284, 0.255, and 0.242 respectively. All had a lag time 0-12 months and positive slopes. Cycles 19 and 21 showed neither strong correlation (<0.1) or confidence values (p=0.2). The

ONI correlations are in accordance to what found with SST anomalies in 3.4.

Warm events tend to occur in both ascending/descending phases after the peak/trough, and have a delay time of 24-36 months, which is similar to findings of Huo and Xiao (2016). The delay time, is probably due to the slow accumulation of solar heat over time in surface oceanic waters. The descending phase of the cycles (Fig. 8), with a generally smaller slope than the ascending phase, produces a quicker response (0-12 months) to the ocean surface SST and ONI that could trigger neutral or cold events more cogently.  Most of the La Niña events occur during the descending phase or approaching the cycle minimum (Fig. 10), when the solar irradiance (SI) decreases slightly as does the number of sunspots. The weakest sunspot cycle (SS#24) has had three La Niña events: 2007-

2009, 2010-2012, 2016-2017 (Fig. 10). A plausible reason is that during this cycle the number of sun spots (i.e. sun activity) is the lowest in the last two centennials (Clette et al., 2014); therefore, less energy has hit the ocean surface allowing a cooling effect. Two important exceptions are La Niña

1988-1989 (22) and 2000-2002 (cycle 23) that occurred in the ascending phase.

**The MEI index.** The Multivariate ENSO Index does not only consider the SST Anomaly but also sea- level pressure (Allan and Ansell, 2006) and other variables. These variables include surface winds (meridional and zonal), surface air temperature and cloudiness (Wolter and Timlin, 1998). The MEI

correlated at slightly lower levels than ONI to SS cycles with $r^2$: 0.784 (p ≤ 0.0001), 0.770

(p<<0.0001), 0.5972 (p ≤ 0.0001); 0.3396 (p ≤ 0.0001); 0.2368 (p=0.0003); and 0.222 (p=0.001) for SS

cycles 19, 22, 23, 24, 20, and 21, respectively.  All of them were in the ascending phase of the cycles with lag times from 12 to 48 months (except cycles 23 and 24), with a positive linear regression slope. Exceptions were 22 and 20 where the $r^2$ was largest with a zero lag time. During the descending phase, as with the ONI, the $r^2$ were lower: 0.321 (p=0.0004, SS#24); 0.3145 (p<<0.0001,

SS#19); 0.2234 (p<<0.0001, SS#22); 0.2088 (p<<0.0001, SS#20), and 0.1438 (p=0.0002, SS#23) with positive slopes (except SS# 20), and lag times were in a larger 0-48 months range; cycle 21 did not have a $r^2$ above 0.010 (p>0.02). The much lower $r^2$, during descending phases, when there is less sun radiation energy (see formula 1), could lead to La Niña events, which has actually occurred over the six cycles. Also, the lower correlations could be because the MEI uses five variables more than the

ONI, and these could thus help obscure the signal from the sun's irradiation.

**PDO.** The Pacific Decadal oscillation gave positive slope linear correlations with SS in most cycles except cycles 20 and 21 (Fig. 9). Correlation coefficients of 0.7677 (p ≤ $10^{-12}$), 06577 (p ≤ $10^{-12}$),

0.6734 (p ≤ $10^{-7}$) and 0.5062 (p ≤ $10^{-7}$) for the ascending phase SS#19 (Apr/54-Nov/58), SS#24

(Jan/08-Feb/14), SS#22 (Sep/86-Jan/89) and SS#23 (May/90-Jun/00) respectively were found. All these coefficients were obtained at a lag time of 12-48 months, except 22 and 23 (lag time = 0). The slopes of the linear regressions were mainly positive during cold phase PDOs (cycles 19, 23 and 24), except cycle 20 when a cold PDO was transitioning to a warm PDO (cycles 21 and 22). The two highest $r^2$ values are at a lag time of 12-36 months, for cycles 19 and 24, as has been reported (e.g.,

Huo and Xiao, 2017). During the descending phase, the correlation $r^2$ tended to be much lower, with the highest 0.3522 (p<<0.00001) and 0.3452 (p<<0.00001) at cycles 19 and 20. Sun spot energy variations on long time scales (van Loon et al. 2007), even with very weak changes, could produce decadal and millennial timescale impacts on global thermohaline circulation that in turn affect heat distribution (Bond et al. 2001, Gray et al., 2013).

**AMO.** The correlations were generally higher at the descending phase of the SS cycles, which is opposite to those for SS vs PDO, ONI, MEI, and the SST Anomaly. However, the highest $r^2$ occurred on ascending (A) and descending (D) phases of SS cycles, thus: 0.700 (p<< $10^{-10}$), 0.558 (p<< $10^{-10}$),

0.468 (p<< $10^{-10}$), 0.434 (p=0.03), 0.411 (p<<0.00001) and 0.191 (p=0.001) for cycles 20A, 22D, 19D,

24D, 21A and 23A, respectively. These high $r^2$ values show a strong degree of correlation, although lower than PDO correlations. The lag time was between 24-48 months. The results found could be explained as The Atlantic Multidecadal Oscillation index has the opposite phase to the PDO (Enfield et al., 2001; Condron et al., 2005); i.e. warm in periods 1930-1964 and 2000-present (cold PDO), and cold in 1965-1999 (warm PDO). The PDO and AMO are inter-decadal cycles of 25-30 years, therefore during the ascending and descending phase they strengthen or weakened.

**Conclusions**

**Period 1954-2017.** Over this period sun spot numbers have decreased from between 225 (SS#21)

and 110 (SS#24) at cycle maxima, to minima SS counts of around 20-25. Thus, the Earth is receiving decreasing solar energy over this almost 7-decade period. The reduction of SS peaks has been associated with the beginning of the Maunder Minimum (Mörner, 2015). Ineson et al. (2014) are projecting lower peaks for the next SS cycle (SS#25) and presently SS counts per month are as low as

1.6 (July 2018) and with an average of 8.5 (Jan-Aug 2018); counts are expected to decrease to 5.3 in

February 2019, actually it decreased to 0.8 (http://www.sws.bom.gov.au/Solar/1/6).

Monthly SS count correlations with SST, SST Anomaly (both 3.4 and 1+2), ONI, MEI, AMO and PDO

through the whole time series (1954-2017) were poor; these had a correlation $r^2$ values averaging

0.020 and a negative linear regression slope. Thus, in the long term there are no strong correlations between SS and PDO, MEI, ONI and SST Anomaly in 3.4 (correlation coefficients were between 0.043

and 0.021). In the case of region 1+2, the correlation was even poorer: <0.005.

The series of correlations at different lag times (6, 12, 24, 36 and 48 months) gave a response time (i.e. the lag time with highest correlation coefficients; Table 1) of 12-24 months for all indexes, except for AMO (48 months), which align to what was previously reported by Kristoufek, (2017), and

Huo and Xiao (2016);

Changes of the SS cycle could have climate impacts. Gil-Alana et al. (2014) have found no significant statistical relation between sun spots and global temperature; however, van Loon et al. (2007)

suggested that even though SS cycles produce weak changes on Solar Irradiation (SI) of about 0.07%

according to Gray et al. (2010), these can still produce decadal and millennial impacts on global thermohaline circulation (Bond et al. 2001, Gray et al., 2016), due chiefly to UV energy fluctuation (Ineson et al., 2014). The changes in UV (<100 nm to 350 nm) and near infrared (>800 nm to >1000

nm) are larger than in the visible range (>350 nm-800 nm) and could have an important impact on global climate (Ermelo et al. 2013).  It is therefore reasonable to expect some impact on the studied oceanographic indexes.  Recent data (Solar Radiation and Climate Experiment Satellite) suggest that the variability of UV radiation during the declining phase of cycle 23 was larger than previous estimates (Harder et al., 2009 and Haigh et al., 2010).  The SI variations are strongly correlated to SS, and even though these are relatively small (Hansen et al., 2013), they may impact surface ocean heat content because: 1) the total SI integrates over all the wavelengths, and 2) the heat capacity of the seawater is huge.  Also, UV radiation penetrates down to 75-100 m depth in the water column (Smyth, 2011), thus adding to the heat content of the deeper layers.

**Individual SS cycles (19-24)**. The SST shows some correlations against individual SS cycles in regions

3.4 and 1+2. In the first (Fig. 3), these were up to 0.219 (SS#23) whilst the lag time was 12-36 months in all cycles (except SS#19), in line with reports from Kristoufek (2017) and Huo and Xiao (2016). On the other hand, in region 1+2, the linear correlation $r^2$ was low at <0.0675 (SS#19), and highly inconstant between cycles. The SS and SST anomaly correlations in 3.4 and 1+2 (Fig. 4), showed important variability with highest values of 0.289 (in 3.4) and 0.396 (in 1+2) during cycles 19 and 24

respectively, with both having the same response time range. These values are within the cold phase of the PDO, suggesting that during this phase the signal from SS is clearer. In the period 1997-2016

the two strongest El Niño (1997-1998 and 2015) and La Niña (2000-2002 and 2010-2012), events occurred, and they were most evident in the 1+2 region. The slopes of the linear correlation were positive for 3.4 and negative for 1+2, and in general it was found that correlations were more inconstant and lower in 1+2 than in 3.4, also the linear regression curves show when SS approaches to zero the value of SST and SST anomaly as well as ONI and MEI, as well as PDO are negative.  These results do suggest that Sun spot activity can influence the SST and SST anomaly behavior, but the relatively weak signals may well reflect high seasonal and interannual variability in coastal oceanographic conditions (Ormaza-González and Cedeño, 2017) that obscures the correlation with

SS. In zone 3.4, correlations are better, although the influence of regional oceanographic and meteorological conditions will still be there as expressed through, e.g., the Southern Oscillation

Index – SOI - (Rasmussen and Carpenter, 1982; Barnston, 2015). The SOI temporal behaviour have also been associated to SS and could enhance or mar the oceanographic indexes of the equatorial

Pacific (Higginson et al., 2004). During cycle 24, the ONI index was highly correlated to SS ($r^2$ up to

0.2510) with a positive slope (Fig. 5).  The ONI index reached 2.6C (Nov-Dec-Jan 2015/2016) and -

1.7C (Oct-Nov-Dec 2010). In cycles 20-21 the $r^2$ was low ($r^2<0.04$); however, from 22 to 24, $r^2$

increased from 0.131 ($p<<0.00006$) to 0.251; thus, confirming the SS activity has a better correlation during the cold phase of PDO.

The PDO aligned best with SS cycles at lag times of 36-48 months during SS#19 (0.625) and SS#24

(0.766) when the PDO was in a cold phase, whilst lower correlations (0.467-0.508 for cycles 20-23)

were found when it is in a warm phase (1979-2000) or in between them. The North Atlantic index counterpart (AMO), gave a variable lower correlation $r^2$ ranged from 0.130 to 0.490 with a response time of 48 months for cycles 23 and 20 respectively. Gray et al. (2016) reported 36-48 months lag for mean-sea-level pressure in the North Atlantic in a study of 32 SS cycles. The slopes of the PDO and

AMO linear regression curves are negative and positive respectively in cycle 21 and 22, but in concordance in 19, 20, 23 and 24 during the cold phase of the PDO. These two interdecadal oscillations proved to be correlated to SS, with the PDO having a higher correlation. Presumably, as the North Pacific Ocean basin has a larger area than the North Atlantic the correlations and lag times may reflect the higher heat storage capacity in the North Pacific.

**Ascending and descending phases**. The SSTs in 1+2 showed higher correlations with SS in the ascending phases, relative to the descending phase ($r^2$ of up to 0.205 and below 0.067 respectively).

In region 3.4, there were again high degrees of correlation of SS and SST ($r^2$ between 0.87 and 0.56), during the ascending phase of the cycles, with a response time of 24-36 months seems to occur at the low or high plateau of the cycles (Fig. 10). The highest $r^2$ of 0.870 in the descending phase in cycle 24 coincided with the strongest El Niño (2015) and the second highest (SS#22) with two consecutive strong El Niño events in 1991-1995; the third and fourth highest corresponded with el

Niño and warm events respectively in 1955-1957 and 1997-1998.  It seems that short time expressions of SS cycles, either at the beginning of their ascending or descending phases, have a trigger effect on the SSTs. This was observed through the polynomial regression curves (Fig. 10) that were found for each SS cycle, as the SI (equation 1) increases and decreases the amount of heating of surface waters follows suit. The polynomial curves (6th order) were fitted with an average $r^2>0.89$.

Thus, the warm events El Niño of: 1957-1958 (SS# 19), 1965-1966 (SS# 20), 1981-1982 (SS# 21),

1987-1988 and 1991-1992 (SS# 22), 1997-1998 (SS# 23), 2015-2016 (SS# 24). On the other hand, the cold events of La Niña tend to occur after an El Niño at the middle of the ascending phases (1988-

1989, 1999-2001, 2010-2012) or when approaching the minimum of the cycles (1973-1974, 1975-

1975; 1995-1996, 2017-2018). The so called equatorial Pacific neutral conditions in 3.4 (see, https://iridl.ldeo.columbia.edu/maproom/ENSO/ENSO_Info.html), seems to span a longer period after La Niña, and vice versa after El Niño.

The ENSO indexes ONI and MEI also showed strong correlations to the ascending phase of the SS

cycles, with a lag time of 24-36 months. The regression linear curves for ascending and descending phases, shows that these indexes are negative when SS number are in the range 0-50, whilst the highest positive values are somewhere between 150-200 sun spots per month. In this analysis, it was besides found that warm events tend to occur in the ascending phase (SI increases) or at the top of the cycle and have a delay time of 24-36 months (as also reported by Huo and Xiao, 2016), whilst cold events are mostly associated with a descending phase (SI decrease) but with a quicker response time of 0-12 months. The MEI index has a similar pattern to the ONI, but with lower correlations that may arise as the MEI takes into consideration six variables that in combination may mask the signal from sun activity.

The linear regression curves (Fig. 11) at SS close to zero the ONI is somewhere between 0C to -2C, whilst in the range 50-200 sun spots the ONI is predominantly positive. Over all, for all ascendant phase together gives an $r^2$ (p<0.01) of 0.11, in some cycles (22 and 23) the $r^2$ was 0.6.  We deem this a clear evidence how SS (read solar radiation) affect the studied indexes.

The PDO was linearly correlated to SS ($r^2$ ranging from 0.285 to 0.768) in the ascending phase with lag times of 24-36 months (Huo and Xiao, 2016), whilst correlations with AMO varied with $r^2$

between 0.160 to 0.700. Similarly, the response time was 24-48 months. These results correspond with those of van Loon et al. (2007), who established that even a small change (1.5 W m$^{-2}$) in sun activity (SI) could produce decadal and millennial time scales influences on thermohaline circulation (Bond et al. 2001, Gray et al., 2016); nonetheless, the Intergovernmental Panel on Climate Change:

IPPC (2001) considers this fact too small to drive climate changes. These influences of this change can be reflected by PDO and AMO indexes, which are found to be in the opposite phase to PDO

(Enfield et al., 2001; Condron et al., 2005). The ascending and descending phases of SS could re- inforce or weaken these indexes.

Recent predictions for an El Niño event in the late northern hemisphere summer of 2018 (see:

http://www.bom.gov.au/climate/enso/) did not occur; then, projections were pushed back to the beginning of autumn (http://www.cpc.ncep.noaa.gov/products/precip/CWlink/MJO/enso.shtml)

and most recently late autumn and winter 2019. Thus, most current models are failing to provide a consistent projection. This is likely related to two re-cooling events in all El Niño areas, that have kept the ONI index within the realm of ENSO neutrality (-0.5C to 0.5C). Also, the PDO indexes have been negative, averaging -0.53, and it is now in its cold phase (https://www.ncdc.noaa.gov/teleconnections/pdo/). This coincides that during 2017 the average smoothed SS counts per month was 21.8, and for 2018 it was 8.5 with many weeks without any SS.

In July the average SS count was just 1.6  (http://www.sws.bom.gov.au/Solar/1/6) with counts expected to decrease to 5.3 in February 2019 (actually they were just 0.8). Therefore, the input of solar heat has been at its lowest values since the 1950s, and its trigger effect on ENSO system is not enough for a full-fledge El Niño event; id est, meteorological (SOI) and oceanographic (ONI or MEI)

are not connected for around three consecutive months (Australian Bureau of Meteorology, http://www.bom.gov.au/climate/enso/#tabs=Overview) . At the time of writing this paper, the expected and modelled 2018 full-fledged El Niño 2018 did not occur, it did not happen, actually the region 3.4 and 1+2 cooled down by the end of 2018. During March 2019, the latest report from http://www.bom.gov.au/climate/enso/#tabs=Overview is saying "The El Niño–Southern Oscillation (ENSO) remains neutral. However, the Bureau's *ENSO Outlook* remains at El Niño WATCH, meaning there is approximately a 50% chance of El Niño developing during the southern hemisphere autumn or winter…". Nonetheless, NOAA view is that a weak El Niño conditions are present now and to expect to continue through the Northern Hemisphere spring 2019 (~55% chance), https://www.cpc.ncep.noaa.gov/products/analysis_monitoring/lanina/enso_evolution-status-fcsts- web.pdf. The fact is, El Niño is not fully-fledged. At this moment (march 24, 2019) the oceanographic and meteorological variables have not fully couple yet.  The ENSO modellers should take into account in some way the presence of SS or any variable that could measure the variability of SI as an input to the models, specially the determinists ones.

[revised manuscript text omitted]

26. Hansen J, Kharecha P, Sato M, Masson-Delmotte V, Ackerman F, et al. (2013) Assessing

''Dangerous Climate Change'': Required Reduction of Carbon Emissions to Protect Young

People, Future Generations and Nature. PLoS ONE 8(12): e81648.

doi:10.1371/journal.pone.0081648

27. Henley, B.J., Gergis, J., Karoly, D.J., Power, S., Kennedy, J., and Folland, C.K.: A Tripole Index for the Interdecadal Pacific Oscillation, Clim. Dynam., 45, 3077, doi:10.1007/s00382-015-

2525-1, 2015.

28. Higginson, M. J., M. A. Altabet, L. Wincze, T. D. Herbert, and D. W. Murray (2004), A solar (irradiance) trigger for millennial-scale abrupt changes in the southwest monsoon?Paleoceanography,19, PA3015, doi:10.1029/2004PA001031.

29. Horning, N., Russell, C., and Goetz, S.: Energy from the Sun to Earth's Surface. In Chapter 2:

Earth's Radiation Balance and the Global Greenhouse, https://people.ucsc.edu/~mdmccar/migrated/ocea80b/public/lectures/lect_notes_1/03_En ergy_Balance_MDM_11F.pdf (last access: 25 April 2018), 2003.

30. Huang, B., Banzon, V.F., Freeman, E., Lawrimore, J., Liu, W., Peterson, T.C., Smith, T.M.,

Thorne, P. W., Woodruff, S. D., and Zhang, H. M.: Extended Reconstructed Sea Surface

Temperature version 4 (ERSST.v4): Part I. Upgrades and intercomparisons, J. Climate, 28,

911–930, doi:10.1175/JCLI-D-14-00006, 2014.

31. Huang, B., Thorne, P.W., Banzon, V.F., Boyer, T., Chepurin, G., Lawrimore, J.H., Menne, M.J.,

Smith, T. M, Vose, R. S., and Zhang, H. M.: Extended Reconstructed Sea Surface

Temperature, Version 5 (ERSSTv5): Upgrades, Validations, and Intercomparisons, J.

Climate, 30, 8179–8205, doi:10.1175/JCLI-D-16-0836.1, 2017.

32. Huo, W.J., and Xiao, Z.N.: The impact of solar activity on the 2015/16 El Niño event,

Atmospheric and Oceanic Science Letters, doi: 10.1080/16742834.2016.1231567, 2016.

33. Ineson, S., Maycock, A. C., Gray, L. J., Scaife, A. A., Dunstone, N. J., Harder, J. W., Knight, J. R.,

Lockwood, M., Manners, J. C., and Wood, R. A.: Regional climate impacts of a possible future grand solar minimum, Nat. Commun., 6, 7535, doi:10.1038/ncomms8535, 2015.

34. Intergovernmental Panel on Climate Change (IPCC) (2001), Third Assessment Report–Climate

Change 2001, The Scientific Basis, Cambridge Univ. Press, New York.

35. Jia, X. and Ge, J. (2017), Modulation of the PDO to the relationship between moderate ENSO

events and the winter climate over North America. Int. J. Climatol, 37: 4275-4287.

doi:10.1002/joc.5083

[revised manuscript text omitted]

58. Ramírez, I.J. & Briones, Understanding the El Niño Costero of 2017: The Definition Problem and Challenges of Climate Forecasting and Disaster Responses, F. Int J Disaster Risk Sci 8:

489. https://doi.org/10.1007/s13753-017-0151-8, 2017.

1. Rasmussen, E. M., and Carpenter, T. H.:  Variations in tropical sea surface temperature and surface wind fields associated with the Southern Oscillation/El Niño, Mon. Weather

Rev., 110, 354-384, 1982.

2. Scafetta, N.: Global temperatures and sun spot numbers. Are they related? Yes, but non- linearly. A reply to Gil-Alana et al. (2014), Physica A., 413, 329-342, doi:

10.1016/j.physa.2014.06.047, 2014.

[revised manuscript text omitted]

---

## Short Comment (SC2) · 3 Apr 2019

We would like to let you know how many reads (views) has had our paper during the discussion time. There have been around 500 views and downloads of the paper, and most of the readers have been USA, where ENSO and PDO are important issues of the Pacific Ocean. So far, nobody has made a comment. We think this is telling the outcomes of the paper are or could be accepted. We deserved some credit for that.

Please see the metrics section.

---

## Author Comment (AC5)

**Do sun spots influence the onset of ENSO and PDO events in the Pacific Ocean?**
**Franklin Isaac Ormaza-González and María Esther Espinoza-Celi**

Final response.

Dear Editor and Reviewers.
First at all, we would like to express our gratitude for the time and patience you have taken to revised and comment our paper. All your comments and suggestions have been very useful. Secondly, we think we have responded if not all most of them with much respect and argumentatively.

Thirdly, we would like to annotate that the during the time the paper has been under discussion, there has been around 500 reads and above 50 fully downloads (Fig. 1) and nearly 100 views and reading in Research Gate Portal (https://www.researchgate.net/publication/328700803_Do_sun_spots_influence_the_onset_of_E NSO_and_PDO_events_in_the_Pacific_Ocean/stats). In the discussion space, there was not any comment in favor or against it and its contents. We think this fact gives credit to the manuscript.

We have already submitted two new versions to AR1 and AR2. Thus, complying with referees' request. The latest version with minor modifications we are going to submit after this phase is over.

**Fig. 1.** Statists records of article views.

[Figure]

Fourthly, we strongly think the paper is contributing to ocean science, in a subject such as the ENSO, which is very important to understand better every day. Right now, there are warm anomalies in area 3.4 that are above El Niño threshold (>0.5C), however the SOI is not fully

connected so far to these anomalies (Fig. 2., http://www.bom.gov.au/climate/enso/). This index has to be <-7.5 for at least 3 consecutive months. Right now (10/April) the latest value over 30 days is -2.6 and getting close to 0.0 quickly. *Id est*, there is not a fully fledge El Niño. The SOI is correlated with sun spots (Higginson *et al.*, 2004). On the other hand, in 1+2, the SST anomalies have been under 0.5C on average, last week was even negative, the ZCIT has moved to 3-5N Latitude, signaling the presence of trade wind and Humboldt from the south and Cromwell undercurrent from west. One of the reasons for what said above is that sun spot are at the lowest numbers since decades ago. During February 2019 the Sun has 28 days spotless (https://spaceweatherarchive.com/2019/03/06/a-month-without-sunspots/).

**Fig. 2**. SOI behavior in time. http://www.bom.gov.au/climate/enso/#tabs=SOI.

[Figure]

Finally, we cannot have any doubts the sun affects directly the SST of sea water. For some reason Galileo Galilei on June 23 of 1613 draw the first ever sun spots (Fig. 3), although Chinese astronomers u observer had registered since 28 BC or even Anaxagoras probably made an account on them in 467 BC (https://www.jstor.org/stable/227857?seq=1#page_scan_tab_contents). Bertrand Russel (1917) said: When we look at the sun we wish to know something about the sun itself, which is ninety-three million miles away….

Thank you.

**Do sun spots influence the onset of ENSO and PDO events in the Pacific Ocean?**
**Franklin Isaac Ormaza-González and María Esther Espinoza-Celi**
Final response.

**Fig. 3.** Galilei drawing of sun spots on 23 June 1613 showing the positions and sizes of a number of sunspots. See, http://galileo.rice.edu/sci/observations/sunspot_drawings.html.

[Figure]

**References**

Bertrand Russell (1917), The Ultimate Constituents of Matter (An address delivered to the Philosophical Society of Manchester in February, 1915. Reprinted from The Monist, July, 1915. Reprinted in his Mysticism and Logic (London: George Allen & Unwin Ltd.: 1917). Also reprinted Totowa, New Jersey: Barnes & Noble Books, 1951. Page 100.)

Higginson, M. J., M. A. Altabet, L. Wincze, T. D. Herbert, and D. W. Murray (2004), A solar (irradiance) trigger for millennial-scale abrupt changes in the southwest monsoon? Paleoceanography,19, PA3015, doi:10.1029/2004PA001031

**Do sun spots influence the onset of ENSO and PDO events in the Pacific Ocean?**
**Franklin Isaac Ormaza-González and María Esther Espinoza-Celi**
Final response.